# Seasonal and diurnal performance of daily forecasts with WRF V3.8.1 over the United Arab Emirates

Oliver Branch[1], Thomas Schwitalla[1], Marouane Temimi[2], Ricardo Fonseca[3], Narendra Nelli[3], Michael Weston[3], Josipa Milovac[4], Volker Wulfmeyer[1]

[1]Institute of Physics and Meteorology, University of Hohenheim, 70593 Stuttgart, Germany

[2] Department of Civil, Environmental, and Ocean Engineering (CEOE), Stevens Institute of Technology, New Jersey, USA

[3] Khalifa University of Science and Technology, Abu Dhabi, United Arab Emirates

[4] Meteorology Group. Instituto de Física de Cantabria, CSIC-University of Cantabria, Santander, Spain

Correspondence to: Oliver Branch (oliver_branch@uni-hohenheim.de)

**Abstract.**

Effective numerical weather forecasting is vital in arid regions like the United Arab Emirates (UAE) where extreme events like heat waves, flash floods, and dust storms are severe. Hence, accurate forecasting of quantities like surface temperatures and humidity is very important. To date, there have been few seasonal-to-annual scale verification studies with WRF at high spatial and temporal resolution.

This study employs a convection-permitting scale (2.7-km grid scale) simulation with WRF-NOAH-MP, in daily forecast mode, from 01 January to 30 November 2015. WRF was verified using measurements of 2-m air temperature (T-2m), 2-m dew point (TD-2m), and 10m wind speed (UV-10m) from 48 UAE WMO-compliant surface weather stations. Analysis was made of seasonal and diurnal performance within the desert, marine, and mountain regions of the UAE.

Results show that WRF represents temperature (T-2m) quite adequately during the day-time with biases ≤+1˚C. There is however a nocturnal cold bias (-1 to -4˚C), which increases during hotter months in the desert and mountain regions. The marine region has the lowest T-2m biases (≤-0.75˚C). WRF performs well regarding TD-2m, with mean biases mostly ≤1˚C. TD-2m over the marine region is overestimated though (0.75-1 ˚C), and nocturnal mountain TD-2m is underestimated (~-2˚C). UV-10m performance on land still needs improvement, and biases can occasionally be large (1-2 m s$^{-1}$). This performance tends to worsen during the hot months, particularly inland with peak biases reaching ~3 m s$^{-1}$. UV-10m are better simulated in

the marine region (bias ≤1 m s⁻¹). There is an apparent relationship between T-2m bias and UV-10m bias, which may indicate issues in simulation of the day-time sea breeze. TD-2m biases tend to be more independent.

Studies such as these are vital for accurate assessment of WRF nowcasting performance and to identify model deficiencies. By combining sensitivity tests, process and observational studies with seasonal verification, we can further improve forecasting systems for the UAE.

## 1 Introduction

In a changing climate, effective numerical weather forecasting is vital in arid regions like the United Arab Emirates (UAE), to predict low-visibility events like fog and dust (e.g., Aldababseh and Temimi, 2017; Chaouch et al., 2017; Karagulian et al., 2019), and extreme events relating to storms and flash floods (Chowdhury et al., 2016; Wehbe et al., 2019), high temperatures, and droughts. These extreme events are expected to become more prevalent under a changing climate (Feng et al., 2014; Zhao

et al., 2020). In fact, climate projections suggest that arid and semi-arid regions are likely to expand in area along with rising temperatures (Huang et al., 2017; Lelieveld et al., 2016; Lu et al., 2007). Hence, it is vital that regional weather forecasting and climate simulations with regional climate models (RCMs) correctly simulate important quantities which characterize extreme events, especially surface temperatures, humidity, winds, and precipitation.

The model chain and configuration used in any simulation can heavily influence the results of such forecasts. Important factors

include, but are not limited to, RCM type (e.g., Coppola et al., 2020), general circulation model (GCM) dataset for boundary forcing (Gutowski et al., 2016; Jacob et al., 2020), horizontal and vertical grid resolutions (e.g., Schwitalla et al., 2017b), physics and dynamics schemes (e.g., Chaouch et al., 2017; Schwitalla et al., 2020), soil/land use/terrain static data, as well as internal model parameter sets for important land surface processes (e.g., Weston et al., 2018).

The Weather Research and Forecasting (WRF) model (Powers et al., 2017; Skamarock et al., 2008) has been used in arid

regions for various forecasting and verification (e.g Branch et al., 2014; Fonseca et al., 2020; Schwitalla et al., 2020; Valappil et al., 2019; Wehbe et al., 2019), and process studies (Becker et al., 2013; Branch and Wulfmeyer, 2019; Karagulian et al., 2019; Nelli et al., 2020a; Wulfmeyer et al., 2014). Currently, there have been few annual-scale verification studies employing the WRF model on a NWP daily forecasting mode at such high spatiotemporal resolution (e.g. $dx < 2 - 3$ km). Horizontal grid scale in particular, is significant because simulations employing convection-permitting (CP) grid spacing $(dx \sim < 4$ km) are

known to outperform those at coarser resolutions, particularly in terms of clouds and precipitation - not least because they don't require a convection parameterization (Bauer et al., 2015, 2011; Prein et al., 2015; Schwitalla et al., 2011, 2017a; Sørland et al., 2018). Furthermore, it is known that land use, soil texture, and terrain interact with planetary boundary layer (PBL) processes in complex feedbacks (e.g., Anthes, 1984; Mahmood et al., 2014; Pielkel and Avissar, 1990; Smith et al., 2014) with

a strong level of land-atmosphere (LA) coupling thought to exist in this region (Koster et al., 2006). Representation of landscape structure and the associated LA feedbacks should therefore be significantly improved when using finer grid resolution. In terms of time scale, seasonal-to-annual simulations are costly, but provide a sufficient time series for robust statistical comparison with observations over different seasons.

This study employs a configuration of WRF, coupled with the NOAH-MP 'multi physics' land surface model (LSM), with modular parameterization options (Niu et al., 2011). In contrast to typical climate mode simulations, WRF is run here in a numerical weather prediction (NWP), or daily forecasting mode in order to keep conditions inside the domain closer to that of the forcing data (see Section 2.3.3 for further details). We also apply high quality/resolution boundary forcing data, improved static data for land use/soils and terrain, high frequency aerosol optical depth, and sea surface temperature data. This WRF configuration was employed and verified by Schwitalla et al., (2020) within a one-day case study of a physics ensemble.

Our main objective is to assess the seasonal and diurnal performance of WRF – both qualitatively and quantitatively – in reproducing surface air temperature, dew point and wind data from 48 WMO-compliant surface weather stations distributed over the UAE.

Another objective is to assess the model performance in different areas of the UAE – which was split broadly into three environments: 1. northern coastline and islands, 2. inland lowland desert areas, and 3. the Al Hajar mountains in the east. The aim is to investigate differences in performance due to expected differences in climate regimes within these zones, and their respective surface/landscape characteristics and how they are dealt with by WRF-NOAH-MP. Factors include, amongst others, the influence of sea surface temperatures in the warm and shallow Arabian Gulf (Al Azhar et al., 2016), representation of albedo (Fonseca et al., 2020) and roughness length parameters (Weston et al., 2019), and limitations in simulations over orography, particularly with respect to the wind field (e.g., Warrach-Sagi et al., 2013). The Al Hajar Mountains have a complex climate with regular coastal fog and convective events (e.g., Branch et al., 2020). Therefore, splitting verification into the above zones (in which the stations are quite evenly distributed, with 16, 14, and 18 stations, respectively) can yield further insights into model performance, and climate characteristics in different environments.

Through ambitious simulations and robust verification, we can gain valuable insights into the regional climate, model performance and take a step towards more skilful weather forecasting with WRF-NOAH-MP in the UAE.

The structure of this work is as follows: We start with our Materials and Methods (Section 2), showing maps of the study area and model domain (2.1), a description of the regional climate (2.2), the model chain, configuration and simulation method (2.3), verification data set (2.4), and verification methods (2.5). Then follows a results and discussion section (3), and finally a summary and outlook (4).

## 2 Materials and Methods

### 2.1 Study area and model domain

The region under investigation is the United Arab Emirates (UAE) located between 22.61−26.43˚N and 51.54−56.55˚E in the far northeast of the Arabian Peninsula (see Figure 1a), with the 48 surface verification stations being spread out across the country. The model domain is shown in Figure 1b and covers a much larger area, a) to be sure of excluding the area with the strong effects of the boundary forcing (i.e., relaxation zone) from the analysis, and b) to incorporate the large scale synoptic weather situation. The model uses a regular latitude-longitude grid and has corner grid cells located at 14.775˚N, 32.225˚N,
43.275˚E, and 65.725˚E

### 2.2 Regional climate

### 2.2.1 Synoptic climate

Weather in the wider region is controlled generally by four predominant patterns, including troughs originating from the Atlantic and Mediterranean Sea in winter, locally forced convective storms over the UAE/Oman Al Hajar Mountains in
summer, and the southerly summer monsoon and cyclones from the Arabian Sea during June and October (Bruintjes and Yates, 2003; Steinhoff et al., 2018). These phenomena are represented in large-scale seasonal climatologies (1979 – 2014 - 8:00 UTC) in Figures 2 and 3 (right-hand panels). To represent the climate, we have used geopotential height at 500 hPa, wind velocity at 850 hPa and mean sea level pressure. Note that winter is represented exclusively by the months of January and February, because these are the months used for our winter analysis during 2015 – for reasons of temporal continuity.  In the climatology,
we can clearly see a typical winter January-February (JF) heat-low centred over Turkey and Iraq and a trough extending down toward the Arabian Peninsula. During summer June, July and August (JJA), we observe much higher temperatures further south, with a heat-low centred over Iran and the UAE. The other two seasons appear are transitional periods.

### 2.2.2 UAE climate

The UAE climate is generally characterized by scarce precipitation and high temperatures. However, annual cycles do exist
with maxima of precipitation and minima of temperatures in winter and the converse in summer. Annual UAE precipitation is between 20 mm in the drier west to 130 mm in the higher Al Hajar Mountains of the east, mainly produced in the winter-spring time period (Sherif et al., 2014). During summer, subtropical subsidence leads to a strong reduction of precipitation and higher temperatures, and consequently summer precipitation represents only around 20% of the annual amounts. However, upper level disturbances from the southern monsoon flows can still transport moisture towards the Arabian Peninsula and the UAE
(Böer, 1997; Schwitalla et al., 2020), and convection is initiated sporadically over the mountains of Oman and the UAE in summertime (Branch et al., 2020).

The neighbouring Arabian Gulf to the north of the UAE also plays a strong role in regional weather conditions. The prevailing winds from the Arabian Gulf are westerly or northwesterly between January and May, but these change to north-westerly and then northerly directions from June toward November. In the Arabian Gulf, which is relatively shallow (maximum depth ~90m), particularly close to the UAE coast, the sea surface can heat rapidly, with temperatures often exceeding 30˚C (Al Azhar et al., 2016). Prevailing winds are augmented by strong sea/land breezes, which develop due to land/sea temperature gradients. Daytime sea breezes can penetrate up to 50 km inland (Eager et al., 2008).

## 2.3 Model chain and simulation method

### 2.3.1 Model chain and physics

The model chain is based on the Weather Research and Forecasting model (WRF, Powers et al., 2017; Skamarock et al., 2008) version 3.8.1 using the Advanced Research WRF (ARW) core, which solves the Euler equations on a discretized horizontal grid, with a terrain-following vertical coordinate system. The domain size and grid spacing matches that of a previous simulation by Schwitalla et al., (2020)), and is comprised of a regular latitude-longitude grid with 900 by 700 cells horizontally (see Figure 1b). In line with our previous statements on CP scale we selected a grid increment of $0.025˚$ ($dx \sim 2779$ m), with no parameterization of deep convection. It was important to extend the domain enough to incorporate influential synoptic conditions upstream to the north, east, and south east. Hence, our grid covers a region of approximately $2500 \times 1945$ km extending up to Iraq in the north, down to the south of Yemen, and well into Pakistan in the east. Care was taken, for reasons of model stability, that domain boundaries did not bisect very large peaks, especially in the complex terrain of Iran. Vertically, 100 levels were used, adjusted so that at least 25 levels were present in the lower 2000 m –to maximise resolution of the strong moisture gradients in the boundary layer and lower troposphere.

WRF was coupled with the NOAH-MP LSM (Niu, 2011) to simulate land-surface processes and land-atmosphere feedbacks. NOAH-MP provides a separate vegetation canopy defined by a canopy top and ground layer including a modified energy balance closure approach. It offers a tile approach where the net longwave radiation and turbulent fluxes are calculated separately for bare soil and the canopy layer. The calculated fluxes over vegetated grid cells are then bulked as a weighted sum of bare soil and canopy fluxes. Furthermore, NOAH-MP is partially modular in structure, providing a suite of optional schemes for several processes, such as radiation budget calculation, stomatal resistance, snow albedo, and others. The same configuration of Milovac et al., (2016) was used for all NOAH-MP options.

Other physics schemes included were RRTMG for long and shortwave radiation transfer (Iacono et al., 2008; Mlawer et al., 1997), the Thompson-Eidhammer microphysics scheme (Thompson and Eidhammer, 2014) (although without the aerosol-

140 aware component activated), the MYNN scheme for the atmospheric surface layer, and the MYNN 2.5 level TKE scheme for the boundary layer (Nakanishi and Niino, 2006) (See Table 1 for a synopsis of physics schemes and their associated references).

### 2.3.2 Initialization and forcing data

### 2.3.2.1 Initial and lateral boundary conditions

These were retrieved from the European Centre for Medium-Range Weather Forecasts (ECMWF) Integrated Forecasting
System (IFS), in the form of 6-hourly operational analysis data on the 41r1 cycle, on model levels. The horizontal grid increment is 0.125˚ (~12 km) with 137 vertical levels up to 0.01 hPa. Soil moisture and soil temperatures are also provided by this model, which assimilates satellite soil moisture data (Albergel et al., 2012) into its coupled Hydrology-Tiled ECMWF Scheme for Surface Exchange over Land (HTESSEL) model (Balsamo et al., 2009).

### 2.3.2.2 Sea surface temperatures (SSTs)

These data were retrieved from the OSTIA project (Donlon et al., 2012) – the data has a 1/20˚ horizontal resolution at a 12-hourly frequency at 00:00 and 12:00 UTC. This data is particularly important in coastal regions like the UAE.

### 2.3.2.3 Aerosol optical depth (AOD) data

These data were retrieved from the ECMWF Monitoring Atmospheric Composition and Climate (MACC) reanalysis (Inness et al., 2013) which interacts with the shortwave radiation scheme to modify radiative transfer and diabatic heating - data has a
155 ~80-km horizontal resolution and a 6-hourly frequency starting from 00:00 UTC.

### 2.3.2.4 Soil texture data

These data are an update from the default Food and Agriculture Organization (FAO) dataset. The new data are based on the Harmonized World Soil Database (HWSD) v 1.2 at 30 arc second resolution, where all the mapping units are reclassified into 12 soil and 4 non-soil types following the United States Department of Agriculture (USDA) soil classification system, as in
the WRF model. For access to the data and more details see Milovac et al., (2018). The WRF default soil texture map based on the FAO data was used for the bottom soil layer.

### 2.3.2.5 Land use data

These data were provided as a combination of a high-resolution dataset for the Emirates of Abu Dhabi and Dubai, provided by the National Center for Meteorology (NCM), and the International Geosphere-Biosphere Programme (IGBP) Moderate

Resolution Infrared Spectroradiometer (MODIS) 20-class land use dataset, included within the WRF package (Figure 4). The Abu Dhabi dataset contained some classes which differed from MODIS IGBP, and these were first reclassified in a logical manner before overwriting the MODIS dataset within the UAE (see Schwitalla et al., (2020) for further details of this process).

### 2.3.2.6 Terrain data

**Here, we used** the Global Multi-resolution Terrain Elevation Data (GMTED) 2010 static dataset ((Danielson and Gesch, 2011))

### 2.3.3 Simulation method

The objective of this study was to run a series of daily forecasts with WRF for the period 01 January to 30 November 2015, with a discarded one-month spin up run from 01 December 2014. Note that December 2014 was not used for verification (observation data was in any case not available at that time. See Section 2.4). It also makes sense not to analyze a winter season split over two years.

The intention of carrying out such a long sequence was to produce a long enough dataset to provide sufficient data points for robust statistical analysis. Forecasts were carried out in a NWP mode, i.e., with daily cold starts - as opposed to a 'climate' mode, which has a single cold start at the outset. In NWP mode, a cold start was initiated each day at 18:00 UTC (22:00 LT) and run for 30 hours, i.e., 6+24 until 00:00 UTC the next day. The first 6 hours of each forecast (18:00 UTC to 00:00 UTC) were then discarded from the analysis. The 6 hours allows time for the atmosphere to spin up after each cold start – in particular for the residual boundary layer to develop and dissipate before the convective boundary layer starts to develop after sunrise (~06:00 LT), and for potential cloud development. Other UAE forecasting studies have also suggested 5-6 hours an appropriate period for model convergence in the UAE region (Chaouch et al., 2017; Weston et al., 2018). After discarding the first 6 hours, a forecast remains for analysis spanning the 24 hours of each day between 00:00 and 23:00 UTC (04:00 to 02:00 LT). See Table 2 for a summary of the simulation method.

By reinitialising the 3D state within the domain itself (as opposed to simply inputting lateral boundary conditions), we ensure the atmospheric state is closer to the forecast provided by ECMWF, than would be the case in e.g., typical climate mode simulations. In climate mode, which is driven only at the boundaries, the WRF simulations may diverge more strongly, particularly toward the centre of the large domain where the study area lies, unless some form of interior nudging were implemented (e.g., Lo et al., 2008).

An exception to the daily reinitialization of state variables was made with the soil moisture field, whose state was intentionally maintained from one successive day to the next, by overwriting the soil moisture state from 18:00 to the next day at 18:00, when the forecast is restarted. The intention is to reduce physical inconsistencies between the soil moisture forecast in the driving GCM model and that of WRF-NOAH-MP. Intuitively that may not seem a large issue given the aridity of the UAE. However, it becomes significant when convective precipitation occurs in WRF, and soils are wetted. Such convective events and flash floods are common in the UAE and Oman, particularly from May to September in the mountains, including during 2015 (Branch et al., 2020; Schwitalla et al., 2020; Wehbe et al., 2019). Hence, the NWP method is a worthwhile method of improving physical consistency. To summarize the NWP configuration: The soil moisture is overwritten at 18:00 from each consecutive day to the next, for the start of each new forecast. The lateral boundary conditions are as for a climate mode run, i.e., input every 6 hours from the forcing data. The atmospheric state within the domain boundaries is reinitialized each day at 18:00.

## 2.4 Datasets for verification

Hourly verification data comes from 48 surface weather stations throughout the UAE (Figure 1a and Appendix Table A1) - quality checked and made available by the National Center for Meteorology (NCM) in Abu Dhabi, UAE. Fields available include air temperature at 2m (T-2m), dew point at 2m (TD-2m) representing humidity, and wind speed at 10m (UV-10m). Data covers the entire period of January 01-November 30 2015. Unfortunately, quality checked observation data for December 2014 was not available and so in the interest of preserving contiguous seasons, the month of December 2015 was omitted from the winter statistics.

## 2.5 Verification method

An aim of the study is to assess WRF's performance on several timescales: annually (January-November), seasonally, day-time and night-time periods, and hourly. Another aim is to assess performance within different regions of the UAE. The exclusive assessment of overall forecast means over the UAE may be valuable, but could obscure variability within the different regions, such as the capturing of high day-time temperatures in the inland deserts, or cooler and windier coastal conditions.

Accordingly, the dataset was split temporally and spatially, as follows.

### 2.5.1 Temporal analysis

2.5.1.1 **Yearly analysis**

Here, all time steps were analysed from 01 January to 30 November (hourly interval).

### 2.5.1.2 Seasonal analysis

Here, we present the most extreme seasons in terms of air temperatures - the (coolest) winter period of January 01-February 28 2015 and the (warmest) summer period of 01 June to 31 August 2015.

### 2.5.1.3 Daytime and night-time periods

For daylight hours we used all hours between 02:00 and 13:00 UTC (06:00-17:00 LT) - and for night-time, 14:00 to 01:00 UTC (18:00-05:00 LT). These hours were selected based on the range of UAE sunrise and sunset which range between ~05:30 and 07:00 LT, and ~17:00 and 18:50 respectively. The intention of separating day and night hours in this way is to examine performance during the nocturnal stable and day-time convective boundary layers. Indeed, several simulations in arid regions have demonstrated nocturnal cold biases and an overestimation of day-time wind speeds (Branch et al., 2014; Schwitalla et al., 2020; Weston et al., 2018).

### 2.5.1.4 Regional analysis

We split the 48 UAE weather stations into 3 regions – marine, mountain, and desert – based upon on surface geophysical characteristics and proximity to water bodies (See Figure 1a). Accordingly, the following criteria were used for grouping the weather stations into regions:

- **Marine** – located on islands or ≤ 5 km inland from the UAE coast, **17 stations.**
- **Mountain** – located in the Al Hajar Mountain area and ≥ 200 m ASL, **16 stations**
- **Desert** – located > 5 km distance inland and < 200 m ASL, **15 stations.**

The only exception made to this classification was for a single station located at 204 m near the sand dunes of Liwa, in the south of the Abu Dhabi emirate. Although the station is quite high, it is remote from the Al Hajar Range and was deemed more suitable for a desert classification. Details on altitude of the regional station groups can be found in Table 3, and a list of individual stations in the Appendix. The desert region is characterised by barren or sparsely vegetated soils (as is most of the UAE), high surface temperatures, and rapid night-time cooling due to radiative losses associated with a dry atmosphere. The Al Hajar mountain region is arid, has generally rocky bare slopes, and lower albedo (e.g., Moody et al., 2005), with gravel plains running along the west side (Sherif et al., 2014).

One can assume some similarity between these regions, particularly when the synoptic situation is relatively homogeneous over scales larger than the study area. Nevertheless, given the large number of stations and length of time series, if regional differences do exist then they should be evident.

### 2.5.2 Verification and Diagnostics

All comparisons were made using NCAR's Model Evaluation Tools V9.0 (MET) package, utilizing a nearest-grid cell approach on an hourly temporal resolution.

To obtain a visual overview of model performance, in terms of closeness of fit, spread of forecast errors, and distribution of residuals, scatterplots divided by region and day/night period are shown in Figure 5. Included are a line of best fit for the data, a 1:1 line of perfect fit, and a 95% confidence ellipse. Then, we plotted regional seasonal statistics of the mean observations (T-2m, TD-2m, and UV-10m) (Figure 6).

To quantify the regional forecast/observation association, error magnitude, and sign during day/night, we show three standard
statistical diagnostics (Pearson correlation coefficient, root mean square error (RMSE), and bias).

The **Pearson correlation coefficient 'r'** measures the strength of linear association between forecast ($f$) and observation ($o$), at all stations at each time step, given as:

$$r = \frac{\sum_{i=1}^{ns} (f_i - \bar{f})(o_i - \bar{o})}{\sqrt{\sum_{i=1}^{ns} (f_i - \bar{f})^2} \sqrt{\sum_{i=1}^{ns} (o_i - \bar{o})^2}}$$

where $f_i$ and $o_i$ are the forecast and observation at each observation point $i$, $\bar{f}$ and $\bar{o}$ are forecast and observation averages, $ns$
indicates the total number of observations at each time step (i.e., number of stations), and overbars indicate the mean. Occasionally $ns$ was reduced slightly whenever a missing value occurred.

The **root mean square error (RMSE)** is a scale-dependent diagnostic defined simply as the square root of the mean square error (MSE) of the forecast:

$$RMSE = \sqrt{MSE} = \sqrt{\frac{1}{ns} \sum_{i=1}^{ns} (f_i - o_i)^2}$$

The **Bias** is a measure of overall error, including sign, defined as:

$$Bias = \frac{1}{ns} \sum_{i=1}^{ns} (f_i - o_i) = (\bar{f} - \bar{o})$$

These diagnostics were generated for 2015 for the region and time period and their temporal distribution expressed in boxplots (Section 3, Figure 7) showing mean, median, 25%-75% percentiles (box range), and 5% and 95% percentiles (whiskers).

Finally, a closer look at the diurnal evolution of the forecast is useful to investigate performance at specific times of day such as local noon and at PBL transition periods, where models often have biases. Hence, we generated mean hourly cycles of the spatial mean and spatial standard deviations for both forecast and observations. The mean at each hour is calculated as:

$$Mean(h) = \frac{1}{T}\sum_{t=1}^{T}\frac{1}{ns}\sum_{i=1}^{ns} o_i \ or \ f_i$$

The spatial standard deviation (σ) at each hour is given as:

$$\sigma(h) = \frac{1}{T}\sum_{t=1}^{T}\sqrt{\frac{1}{ns-1}\sum_{i=1}^{ns}(o_i - \bar{o})^2} \qquad or \ f_i - \bar{f}$$

For the diurnal analysis, we selected the two most extreme seasons in terms of temperature - the (coolest) winter period of January-February (Figure 8) and the (warmest) summer period of June-August (Figure 9), 2015. Again, these figures are divided by region.

## 3 Results and Discussion

In this section, we present a discussion of the results. Before examining the model performance however, we first discuss the study period of 2015 in context of the long term climate and El Niño (3.1) to assess the representativeness of the 2015 study period. We then discuss differences in regional climate and their significance to our verification (3.2). Finally, we evaluate the regional model output of T-2m, TD-2m and UV-10m fields across the seasons and time of day (3.3).

### 3.1 2015 in context

Our study period is 2015 from 01 January to 30 November (during which time the full verification dataset was available). 2015 was considered one of the strongest El Niño periods since 1950 (L'Heureux et al., 2017) with an Oceanic Niño Index (ONI) index of up to 2.6 towards the end of the year (see Table 4). A high positive ONI indicates a stronger El Niño event (negative indicate La Niña events). El Niño Southern Oscillation (ENSO) is known to impact upon the climate in this region, including temperatures and precipitation in the UAE (AlEbri et al., 2016; Almazroui, 2012; Chandran et al., 2016) so one might expect significant climate anomalies during 2015. Hence, a comparison was made between the long term climatology and the year

2015, based on ECMWF ERA5 reanalysis data. In Figures 2 and 3, from the geopotential height field, we can see that a positive 2015 winter temperature anomaly exists to the north of the UAE, extending from Turkey to the Caspian Sea (Figure 2, top left). However, conditions over the UAE show less deviation in terms of the temperature, pressure and wind fields. As the year progresses, and the ONI increases, the temperature anomaly becomes more pronounced further south, especially in JJA when higher 2015 temperatures extend further south toward Oman and Yemen than apparent in the climatology (Figure 3, top panels). Overall though, synoptic conditions over the Arabian Peninsula don't appear to be markedly different. They are similar enough in fact, to consider the 2015 regional climate as representative of the climate in general.

### 3.2 Regional and seasonal characteristics

An assessment of regional distributions reveals that clear differences in means and variability do exist (Figure 6). As expected, the marine region is dominated by the Arabian Gulf characteristics, with more moderate temperature maxima and minima (Figure 6a), greater humidity (Figure 6b), and higher wind speeds (Figure 6c) than the inland desert for instance (Figure 6). Hence marine temperatures are lower than at the desert stations in the summer months but remain higher in winter and autumn. In fact, the desert stations have the most extreme T-2m range in all seasons, reflecting the lower heat capacity surface, and consequent strong day-time surface heating. Rapid nocturnal cooling also occurs due to radiative losses in a much drier inland environment. The mountain region is only a little cooler than the desert (~1°C) in summer and autumn with the difference further reduced during spring and winter. The majority of mountain stations are located at fairly moderate altitudes (mean altitude 430 m, Table 3) with only one station located over 1000 m high (station ID 41229 - 1485 m ASL, see Table A1 in Appendix). Even so, one might have expected larger differences. However, there could be reasons other than the temperature lapse rate for this, such as differences in mountain and desert cloud cover for instance (Branch et al., 2020; Yousef et al., 2019), or in albedo (e.g Nelli et al., 2020b).

TD-2m, or dew point temperature, is a standard measure of humidity and is in most cases relatively independent of the ambient temperature. It is also a reliable measure of how humid the air feels in terms of human comfort (Wood, 1970). In a hot (and warming) climate like the UAE, forecasting TD-2m accurately is therefore important for society. Regionally, we observe considerable differences in TD-2m (Figure 6b), which are more or less expected due to coastal/land gradients and variation in vertical transport/distribution of vapor in different environments. Table 5 shows the difference in observed T-2m and TD-2m means. The inland atmosphere tends to be humid in summer when temperatures are high, but even closer to saturation in autumn and winter as temperatures fall, but humidity remains high. This seasonal range is particularly pronounced in the mountain regions reflecting the predominance of annual rainfall occurring during winter in the mountains and gravel plains of the north-eastern part of the UAE (Sherif et al., 2014; Wehbe et al., 2019). In winter and spring, the marine region is closer to saturation than in the other regions (T-2m minus TD-2m = -8 to -11°C); however, a reversal of this relationship occurs in summer and autumn as the mountain and desert regions become more humid.

There are significant regional differences in UV-10m, with marine UV-10m being 0.5-1 m s$^{-1}$ higher than in other regions (Figure 6c) and also more variable. This is not unexpected, due to low surface roughness, strong land-sea temperature gradients, and associated land-sea breezes. Desert UV-10m is the lowest all year round, and mountain UV-10m falls in between those of the desert and marine regions. In general, UV-10m is highest in spring and autumn.

These regional differences justify the need for regional splitting of the dataset and are further addressed below, in conjunction with model performance.

### 3.3 Model evaluation

Although the simulation of T-2m, TD-2m and UV-10m and causes for any biases may be physically linked, we nevertheless
first examine each field individually for clarity.**3.3.1 T-2m**

In the scatter plots (Figure 5a-5h) we observe that in the day-time, T-2m appears well estimated for the UAE on the whole (Figure 5a) (+0.44˚C) and errors are well distributed over the T-2m range. However, this agreement obscures some compensating regional biases; namely overestimation in the desert (+0.71˚C) and mountains (+1.06˚C), and underestimation in the marine region (-0.93˚C).

Reasons for the warm bias may be attributable to a combination of reasons. Firstly, a WRF overestimation of downwelling surface shortwave radiation has been observed before (Fonseca et al., 2020; Nelli et al., 2020b). This has been attributed to a lack of cloud cover, but may also relate to the performance of the radiative transfer scheme and interaction with aerosols. Secondly, the soil representation, such as soil texture classification – and associated parameters like heat capacity, thermal diffusivity, and albedo – may require adjustment. Underestimations of albedo in WRF have recently been observed particularly
for bright desert soils where measurements show typical albedo values of 0.3 to 0.34 (Nelli et al., 2020b). The WRF albedo value in this study is around 0.23 for much of the UAE lowlands, which would likely result in a too-high net radiation and sensible heating, especially on dry soils. This is consistent with the reported positive day-time temperature biases in the inland desert. A third factor may be the prescribed aerodynamic roughness length parameters used by WRF. Nelli et al., (2020a) found that a new value for the parameter, derived from eddy covariance measurements, reduced the warm day-time bias in
WRF simulations (Nelli et al., 2020b). These causes may account for some or all of the day-time temperature biases and therefore need to be considered for future simulations in this region.

Nocturnally, we observe a cold bias over the UAE (Figure 5e). This is quantified in Figure 7b as a mean negative bias of just over -2˚C. One can also see that this nocturnal bias tends to worsen with an increase in daily T-2m, which implies that the cold

bias gets worse in the hotter months. This is confirmed in the seasonal diurnal cycles (Figure 8a and 9a) where the mean nocturnal bias in winter is ~ -2˚C, but increases to greater than -4 ˚C in summer. This nocturnal cold bias is reflected in all sub-regions, but not to the same degree. The best nocturnal performance is in the marine region (Figure 5g) (bias of -0.75 ˚C), with an even error distribution across the temperature range. The largest nocturnal cold bias is in the desert region (-3.1 ˚C) (Figure 5h), with a steady increase in bias with temperature. The switch from positive to cold biases usually occurs more or less around the twice-daily transition times of the boundary layer between stable and convective states. Such arid nocturnal biases have been noted before (Branch et al., 2014; Fekih and Mohamed, 2017; Weston et al., 2018). It may be that a too-dry lower atmosphere results in a lower downward flux of longwave, as found by (Fonseca et al., 2020) in a comparison of WRF with radiation measurements. All else being equal this dryness would lead to a reduction of 'buffering' at night-time. They also found a too-high upward ground heat flux during the night, which could be associated with sub-optimal soil parameters or a too-strong soil-air temperature gradient. Overall, their net radiation losses at night were higher in WRF than from the radiation measurements.

### 3.3.2 TD-2m

TD-2m is relatively well estimated in 2015 over the UAE as a whole, with correlations around 0.7 and biases of less than 1˚C (Figure 7d and 7e, UAE sections). However, we can look at regional/seasonal differences for more detail. In the desert and marine regions, the biases are ≤1˚C during both day and night. Marine TD-2m is slightly overestimated in general, indicating the model to be more humid over the Gulf and coast than observed. Mountain nocturnal dew points are more of a problem with a negative bias of ~ -2˚C, and a larger error spread than the other regions (Figure 7e). There is also a corresponding T-2m nocturnal bias of ~ -2˚C which could indicate a deficiency in the longwave surface budget as just mentioned, but also a model deficiency in representing the intermittent shear-driven turbulence that appears in night-time stable boundary layers. However, such biases in complex terrain have been already well documented (e.g., Warrach-Sagi et al., 2013; Zhang et al., 2013). One of the reasons cited is that the CP scale is not fine enough to resolve mountain slopes, and therefore cannot capture certain processes in the same way that large-eddy scale models can, with grid spacings on the order of $\Delta x = 100m$. While such fine resolutions may be appropriate in a research context though, they may remain prohibitively expensive and inappropriate in the context of operational forecasting.

An additional problem in complex terrain is the validity of the traditional Monin-Obukhov similarity theory (MOST) (e.g., see Foken, 2006) that is typically used in atmospheric models, including WRF, for calculation of model diagnostics like T-2m or TD-2m. MOST assumes homogeneous underlying land surface and stationary fluxes, and there are multiple evidences that in complex and heterogeneous landscapes MOST needs significant improvements in scaling of turbulent kinetic energy profiles in the lowest part of the boundary layer (e.g., Figueroa-Espinoza et al., 2014; Wulfmeyer et al., 2018). The latter may affect

representation of the heat, moisture, and momentum transport from the land surface to the atmosphere, and if misrepresented
may lead to such high biases in the surface layer model diagnostics.

Seasonally, diurnal TD-2m is quite well reproduced in both winter and summer (Figures 8 and 9). The mountain nocturnal
negative bias becomes more significant in summer (Figure 9e). In the desert, a positive bias occurs over midday starting around
10 am LT (Figure 9k) showing an overestimation of water vapor in summer. This is likely to be too early in the day for a sea
breeze driven anomaly but may relate to simulated soil moisture being higher than reality. This was observed in a study by
Wehbe et al. (2018) that found a wet bias in dry soils and a dry bias in wetter soils in WRF over the UAE when not coupled
with a more advanced hydrological model.

### 3.3.3 UV-10m

WRF overestimates UV-10m during the day and night, in all regions and seasons. Positive biases of 1-2 m s$^{-1}$ are typical over
the whole year (seen in Figure 7h). Mountain day-time biases are strongest at 2 m s$^{-1}$, followed by day-time desert biases at
1.5 m s$^{-1}$. Marine biases are lowest with mean biases of <1 m s$^{-1}$. Notably, there is a trend where positive biases increase with
wind speed (Figure 5p, 5q, 5s). There is a significant increase in bias during the day-time, and also in the summer, particularly
in the mountain and desert regions (Figure 9f and 9i). In fact, the strongest wind biases occur in the same situations when day-
time T-2m is overestimated, particularly in the mountain and desert regions (Figures 7, 8, 9), hinting at a relationship between
the two. Indeed, it is likely that a too-strong sea breeze may account for this. During summer, the desert-marine T-2m day-
time gradient is highest (~5 ˚C, see Figure 9g and 9j, red curves) than in winter (~3 ˚C, see Figure 8g and 8j), although the
seasonal warm-biases are similar (~1.5-2 ˚C). The higher gradient coincides with a greater UV-10m bias in summer. Weston
et al., (2018) improved the duration and direction of UAE sea breezes by tuning a thermal roughness length parameter in WRF.
The PBL and surface layer parameterization schemes could also be a cause of the bias. Schwitalla et al., (2020) found an
overestimation of UV-10m in all members of a UAE physics ensemble, with magnitudes of around 1.5 m s$^{-1}$. The bias was
worse when using the MYNN 2.5 TKE PBL and MYNN surface layer schemes, when compared with the Yonsei University
(YSU) scheme (Hong et al., 2006) paired with the MM5 Jiménez surface layer scheme (Jiménez et al., 2012).

Using a non-local PBL scheme like YSU tends to produce a deeper and drier PBL with a stronger vertical mixing, in
comparison to local schemes like MYNN (see Milovac et al., 2016; Yang et al., 2017). This may lead to a reduction in wind
speeds, heat, and moisture close to the surface. However, another study however found that switching between 7 different PBL
schemes had little effect on positive UV bias (Shimada et al., 2011). One additional factor is that there are several parameters
within the MYNN scheme itself, which may benefit from retuning for arid regions like the UAE (e.g., Yang et al., 2017).
However, the total impact of the PBL scheme selection on reproduction of the T-2m, TD-2m and UV-10m diagnostics is not
completely clear. This is because the method of calculation of transfer coefficients/fluxes are executed in NOAH-MP, the PBL

scheme, and the surface layer scheme (SLS) depends on the land surface type. In WRF, PBL schemes are generally coupled to the SLS, and typically all variables between the land surface and lowest model layer are diagnosed (e.g. T-2m, U-10m, V-10m). These calculations in the SLS are based on Monin-Obhukov similarity theory, and are represented in the model as hard-coded parameters and/or formulations of similarity functions. The latter are used to obtain dimensionless bulk transfer coefficients which are used for calculating momentum, heat, and moisture fluxes, and for diagnosing near surface quantities like T-2m. These coefficients re-enter the LSM and are to calculate the surface fluxes which then enter the PBL scheme, as the lower boundary condition. Therefore, bias in near-surface variables is strongly related to the choice of LSM and SLS. In this WRF configuration, the communication link between the SLS and NOAH-MP is broken, as NOAH-MP itself calculates transfer coefficients and diagnostics over land surfaces, effectively bypassing the SLS (Nielson et al, 2013). The SLS only becomes active over water surfaces. This means that when NOAH-MP is used, the LSM probably has a stronger impact on the bias of near surface variables than the PBL and SLS (e.g. Milovac et al. 2016).

Incorrect aerodynamic roughness length parameters, as mentioned previously, may also play a large role in determining UV-10m – this parameter is used within the surface layer scheme. Nelli et al., (2020a) found positive wind speed biases over the same region when wind speeds were $< 4$ m s$^{-1}$ and negative biases for wind speeds which were $> 6$ m s$^{-1}$ within a WRF V3.8 simulation. We have a similar behaviour at night in the marine and desert regions, as exhibited by the positive-to-negative distribution of errors increasing with wind speed. Nelli et al., (2020a) reduced these biases by retuning the roughness length parameter based on eddy covariance measurements (Nelli et al., 2020b).

Another possibility is the length of the forecast spin-up, the required length of which may still be uncertain. We have already mentioned that Chaouch et al., (2017) cited a 5-h spin-up as being sufficient, but Hahmann et al., (2015) posits that the necessary spin-up over land could be 12 hours or even more (primarily for effective use of the PBL scheme). However, such long spin-ups are likely to be (i) prohibitively expensive and (ii) too time consuming for forecasting purposes.

## 4 Summary and Outlook

The aim of this study was to (i) assess the skill of WRF-NOAH-MP in reproducing surface quantities over the UAE, (ii) identify regional, seasonal, and diurnal differences in performance and (iii) estimate potential sources of model deficiencies. We have demonstrated the value of splitting the model evaluation temporally and spatially. For while assessment of diagnostics for the whole UAE region remains useful, it can obscure regional, diurnal and seasonal differences and also compensating biases, all of which are scientifically interesting, and importantly may reveal information on model performance with respect to specific processes and land surface types, and how they are simulated.

An analysis of model predictions has revealed that WRF-NOAH-MP represents the mean T-2m field reasonably well during the day-time, although with a tendency for slight overestimation (≤1˚C). The nocturnal T-2m is underestimated more strongly though (1-4˚C), and with larger biases during the hotter months, particularly in the desert and mountains, likely due to a combination of deficiencies. The marine region has the lowest T-2m biases, which is encouraging, and highlights the value of ingesting quality SST data, especially in coastal regions. WRF shows a good performance regarding TD-2m in general, with mean biases being ≤1˚C. Humidity over the marine region tends to be slightly overestimated though, whilst nocturnal mountain TD-2m is underestimated (bias ~-2˚C). UV-10m performance on land still needs be improved, with biases of 1-2 m s$^{-1}$. Furthermore, performance for UV-10m tends to worsen during the hot months, particularly inland. UV-10m in the marine region is generally much better simulated than in the other regions (bias ≤1 m s$^{-1}$). There is an apparent relationship between T-2m bias and UV-10m bias, and this could be due to deficiencies in sea-land breeze simulation. TD-2m biases appear to be more independent. The only exception to this is during the night, when T-2m and TD-2m biases do appear linked.

Ultimately, no model downscaling forecast (at scales economically viable for forecasting) can be expected to exhibit exceptional skill in all conditions. A caveat generally when evaluating models is that one must factor in a certain level of error in station or gridded observational datasets themselves (e.g., as discussed by Prein and Gobiet, 2017). Nevertheless, assuming a high level of observational accuracy, we have discussed several avenues for improvement on this application of WRF. For instance, we should continue to devise and ingest new and improved datasets for land cover, terrain and soil texture, and albedo. In particular, within a vegetation sparse region like the UAE, soil texture, moisture and other parameters are likely to be of prime importance. Certainly, ingesting SST data appears to have been valuable, given the lower coastal biases in all variables.

We have mentioned several very useful experiments carried out on parameters like aerodynamic and thermal roughness lengths (Nelli et al., 2020a; Weston et al., 2018), and also process-based observational studies related to the surface energy balance, and verification studies (Fonseca et al., 2020; Nelli et al., 2020b). Further experiments should now be coordinated in order to improve model predictions further. In terms of parameterization schemes, ensemble experiments (in the manner of Chaouch et al., 2017; Milovac et al., 2016; Schwitalla et al., 2020) are still required to identify optimal land surface/surface layer/PBL/microphysics combinations for arid regions. Such studies can also address the tuneable parameters defined inside parameterization schemes similarly to those conducted by Quan et al. (2016) and Yang et al. (2017). The most relevant ones can then be measured during dedicated field campaigns and subsequently ingested in the model.

Seasonal scale studies such as these are vital for accurate assessment of WRF nowcasting performance and to identify model deficiencies and areas for improvement. By combining seasonal verification with sensitivity tests, and process and observational studies, we will move towards improved forecasting systems for the UAE, and other arid regions.

## Appendix

**Observation stations**

See Table A1 for details on individual weather stations.

**Code availability**

**WRF** - To download the WRF source code, users need to register on the following website: http://www2.mmm.ucar.edu/wrf/users/download/wrf-regist.php.

The **namelist**.input file which is used for the WRF configuration, and **scripts for running WRF in NWP mode** are uploaded with open access to Zenodo:

DOI: 10.5281/zenodo.3894491

**Model Evaluation Tools V9.0 (MET)** open source - NCAR Research Applications Laboratory – Generation of verification statistics. Available from: https://ral.ucar.edu/solutions/products/model-evaluation-tools-met

**NCAR Command Language (NCL) V6.2** open source – Graphics, and used for overwriting soil moisture data when running NWP mode.

Available from: https://www.ncl.ucar.edu/

**ArcGIS V10.5** proprietary – Graphics and Mapping

Information: https://www.esri.com/en-us/arcgis/products/arcgis-desktop/overview

**Originlab 2020 V9.7.0.185** (Academic) proprietary – Statistical analysis and Graphics

Available from: https://www.originlab.com/index.aspx?go=Products/Origin

**Data availability**

**WRF output data** - available, on reasonable request as it is extremely large in size (many TB). It is archived on the German Climate Computing Center (Deutsches Klimarechenzentrum, DKRZ) and will be there for a minimum of 10 years.

**Verification data** - uploaded to Zenodo in the form of Excel files – open access. Data is courtesy of NCM, UAE:

Observation data

https://zenodo.org/deposit/3894544

Verification statistics dataset

https://zenodo.org/record/4004195

**Team List**

Oliver Branch[1], Thomas Schwitalla[1], Marouane Temimi[2], Ricardo Fonseca[2], Narendra Nelli[2], Michael Weston[2], Josipa Milovac[3], Volker Wulfmeyer[1]

[1]Institute of Physics and Meteorology, University of Hohenheim, 70593 Stuttgart, Germany

[2] Khalifa University of Science and Technology, Abu Dhabi, United Arab Emirates

[3] Meteorology Group. Instituto de Física de Cantabria, Santander, Spain

**Author contributions.**

O. Branch is the first author who conceived the experiment, carried out the simulations and analysis, and wrote the publication.

515    T. Schwitalla contributed greatly with scientific support and co-writing of the manuscript, provided much technical assistance, and formatted the observation data for use in the MET software. Marouane Temimi, Ricardo Fonseca, Narendra Nelli, Michael Weston, and Volker Wulfmeyer provided specialist scientific support and assisted with the drafting and improvement of key aspects of the manuscript.

**Conflicts of interest**

The authors declare that they have no conflict of interest.

**Acknowledgements.**

This material is based on work supported by the UAE Research Program for Rain Enhancement Science, under the National Center of Meteorology, Abu Dhabi, UAE. Furthermore, we are grateful to the High Performance Computing Center Stuttgart

(HLRS) for providing support and computing time on the XC40 system. We are also grateful to ECMWF for providing operational analysis data.

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

**Figures**

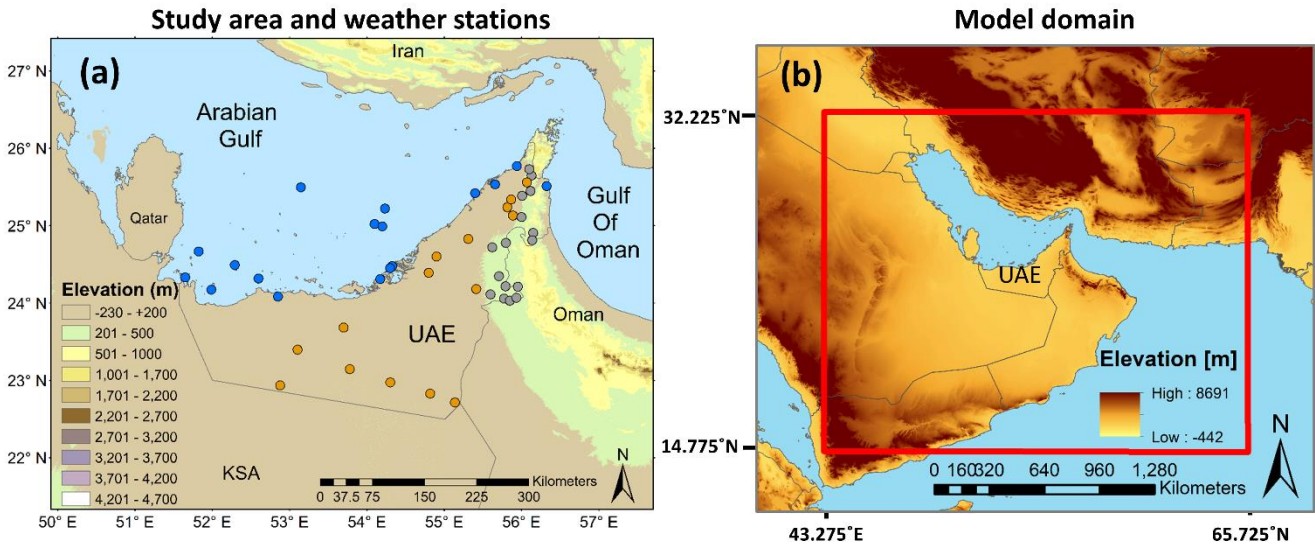

**Figure 1: Panel (a) is a closeup of the study area overlaid with classified topography and 48 UAE surface weather stations used for verification of WRF. Weather data was provided by the National Centre for Meteorology (NCM) in the UAE. The weather stations were grouped into geophysical regions for statistical analysis. The 17 blue dots indicate coastal/marine stations (criteria – on islands or within 5 km from coastline). The 16 grey dots are mountain stations (any station ≥200 m a.s.l. and > 5 km from coast). The 15 orange dots are inland desert stations (criteria –all remaining stations). Panel (b) is the 900 × 700 grid cell model domain ($\Delta x$ 2.7 km, 2430 × 1890 km). The four corner model grid cells are located at 14.775˚N, 32.225˚ N, 43.275˚E, and 65.725˚E.**

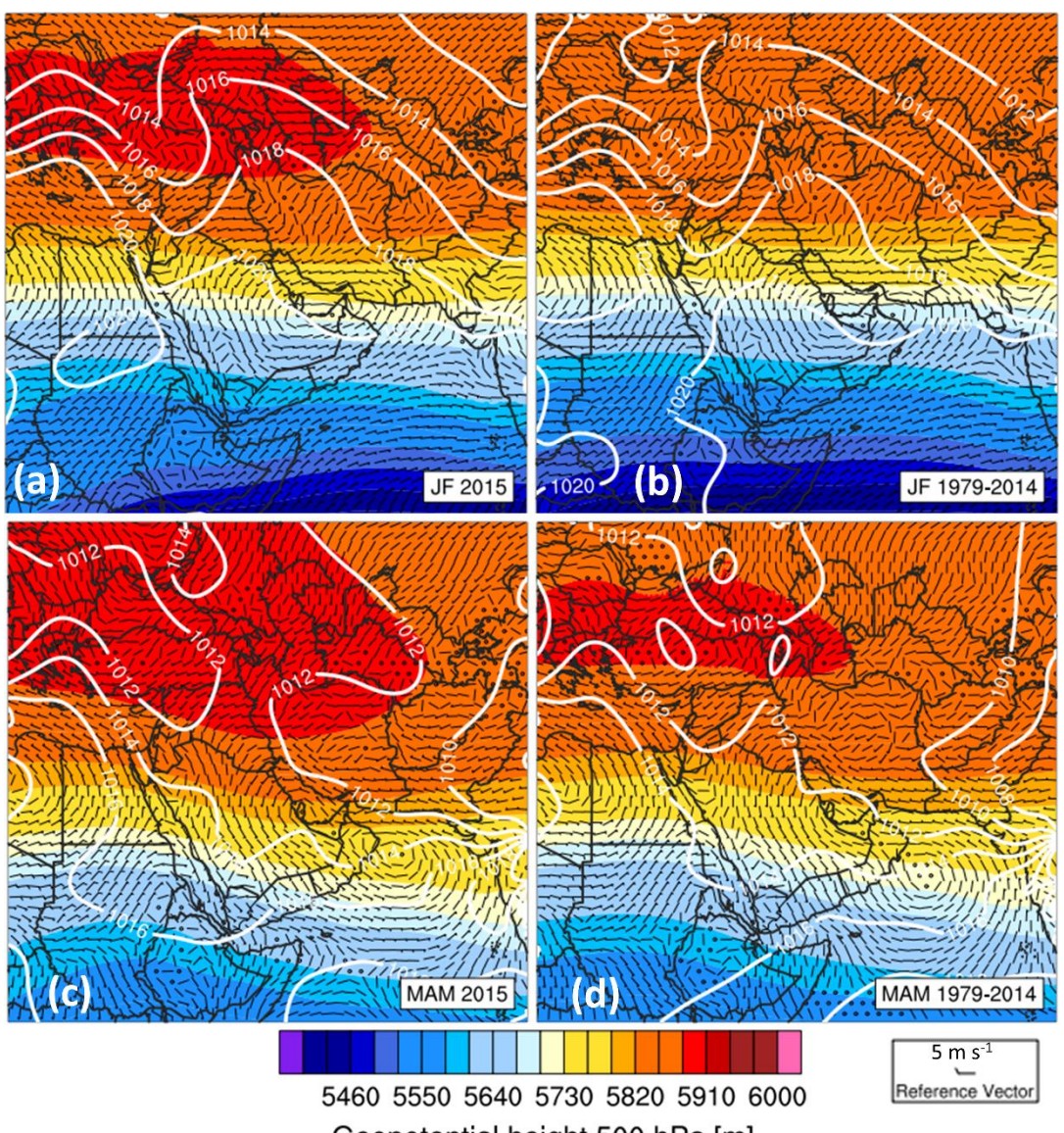

**FIgure 2: Comparison of the 2015 (a) winter (January-February, JF) and (c) spring (March-May, MAM) large-scale fields at 08:00 UTC. (b) and (d) are an equivalent 36 year climatology between 1979 and 2014. Variables shown are geopotential height at 500 hPa [m; shading], wind velocity at 850 hPa [m s-1, see reference vector at bottom right] and mean sea level pressure [hPa; white contours]. Data is taken from the ECMWF ERA5 reanalysis dataset.**

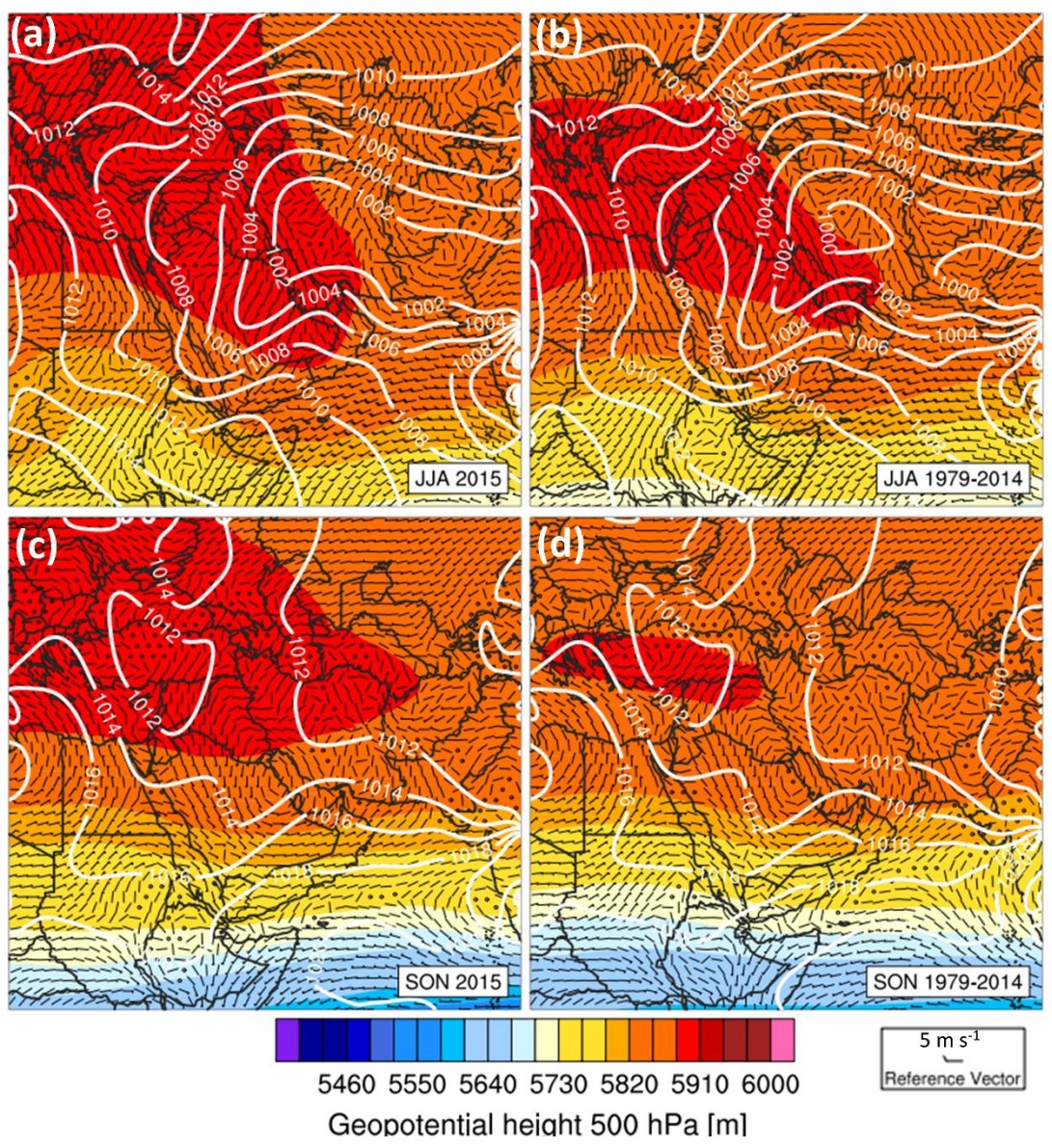

**Figure 3:** As for Figure 2 but for summer and autumn (Jun-Aug upper panels and Sep-Nov, lower panels). Data also taken from the ECMWF ERA5 reanalysis dataset.

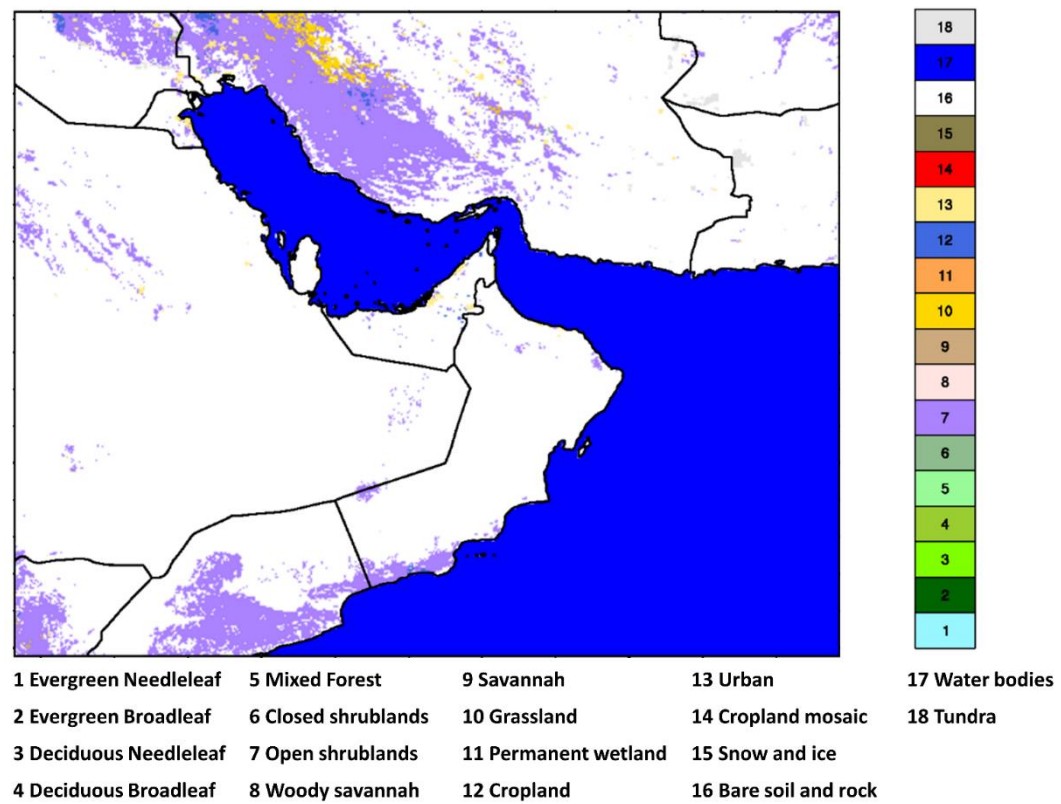

**1 Evergreen Needleleaf**   **5 Mixed Forest**   **9 Savannah**   **13 Urban**   **17 Water bodies**

**2 Evergreen Broadleaf**   **6 Closed shrublands**   **10 Grassland**   **14 Cropland mosaic**   **18 Tundra**

**3 Deciduous Needleleaf**   **7 Open shrublands**   **11 Permanent wetland**   **15 Snow and ice**

**4 Deciduous Broadleaf**   **8 Woody savannah**   **12 Cropland**   **16 Bare soil and rock**

**Figure 4: Map of whole model domain with the land cover data set used in the simulation. It is a composite of the standard 30 arc second (~1 km) IGBP 21 class MODIS dataset included as standard with WRF, with 2 local datasets superimposed: Abu Dhabi and Dubai Emirates, obtained respectively from the Environment Agency of Abu Dhabi (EAD) and the International Center for Biosaline Agriculture (ICBA) in Dubai. The local datasets were first reclassified in a logical manner into MODIS categories. 18 classes are shown here. There is a reduction in resolution due to the grid increment of 2.7 km.**

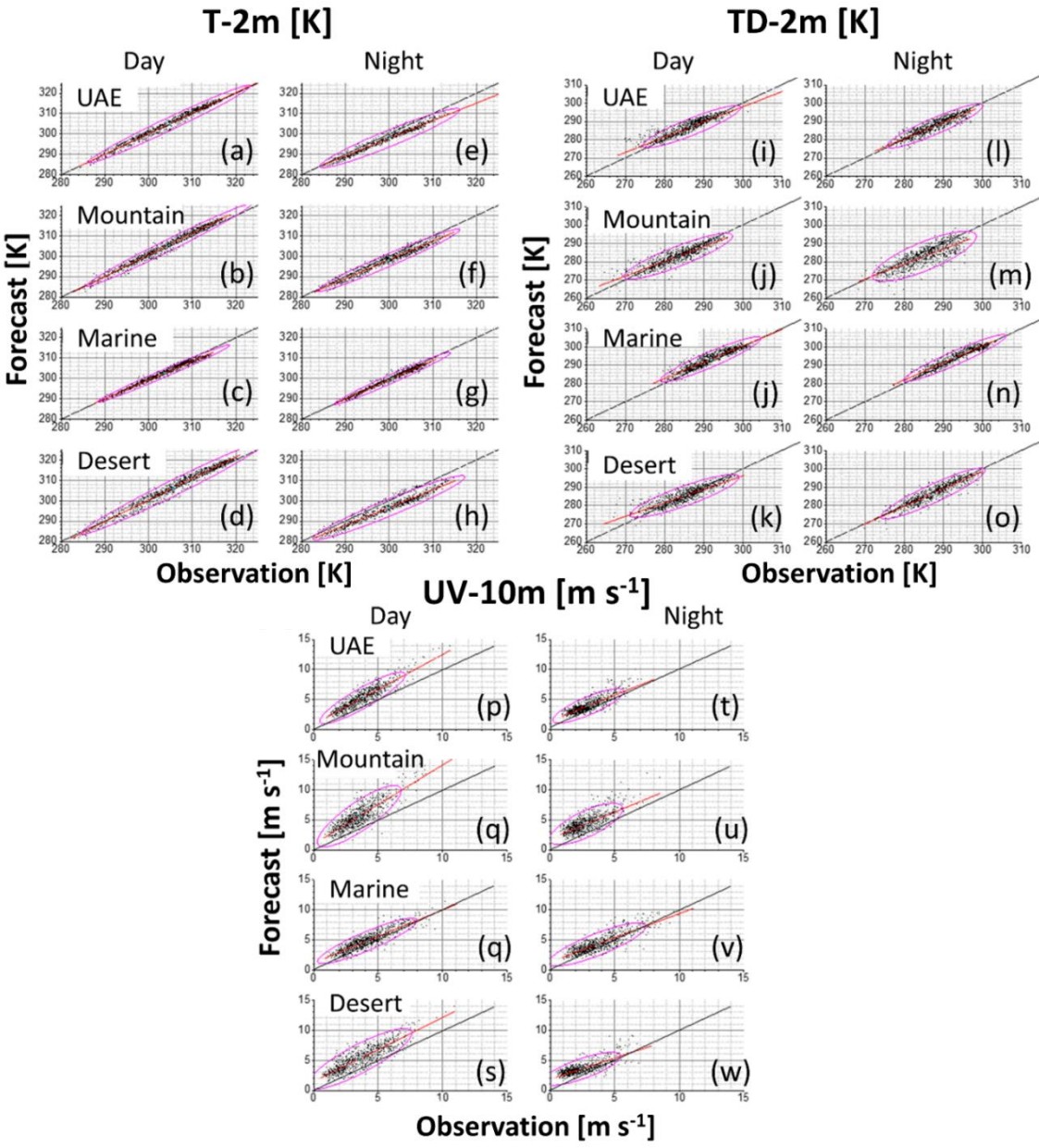

**Figure 5: Scatter plots of observation vs forecast for all time steps over the period of January-November 2015, comparing each weather station at the corresponding WRF grid point. The plots are split by day-time (left panels) and night-time periods (right) (respectively, day 06:00-17:00 (left panels) and night 18:00-05:00 (right) in local time), and by region (UAE, Mountain, Marine, Desert). The variables compared are 2-m air temperature (T-2m, K) in panels (a – h), 2-m dew point (T$_D$-2m, K) in panels (i – o), and 10-m wind speed (UV-10m, m s$^{-1}$) in panels (p – w). Also shown is a line of best fit (red) and a line of perfect fit (black), and 95% confidence ellipse (magenta).**

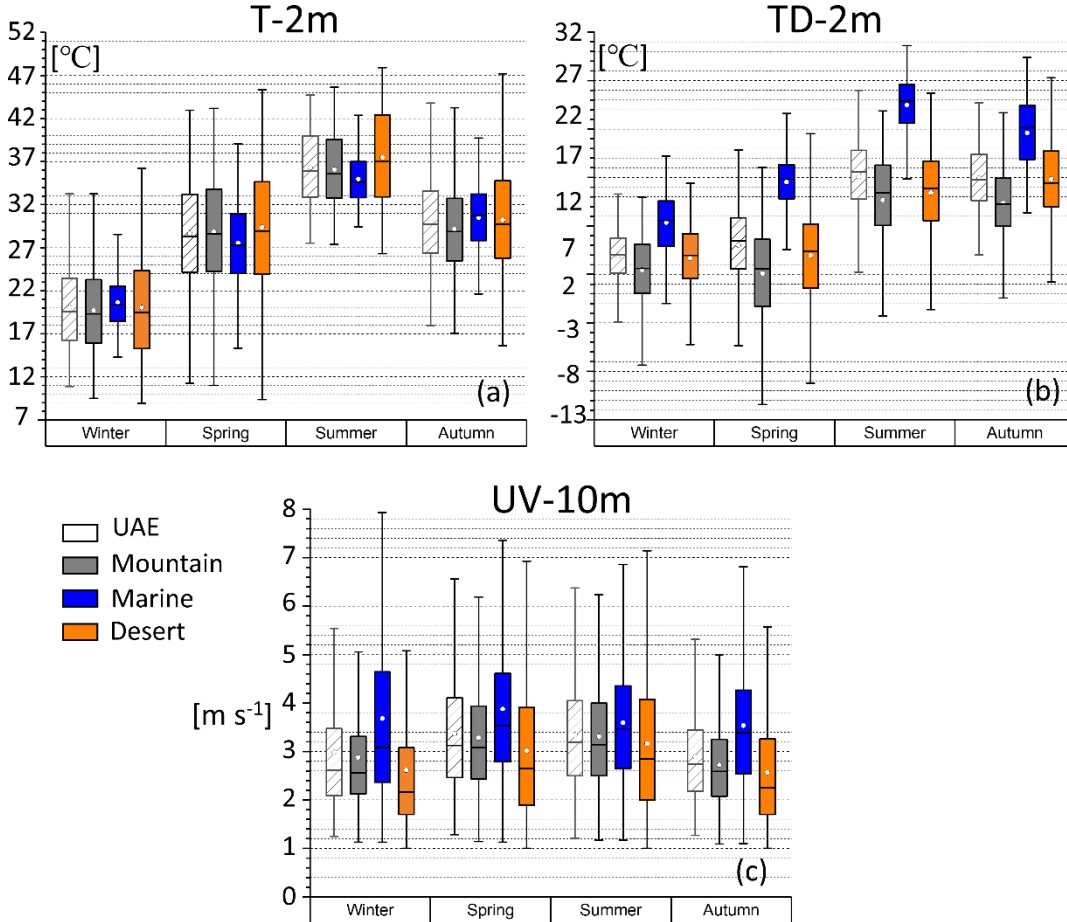

**Figure 6: Regional seasonal statistics of mean observations (T-2m (a), TD-2m (b), and UV-10m (c)). Box plots show the mean as a centre line, median as a dot, box ends are 25% and 75% percentiles, and whiskers are 5% and 95% percentiles.**

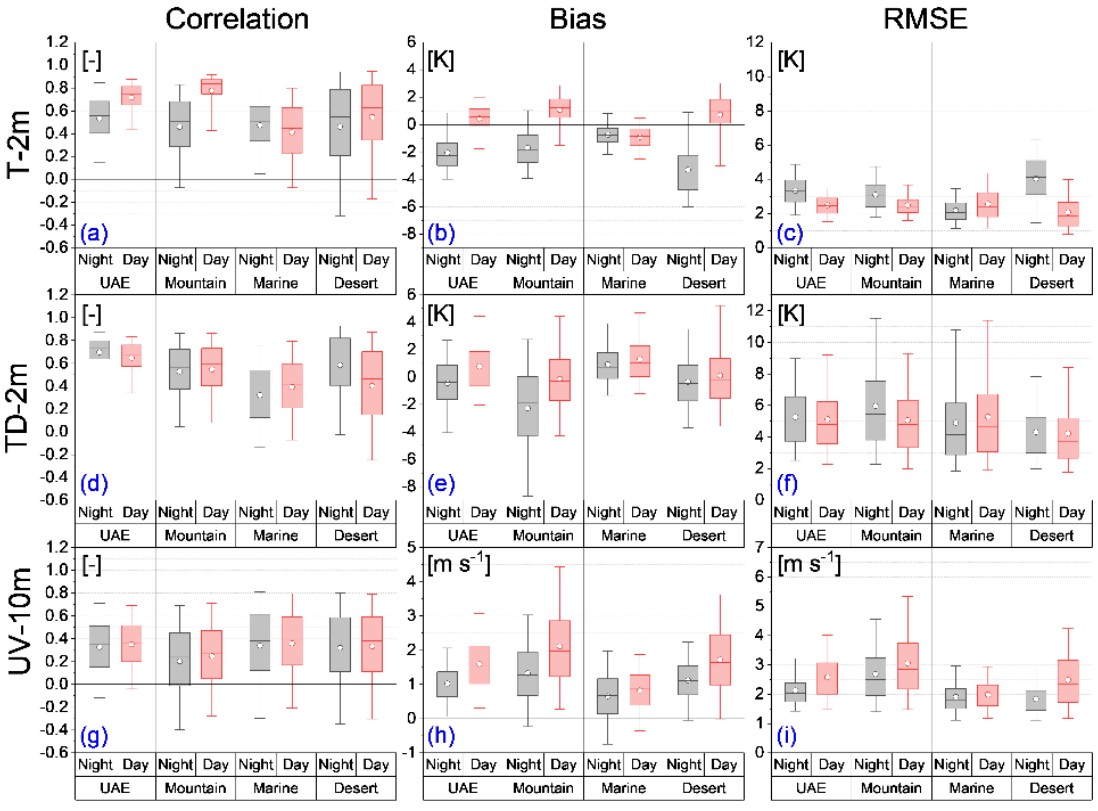

Figure 7: Box plots of T-2m, TD-2m, and UV-10m (respectively, panels (a-c), (d-f) and (g-i)) for all time steps over the period of January-November 2015. Statistics are divided by region (UAE, Mountain, Marine, Desert) and then by night-time and day-time hours (respectively, night 18:00-05:00 (grey boxes) and day 06:00-17:00 (red boxes) in local time). Statistics shown are Pearson correlation (left panels), Bias (centre) and RMSE (right). On the box plots the centre line represents the mean, the white circle is the median, box ends represent 25% and 75% percentiles and the whiskers are 5% and 95 % percentiles. Also marked is a horizontal zero reference line for the Pearson and Bias statistics.

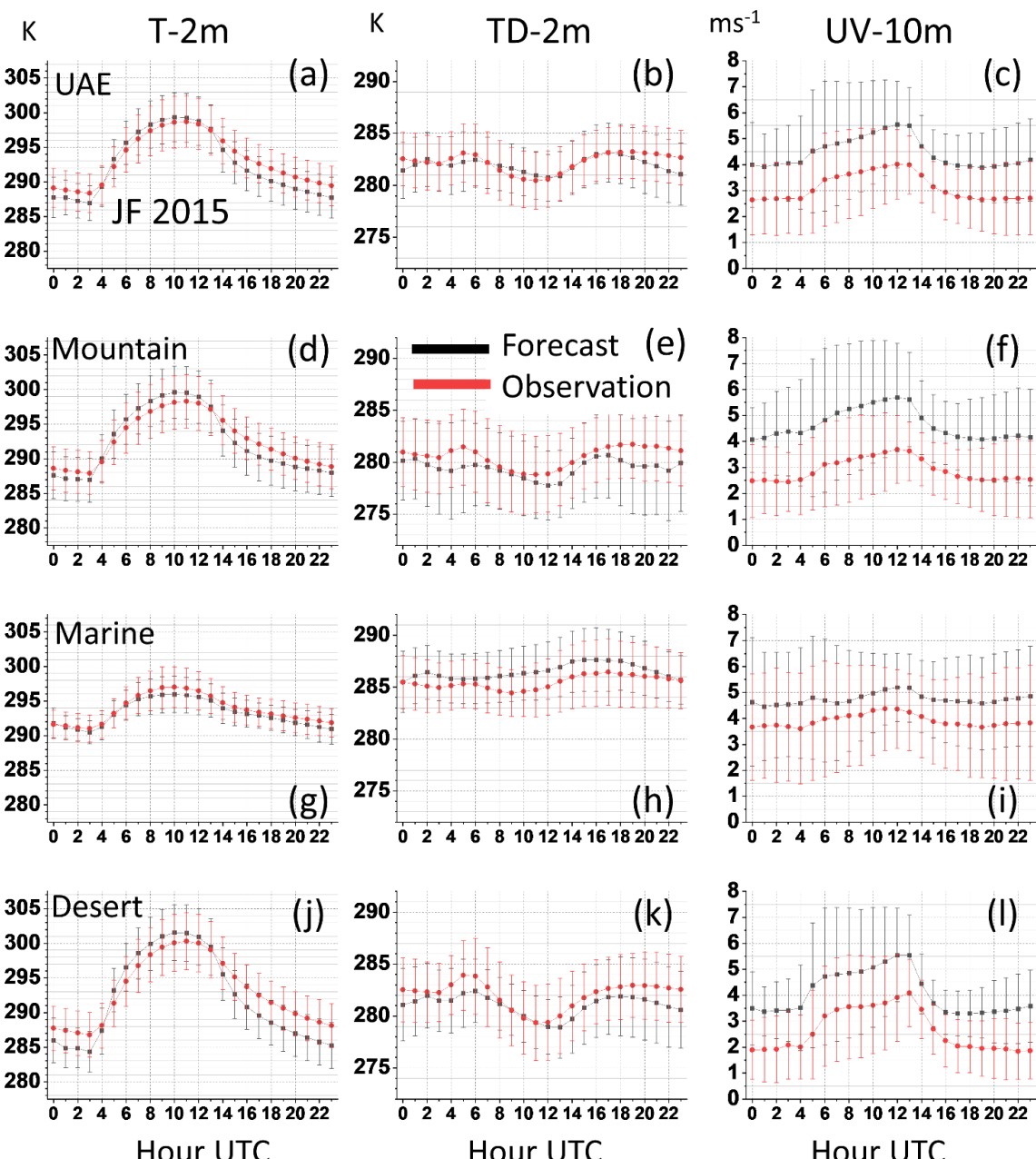

**Figure 8: Winter diurnal cycles of spatial mean values of forecast (black lines ) vs observations (red) - January-February, 2015. The error bars represent the mean spatial standard deviation for each hour. Variables shown are T-2m (K, left panels), TD-2m (K, centre) and UV-10 (m s$^{-1}$, right). Again the statistics are divided by region (UAE (top row), Mountain (2nd), Marine (3rd), Desert (4th)).**

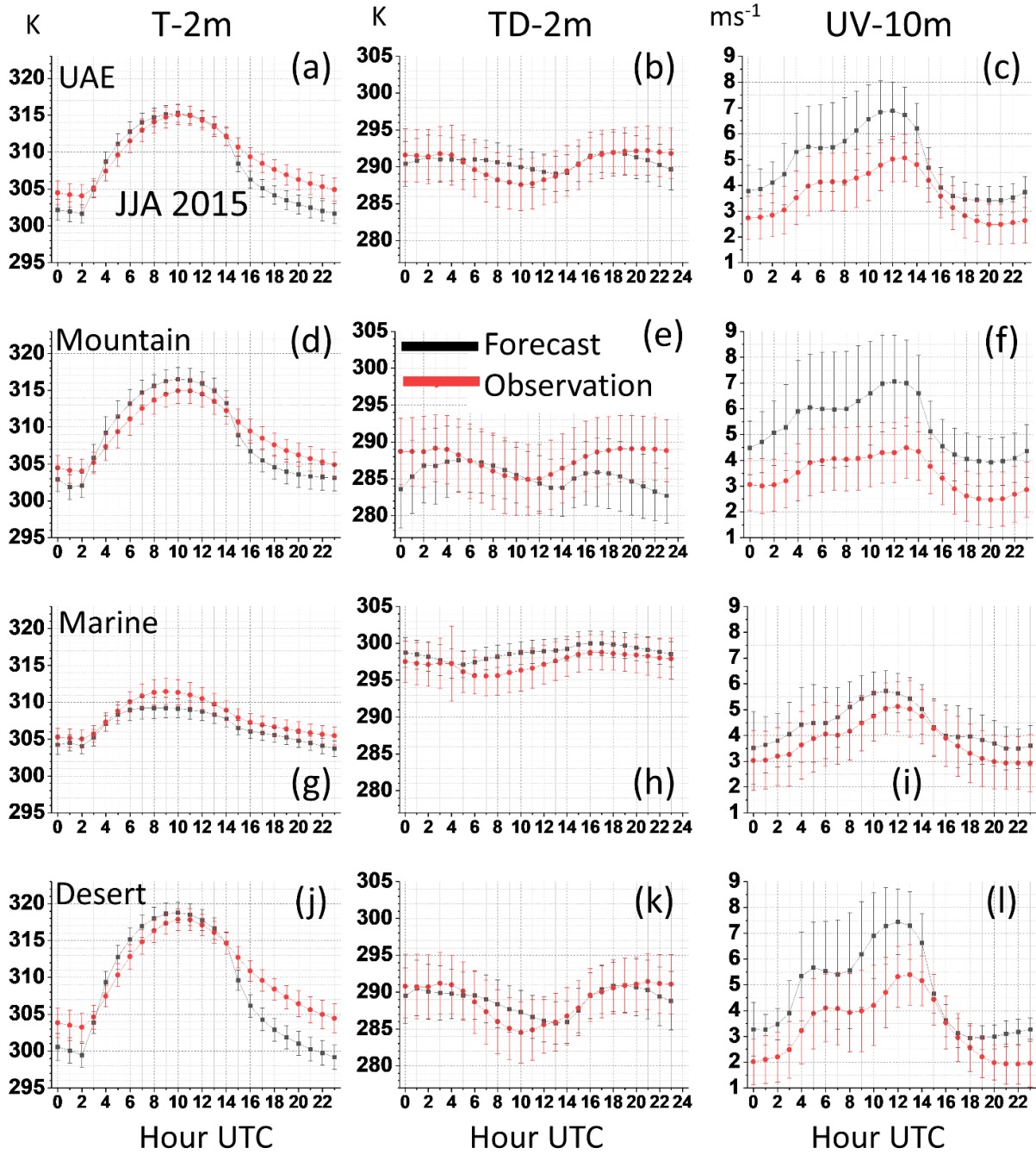

Figure 9: Summer diurnal cycles. As for Figure 8 except for the period June-August, 2015.

**Tables**

**Table 1: Selected physics schemes in WRF for sub-grid processes**

| Physics type | Scheme/Option | Reference |
|---|---|---|
| **Land surface scheme** | NOAH-MP | Niu et al., 2011 |
| **Atmospheric surface layer** | MYNN | Nakanishi and Niino, 2006 |
| **Atmospheric boundary layer** | MYNN 2.5 level TKE | Nakanishi and Niino, 2006 |
| **SW radiation** | RRTMG | Mlawer et al., 1997 |
| **LW radiation** | RRTMG | Iacono et al., 2008 |
| **Microphysics** | Thompson-Eidhammer | Thompson and Eidhammer, 2014 |

**Table 2: Summary of main aspects of simulation**

| | | |
|---|---|---|
| **Total duration of daily forecasts** | 01 December 2014 to 30 November 2015 | |
| **Period of analysis** | 01 January 2015 to 30 November 2015 | |
| **WRF output frequency** | 1-hourly | |
| **Verification data frequency** | 1-hourly | 48 surface weather stations |
| **Boundary forcing frequency** | 6-hourly | ECMWF operational analysis (0.12˚) |
| **SST forcing frequency** | 6-hourly | OSTIA data |
| **AOD forcing frequency** | 6-hourly | ECMWF MACC reanalysis |
| **Land use data** | Static | MODIS IGBP - 21 classes |
| **Soil texture** | Static | Modified HWSD (Milovac et al. 2018) |
| **Terrain** | Static | GMTED 2010 |
| **Cold start initialisation** | 18:00 UTC daily | |
| **Fields for reinitialisation** | All except soil moisture – all four soil levels | |
| **Forecast length** | 30 hours (first 6 hours discarded) | |
| **Forecast analysis** | 24 hours - 00:00 to 23:00 UTC | |
| **Model integration timestep** | 15 seconds | |

**Table 3: Number and altitude statistics for the regions – Marine, Desert and Mountain**

| Region | Number of stations | Mean altitude (m) | Minimum (m) | Maximum (m) |
|---|---|---|---|---|
| Marine | 17 | 13.8 | 0 | 101 |
| Mountain | 16 | 430.2 | 303 | 1485 |
| Desert | 15 | 120.0 | 114 | 204 |

**Table 4: 2015 Oceanic Niño Index (ONI) [3 month running mean of ERSST.v5 SST anomalies in the Niño 3.4 region (50˚N-50˚S, 120˚-170˚W)], based on centered 30-year base periods updated every 5 years – NOAA.**

| Jan | Feb | Mar | Apr | May | Jun | Jul | Aug | Sep | Oct | Nov | Dec |
|-----|-----|-----|-----|-----|-----|-----|-----|-----|-----|-----|-----|
| **0.6** | 0.6 | 0.6 | 0.8 | 1 | 1.2 | 1.5 | 1.8 | 2.1 | 2.4 | 2.5 | 2.6 |

**Table 5: Seasonal and regional differences in observed T-2m and TD-2m means to show the closeness to saturation. Included are the number of time steps for each period ($N_T$). Note that this is not a mean of the T-2m/TD-2m differences calculated at each time step, but an overall difference in means.**

| Season | Region | $N_T$ total | Mean (T-2m - Td-2m) [°C] |
|--------|--------|-------------|--------------------------|
| Winter | UAE | 1416 | 11.2 |
| Winter | Mountain | 1416 | 12.2 |
| Winter | Marine | 1416 | 8.3 |
| Winter | Desert | 1416 | 11.4 |
| Spring | UAE | 2207 | 18.6 |
| Spring | Mountain | 2207 | 21.7 |
| Spring | Marine | 2207 | 11.0 |
| Spring | Desert | 2207 | 20.4 |
| Summer | UAE | 2207 | 21.1 |
| Summer | Mountain | 2208 | 17.3 |
| Summer | Marine | 2208 | 18.2 |
| Summer | Desert | 2207 | 16.6 |
| Autumn | UAE | 2042 | 14.0 |
| Autumn | Mountain | 2182 | 10.2 |
| Autumn | Marine | 2176 | 14.5 |
| Autumn | Desert | 2051 | 11.6 |

**Table A1 (appendix): List of weather stations used for verification of WRF, including ID, coordinates, altitude and assigned region**

| Number | Name | Station ID | Lon | Lat | Altitude (m.a.sl) | Region |
|---|---|---|---|---|---|---|
| 1 | AlAryam | 41202 | 54.1719 | 24.3083 | 11 | Marine |
| 2 | AlDhaid | 41203 | 55.8169 | 25.2369 | 104 | Desert |
| 3 | AlFaqa | 41204 | 55.6214 | 24.7189 | 215 | Mountain |
| 4 | AlMalaiha | 41209 | 55.8881 | 25.1306 | 152 | Desert |
| 5 | AlQor | 41212 | 56.1519 | 24.9064 | 228 | Mountain |
| 6 | AlRuwais | 41214 | 52.8497 | 24.0833 | 13 | Marine |
| 7 | AlShiweb | 41215 | 55.7981 | 24.7761 | 292 | Mountain |
| 8 | AbuDhabi | 41217 | 54.3278 | 24.4772 | 8 | Marine |
| 9 | AlAin | 41218 | 55.7933 | 24.2156 | 302 | Mountain |
| 10 | Dalma | 41220 | 52.2914 | 24.4908 | 10 | Marine |
| 11 | Damsa | 41221 | 55.4133 | 24.18 | 169 | Desert |
| 12 | Dhudna | 41223 | 56.325 | 25.511 | 51 | Marine |
| 13 | FalajAlMoalla | 41224 | 55.8661 | 25.3378 | 96 | Desert |
| 14 | Hamim | 41225 | 54.3028 | 22.9736 | 115 | Desert |
| 15 | Hatta | 41226 | 56.138 | 24.811 | 304 | Mountain |
| 16 | JabalHafeet | 41227 | 55.7753 | 24.0567 | 910 | Mountain |
| 17 | JabalMebreh | 41229 | 56.1294 | 25.6469 | 1485 | Mountain |
| 18 | KhatamAlShaklah | 41230 | 55.9519 | 24.2111 | 406 | Mountain |

| 19 | MadinatZayed | 41231 | 53.6986 | 23.6817 | 113 | Desert |
| 20 | Makassib | 41232 | 51.824 | 24.666 | 0 | Marine |
| 21 | Manama | 41233 | 56.0081 | 25.3853 | 204 | Mountain |
| 22 | Masafi | 41234 | 56.1172 | 25.4475 | 453 | Mountain |
| 23 | Mezaira | 41235 | 53.7786 | 23.145 | 204 | Desert |
| 24 | Mezyed | 41236 | 55.8478 | 24.0286 | 316 | Mountain |
| 25 | Mukhariz | 41237 | 52.8778 | 22.9347 | 142 | Desert |
| 26 | Owtaid | 41238 | 53.1028 | 23.3956 | 145 | Desert |
| 27 | Qasyoura | 41240 | 54.8194 | 22.8286 | 95 | Desert |
| 28 | Raknah | 41242 | 55.7081 | 24.3456 | 282 | Mountain |
| 29 | RasMusherib | 41243 | 51.65 | 24.33 | 0 | Marine |
| 30 | SaihAlSalem | 41246 | 55.3119 | 24.8275 | 78 | Desert |
| 31 | SirBaniYas | 41248 | 52.5978 | 24.3169 | 101 | Marine |
| 32 | SirBuNair | 41249 | 54.2339 | 25.22 | 4 | Marine |
| 33 | Tawiyen | 41251 | 56.0703 | 25.56 | 164 | Desert |
| 34 | UmAzimul | 41252 | 55.1386 | 22.7142 | 114 | Desert |
| 35 | UmGhafa | 41253 | 55.9333 | 24.0667 | 361 | Mountain |
| 36 | UmmAlQuwain | 41254 | 55.6583 | 25.5333 | 12 | Marine |
| 37 | Yasat | 41255 | 51.9883 | 24.1722 | 15 | Marine |
| 38 | ALEjeili | 41256 | 54.1 | 25.02 | 0 | Marine |
| 39 | Ajman | 41258 | 55.4 | 25.42 | 0 | Marine |

| 40 | AlRass | 41259 | 54.3 | 24.45 | 3 | Marine |
|----|--------|-------|------|-------|---|--------|
| 41 | AlAjban | 41260 | 54.9 | 24.6 | 51 | Desert |
| 42 | AlShuaibah | 41261 | 55.6 | 24.11 | 209 | Mountain |
| 43 | Arylah | 41262 | 54.2 | 24.99 | 0 | Marine |
| 44 | Ashaab | 41264 | 54.8 | 24.39 | 58 | Desert |
| 45 | JabalYanas | 41266 | 56.1 | 25.73 | 684 | Mountain |
| 46 | RasAlkhaimah | 41267 | 55.94 | 25.77 | 7 | Marine |
| 47 | Shoukah | 41269 | 56 | 25.11 | 232 | Mountain |
| 48 | AbuAlBukhoosh | 41274 | 53.146 | 25.495 | 0 | Marine |

100

# Seasonal and diurnal performance of daily forecasts with WRF V3.8.1 over the United Arab Emirates

Oliver Branch[1], Thomas Schwitalla[1], Marouane Temimi[2], Ricardo Fonseca[3], Narendra Nelli[3], Michael Weston[3], Josipa Milovac[4], Volker Wulfmeyer[1]

[1]Institute of Physics and Meteorology, University of Hohenheim, 70593 Stuttgart, Germany

[2] Department of Civil, Environmental, and Ocean Engineering (CEOE), Stevens Institute of Technology, New Jersey, USA

[3] Khalifa University of Science and Technology, Abu Dhabi, United Arab Emirates

[4] Meteorology Group. Instituto de Física de Cantabria, CSIC-University of Cantabria, Santander, Spain

Correspondence to: Oliver Branch (oliver_branch@uni-hohenheim.de)

**Abstract.**

Effective numerical weather forecasting is vital in arid regions like the United Arab Emirates (UAE) where extreme events like heat waves, flash floods, and dust storms are severe. Hence, accurate forecasting of quantities like surface temperatures and humidity is very important. To date, there have been few seasonal-to-annual scale verification studies with WRF at high spatial and temporal resolution.

This study employs a convection-permitting scale (2.7 km grid scale) simulation with WRF-~~NOAHMP~~NOAH-MP, in daily forecast mode, from 01 January ~~01~~ to 30 November ~~30~~ 2015. WRF was verified using measurements of 2 m air temperature (T-2m), 2-m dew point (TD-2m), and 10 m wind speed (UV-10m) from 48 UAE WMO-compliant surface weather stations. Analysis was made of seasonal and diurnal performance within the desert, marine, and mountain regions of the UAE.

Results show that WRF represents temperature (T-2m) quite adequately during the ~~daytime~~day-time with biases ≤+1˚C. There is however a nocturnal cold bias (-1 to -4˚C), which increases during hotter months in the desert and mountain regions. The marine region has the lowest T-2m biases (≤-0.75˚C). WRF performs well regarding TD-2m, with mean biases mostly ≤1˚C. TD-2m over the marine region is overestimated though (0.75-1 ˚C), and nocturnal mountain TD-2m is underestimated (~-2˚C). UV-10m performance on land still needs improvement, and biases can occasionally be large (1-2 m s$^{-1}$). This performance tends to worsen during the hot months, particularly inland with peak biases reaching ~3 m s$^{-1}$. UV-10m are better simulated in

25  the marine region (bias ≤1 m s⁻¹). There is an apparent relationship between T-2m bias and UV-10m bias, which may indicate issues in simulation of the ~~daytime~~day-time sea breeze. TD-2m biases tend to be more independent.

Studies such as these are vital for accurate assessment of WRF nowcasting performance and to identify model deficiencies. By combining sensitivity tests, process and observational studies with seasonal verification, we can further improve forecasting systems for the UAE.

## 1 Introduction

In a changing climate, effective numerical weather forecasting is vital in arid regions like the United Arab Emirates (UAE), to predict low-visibility events like fog and dust (e.g., Aldababseh and Temimi, 2017; Chaouch et al., 2017; Karagulian et al., 2019), and extreme events relating to storms and flash floods (Chowdhury et al., 2016; Wehbe et al., 2019), high temperatures, and droughts. These extreme events are expected to become more prevalent under a changing climate (Feng et al., 2014; Zhao

et al., 2020). In fact, climate projections suggest that arid and semi-arid regions are likely to expand in area along with rising temperatures (Huang et al., 2017; Lelieveld et al., 2016; Lu et al., 2007). Hence, it is vital that regional weather forecasting and climate simulations with regional climate models (RCMs) correctly simulate important quantities which characterize extreme events, especially surface temperatures, humidity, winds, and precipitation.

The model chain and configuration used in any simulation can heavily influence the results of such forecasts. Important factors

include, but are not limited to, RCM type (e.g., Coppola et al., 2020), general circulation model ~~dataset~~ (GCM) dataset for boundary forcing (Gutowski et al., 2016; Jacob et al., 2020), horizontal and vertical grid resolutions (e.g., Schwitalla et al., 2017b), physics and dynamics schemes (~~e.g.~~e.g., Chaouch et al., 2017; Schwitalla et al., 2020), soil/land use/terrain static data, as well as internal model parameter sets for important land surface processes (~~e.g.~~e.g., Weston et al., 2018).

The Weather Research and Forecasting (WRF) model (Powers et al., 2017; Skamarock et al., 2008) has been used in arid

regions for various forecasting and verification (e.g Branch et al., 2014; Fonseca et al., 2020; Schwitalla et al., 2020; Valappil et al., 2019; Wehbe et al., 2019), and process studies (Becker et al., 2013; Branch and Wulfmeyer, 2019; Karagulian et al., 2019; Nelli et al., 2020a; Wulfmeyer et al., 2014). Currently, there have been few annual-scale verification studies employing the WRF model on a NWP daily forecasting mode at such high spatiotemporal resolution (e.g. $dx < 2 - 3\ km$). Horizontal grid scale in particular, is significant because simulations employing convection-permitting (CP) grid spacing ($dx \sim < 4\ km$) are

known to outperform those at coarser resolutions, particularly in terms of clouds and precipitation - not least because they don't require a convection parameterization (Bauer et al., 2015, 2011; Prein et al., 2015; Schwitalla et al., 2011, 2017a; Sørland et al., 2018). Furthermore, it is known that land use, soil texture, and terrain interact with planetary boundary layer (PBL) processes in complex feedbacks (~~e.g.~~e.g., Anthes, 1984; Mahmood et al., 2014; Pielkel and Avissar, 1990; Smith et al., 2014)

with a strong level of land-atmosphere (LA) coupling thought to exist in this region (Koster et al., 2006). Representation of landscape structure and the associated LA feedbacks should therefore be significantly improved when using finer grid resolution. In terms of time scale, seasonal-to-annual simulations are costly, but provide a sufficient time series for robust statistical comparison with observations over different seasons.

This study employs a ~~verified~~ configuration of WRF, coupled with the NOAH-MP 'multi physics' land surface model (LSM), with modular parameterization options (Niu et al., 2011). In contrast to typical climate mode simulations, WRF is run here in a numerical weather prediction (NWP), or daily forecasting mode in order to keep conditions inside the domain closer to that of the forcing data (see Section 2.3.3 for further details). We also apply high quality/resolution boundary forcing data, improved static data for land use/soils and terrain, ~~and~~ high frequency aerosol optical depth, and sea surface temperature data. This WRF configuration was employed and verified by Schwitalla et al., (2020) within a one-day case study of a physics ensemble.

Our main objective is to assess the seasonal and diurnal performance of WRF – both qualitatively and quantitatively – in reproducing surface air temperature, dew point and wind data from 48 WMO-compliant surface weather stations distributed over the UAE.

Another objective is to assess the model performance in different areas of the UAE – which was split broadly into three environments: 1. northern coastline and islands, 2. inland lowland desert areas, and 3. the Al Hajar mountains in the east. The aim is to investigate differences in performance due to expected differences in climate regimes within these zones, and their respective surface/landscape characteristics and how they are dealt with by WRF-NOAH-MP. Factors include, amongst others, the influence of sea surface temperatures in the warm and shallow Arabian Gulf (Al Azhar et al., 2016), representation of albedo (Fonseca et al., 2020) and roughness length parameters (Weston et al., 2019), and limitations in simulations over orography, particularly with respect to the wind field (~~e.g.~~e.g., Warrach-Sagi et al., 2013). The Al Hajar Mountains have a complex climate with regular coastal fog and convective events (~~e.g.~~e.g., Branch et al., 2020). Therefore, splitting verification into the above zones (in which the stations are quite evenly distributed, with 16, 14, and 18 stations, respectively) can yield further insights into model performance, and climate characteristics in different environments.

Through ambitious simulations and robust verification, we can gain valuable insights into the regional climate, model performance and take a step towards more skilful weather forecasting with WRF-NOAH-MP in the UAE.

The structure of this work is as follows: We start with our Materials and Methods (Section 2), showing maps of the study area and model domain (2.1), a description of the regional climate (2.2), the model chain, configuration and simulation method (2.3), verification data set (2.4), and verification methods (2.5). Then follows a results and discussion section (3), and finally a summary and outlook (4).

## 2 Materials and Methods

### 2.1 Study area and model domain

85    The region under investigation is the United Arab Emirates (UAE) located between 22.61−26.43˚-N and 51.54−56.55˚-E in the far ~~north east~~northeast of the Arabian Peninsula (see Figure 1a), with the 48 surface verification stations being spread out across the country. The model domain is shown in Figure 1b and covers a much larger area, a) to be sure of excluding the area with the strong effects of the boundary forcing (~~i.e.~~i.e., relaxation zone) from the analysis, and b) to incorporate the large scale synoptic weather situation. The model uses a regular latitude-longitude grid and has corner grid cells ~~are~~located at 14.775-˚

90    N, 32.225˚-N, 43.275-˚-E, and 65.725˚-E

### 2.2 Regional climate

#### 2.2.1 Synoptic climate

Weather in the wider region is controlled generally by four predominant patterns~~weather systems~~, including troughs originating from the Atlantic and Mediterranean Sea in winter, locally forced convective storms over the UAE/Oman Al Hajar Mountains

in summer, and the southerly summer monsoon and cyclones from the Arabian Sea during June and October (Bruintjes and Yates, 2003; Steinhoff et al., 2018). These phenomena are represented in large-scale seasonal climatologies (1979 – 2014 - 8:00 UTC) in Figures 2 and 3 (right-hand panels). To represent the climate, we have used geopotential height at 500 hPa, wind velocity at 850 hPa and mean sea level pressure. Note that winter is represented exclusively by the months of January and February, because these are the months used for our winter analysis during 2015 – for reasons of temporal continuity. In the

climatology, we can clearly see a typical winter January-February (JF) heat-low centred over Turkey and Iraq and a trough extending down toward the Arabian Peninsula. During summer June, July and August (JJA), we observe much higher temperatures further south, with a heat-low centred over Iran and the UAE. The other two seasons appear are transitional periods.

#### 2.2.2 UAE climate

The UAE climate is generally characterized by scarce precipitation and high temperatures. However, annual cycles do exist with maxima of precipitation and minima of temperatures in winter and the converse in summer. Annual UAE precipitation is between 20 mm in the drier west to 130 mm in the higher Al Hajar Mountains of the east, mainly produced in the winter-spring time period (Sherif et al., 2014). During summer, subtropical subsidence leads to a strong reduction of precipitation and higher temperatures, and consequently summer precipitation represents only around 20% of the annual amounts. However, upper

level disturbances from the southern monsoon flows can still transport moisture towards the Arabian Peninsula and the UAE

(Böer, 1997; Schwitalla et al., 2020), and convection is initiated sporadically over the mountains of Oman and the UAE in summertime (Branch et al., 2020).

The neighbouring Arabian Gulf to the north of the UAE also plays a strong role in regional weather conditions. The prevailing winds from the Arabian Gulf are westerly or northwesterly between January and May, but these change to north-westerly and then northerly directions from June toward November. ~~The sea surface of~~ In the Arabian Gulf, which is relatively shallow (maximum depth ~90m), particularly close to the UAE coast, the sea surface can heat rapidly, with temperatures often exceeding 30˚C (Al Azhar et al., 2016). Prevailing winds are augmented by strong sea/land breezes, which develop due to land/sea temperature gradients. Daytime sea breezes can penetrate up to 50 km inland (Eager et al., 2008).

## 2.3 Model chain and simulation method

### 2.3.1 Model chain and physics

The model chain is based on the Weather Research and Forecasting model (WRF, Powers et al., 2017; Skamarock et al., 2008) version 3.8.1 using the Advanced Research WRF (ARW) core, which solves the Euler equations on a discretized horizontal grid, with a terrain-following vertical coordinate system. The domain size and grid spacing matches that of a previous simulation by Schwitalla et al., (2020)), ~~was selected (following a twin experiment by Schwitalla et al., (2020))~~ and is comprised of a regular latitude-longitude grid with 900 ~~(x)~~ by 700 ~~(y)~~ ~~grid~~ cells, horizontally (see Figure 1b). In line with our previous statements on CP scale we selected a grid increment of 0.025˚ ($dx \sim 2779$ m), with no parameterization of deep convection. It was important to extend the domain enough to incorporate influential synoptic conditions upstream to the north, east, and south east. Hence, our grid covers a region of approximately $2500 \times 1945$ km extending up to Iraq in the north, down to the south of Yemen, and well into Pakistan in the east. Care was taken, for reasons of model stability, that domain boundaries did not bisect very large peaks, especially in the complex terrain of Iran. Vertically, 100 levels were used, adjusted so that at least 25 levels were present in the lower 2000 m –to maximise resolution of the strong moisture gradients in the boundary layer and lower troposphere.

WRF was coupled with the NOAH-MP LSM (Niu, 2011) to simulate land-surface processes and land-atmosphere feedbacks. NOAH-MP provides a separate vegetation canopy defined by a canopy top and ground layer including a modified energy balance closure approach. It offers a tile approach where the net longwave radiation and turbulent fluxes are calculated separately for bare soil and the canopy layer. The calculated fluxes over vegetated grid cells are then bulked as a weighted sum of bare soil and canopy fluxes. Furthermore, NOAH-MP is partially modular in structure, providing a suite of optional schemes

for several processes, such as radiation budget calculation, stomatal resistance, snow albedo, and others. The same
configuration of Milovac et al., (2016) was used for all NOAH-MP options.

Other physics schemes included were RRTMG for long and shortwave radiation transfer (Iacono et al., 2008; Mlawer et al., 1997), the Thompson-Eidhammer microphysics scheme (Thompson and Eidhammer, 2014) (although without the aerosol-aware component activated), the MYNN scheme for the atmospheric surface layer, and the MYNN 2.5 level TKE scheme for the boundary layer (Nakanishi and Niino, 2006) (See Table 1 for a synopsis of physics schemes and their associated references).

**2.3.2 Initialization and forcing data**

### 2.3.2.1 Initial and lateral boundary conditions

These were retrieved from the European Centre for Medium-Range Weather Forecasts (ECMWF) Integrated Forecasting System (IFS), in the form of 6-hourly operational analysis data on the 41r1 cycle, on model levels. The horizontal grid increment is 0.125° (~12 km) with 137 vertical levels up to 0.01 hPa. Soil moisture and soil temperatures are also provided by
this model, which assimilates satellite soil moisture data (Albergel et al., 2012) into its coupled Hydrology-Tiled ECMWF Scheme for Surface Exchange over Land (HTESSEL) model, HTESSEL (Balsamo et al., 2009).

### 2.3.2.2 Sea surface temperatures (SSTs)

These data were retrieved from the OSTIA project (Donlon et al., 2012) – the data has a 1/20° horizontal resolution at a 12-hourly frequency at 00:00 and 12:00 UTC. This data is particularly important in coastal regions like the UAE.

### 2.3.2.3 Aerosol optical depth (AOD) data

These data were retrieved from the ECMWF Monitoring Atmospheric Composition and Climate (MACC) reanalysis (Inness et al., 2013) which interacts with the shortwave radiation scheme to modify radiative transfer and diabatic heating - data has a ~80-km horizontal resolution and a 6-hourly frequency starting from 00:00 UTC.

### 2.3.2.4 Soil texture data

These data are an update from the default Food and Agriculture Organization (FAO) dataset. The new data are based on the Harmonized World Soil Database (HWSD) v 1.2 at 30 arc second resolution, where all the mapping units are reclassified into 12 soil and 4 non-soil types following the United States Department of Agriculture (USDA) soil classification system, as in the WRF model. For access to the data and more details see Milovac et al., (2018). The WRF default soil texture map based on the FAO data was used for the bottom soil layer.

### 2.3.2.5 Land use data

These data were provided as a combination of a high-resolution dataset for the Emirates of Abu Dhabi and Dubai, provided by the National Center for Meteorology (NCM), and the International Geosphere-Biosphere Programme (IGBP) Moderate Resolution Infrared Spectroradiometer (MODIS) 20-class land use dataset, included within the WRF package (Figure 4). The Abu Dhabi dataset contained some classes which differed from MODIS IGBP, and these were first reclassified in a logical manner before overwriting the MODIS dataset within the UAE (see Schwitalla et al., (2020) for further details of this process).

### 2.3.2.6 Terrain data

Here, we used the Global Multi-resolution Terrain Elevation Data (GMTED) 2010 static dataset ((Danielson and Gesch, 2011))

### 2.3.3 Simulation method

The objective of this study was to run a series of daily forecasts with WRF for the period 01 January to 30 November 2015, with a discarded one-month spin up run from 01 December 2014. Note that December 2014 was not used for verification (observation data was in any case not available at that time. See Section 2.4). It also makes sense not to analyze a winter season split over two years.

The intention of carrying out such a long sequence was to produce a long enough dataset to provide sufficient data points for robust statistical analysis. Forecasts were carried out in a NWP mode, i.e.i.e., with daily cold starts - as opposed to a 'climate' mode, which has a single cold start at the outset. In NWP mode, a cold start was initiated each day at 18:00 UTC (22:00 LT) and run for 30 hours, i.e.i.e., 6+24 until 00:00 UTC the next day. The first 6 hours of each forecast (18:00 UTC to 00:00 UTC) were then discarded from the analysis. The 6 hours allows time for the atmosphere to spin up after each cold start – in particular for the residual boundary layer to develop and dissipate before the convective boundary layer starts to develop after sunrise (~06:00 LT), and for potential cloud development. Other UAE forecasting studies have also suggested 5-6 hours an appropriate period for model convergence in the UAE region (Chaouch et al., 2017; Weston et al., 2018). After discarding the first 6 hours, a forecast remains for analysis spanning the 24 hours of each day between 00:00 and 23:00 UTC (04:00 to 02:00 LT). See Table 2 for a summary of the simulation method.

By reinitialising the 3D state within the domain itself (as opposed to simply inputting lateral boundary conditions), we ensure the atmospheric state is closer to the forecast provided by ECMWF, than would be the case in e.g., typical climate mode simulations. By reinitialising the 3D state inside the domain, we keep the atmospheric simulation closer to the forecast provided

by ECMWF than would be the case in climate mode. In climate mode, which is driven only at the boundaries, the WRF simulations may diverge more strongly, particularly toward the centre of the large domain where the study area lies, unless some form of interior nudging were implemented (e.g.e.g., Lo et al., 2008).

An exception to the daily reinitialization of state variables was made with the soil moisture field, whose state was intentionally maintained from one successive day to the next, rather than being overwrittenby overwriting the soil moisture state from 18:00 to the next day at 18:00, when the forecast is restarted. The intention is to reduce physical inconsistencies between the soil moisture forecast in the driving GCM model and that of WRF-NOAH-MP. Intuitively that may not seem a large issue given the aridity of the UAE. However, it becomes significant when convective precipitation occurs in WRF, and soils are wetted. Such convective events and flash floods are common in the UAE and Oman, particularly from May to September in the mountains, —including during 2015 (Branch et al., 2020; Schwitalla et al., 2020; Wehbe et al., 2019). Hence, the NWP method is a worthwhile method of improving physical consistency. To summarize the NWP configuration: The soil moisture is overwritten at 18:00 from each consecutive day to the next, for the start of each new forecast. The lateral boundary conditions are as for a climate mode run, i.e., input every 6 hours from the forcing data. The atmospheric state within the domain boundaries is reinitialized each day at 18:00.

## 2.4 Datasets for verification

Hourly verification data comes from 48 surface weather stations throughout the UAE (Figure 1a and Appendix Table A1) - quality checked and made available by the National Center for Meteorology (NCM) in Abu Dhabi, UAE. Fields available include air temperature at 2m (T-2m), dewpointdew point at 2m (TD-2m) representing humidity, and wind speed at 10m (UV-10m). Data covers the entire period of January 01-November 30 2015. Unfortunately, quality checked observation data for December 2014 was not available and so in the interest of preserving contiguous seasons, the month of December 2015 was omitted from the winter statistics.

## 2.5 Verification method

An aim of the study is to assess WRF's performance on several timescales: year annually (January-November), seasonally, daytimeday-time and nighttimenight-time periods, and hourly. Another aim is to assess performance within different regions of the UAE. The exclusive assessment of overall forecast means over the UAE may be valuable, but could obscure variability within the different regions, such as the capturing of high daytimeday-time temperatures in the inland deserts, or cooler and windier coastal conditions.

Accordingly, the dataset was split temporally and spatially, as follows.

### 2.5.1 Temporal analysis

#### 2.5.1.1 Yearly analysis

Here, all ~~timesteps~~time steps were analysed from 01 Jan~~uary 01~~ to 30 Nov~~ember 30~~ (hourly interval).

#### 2.5.1.2 Seasonal analysis

Here, we present the most extreme seasons in terms of air temperatures - the (coolest) winter period of January 01-February 28 2015 and the (warmest) summer period of 01 June to 31 August~~31,~~ 2015.

#### 2.5.1.3 Daytime and ~~nighttime~~night-time periods

For daylight hours we used all hours between 02:00 and 13:00 UTC (06:00-17:00 LT) - and for ~~nighttime~~night-time, 14:00 to 01:00 UTC (18:00-05:00 LT). These hours were selected based on the range of UAE sunrise and sunset which range between ~05:30 and 07:00 LT, and ~17:00 and 18:50 respectively. The intention of separating day and night hours in this way is to examine performance during the nocturnal stable and ~~daytime~~day-time convective boundary layers. Indeed, several simulations in arid regions have demonstrated nocturnal cold biases and an overestimation of ~~daytime~~day-time wind speeds (Branch et al., 2014; Schwitalla et al., 2020; Weston et al., 2018).

#### 2.5.1.4 Regional analysis

We split the 48 UAE weather stations into 3 regions – marine, mountain, and desert – based upon on surface geophysical characteristics and proximity to water bodies (See Figure 1a). Accordingly, the following criteria were used for grouping the weather stations into regions:

- **Marine** – located on islands or ≤ 5 km inland from the UAE coast~~—.~~ 17 stations.
- **Mountain** – located in the Al Hajar Mountain area and ≥ 200 ~~m.a.s.l~~m ASL~~—.~~ 16 stations
- **Desert** – located > 5 km distance inland and < 200 ~~m.a.s.l~~m ASL~~—.~~ 15 stations.

The only exception made to this classification was for a single station located at 204 m near the sand dunes of Liwa, in the south of the Abu Dhabi emirate. Although the station is quite high, it is remote from the Al Hajar Range and was deemed more suitable for a desert classification. Details on altitude of the regional station groups can be found in Table 3, and a list of individual stations in the Appendix. The desert region is characterised by barren or sparsely vegetated soils (as is most of the UAE), high surface temperatures, and rapid ~~nighttime~~night-time cooling due to radiative losses associated with a dry

atmosphere. The Al Hajar mountain region is arid, has generally rocky bare slopes, ~~with~~ and lower albedo (~~e.g.~~e.g., Moody et al., 2005)~~.~~ with gravel plains running along the west side (Sherif et al., 2014).

One can assume some similarity between these regions, particularly when the synoptic situation is relatively homogeneous over scales larger than the study area. Nevertheless, given the large number of stations and length of time series, if regional differences do exist then they should be evident.

### 2.5.2 Verification and Diagnostics

All comparisons were made using NCAR's Model Evaluation Tools V9.0 (MET) package, utilizing a nearest-grid cell approach on an hourly temporal resolution.

~~In order to get~~To obtain a visual overview of model performance, in terms of closeness of fit, spread of forecast errors, and distribution of residuals, scatterplots divided by region and day/night period are shown in Figure 5. Included are a line of best fit for the data, a 1:1 line of perfect fit, and a 95% confidence ellipse. Then, we plotted regional seasonal statistics of the mean observations (T-2m, TD-2m, and UV-10m) (Figure 6).

To quantify the regional forecast/observation association, error magnitude, and sign during day/night, we show three standard statistical diagnostics (Pearson correlation coefficient, root mean square error (RMSE), and bias).

The **Pearson correlation coefficient 'r'** measures the strength of linear association between forecast ($f$) and observation ($o$), at all stations at each ~~timestep~~time step, given as:

$$r = \frac{\sum_{i=1}^{ns} (f_i - \overline{f})(o_i - \overline{o})}{\sqrt{\sum_{i=1}^{ns} (f_i - \overline{f})^2} \sqrt{\sum_{i=1}^{ns} (o_i - \overline{o})^2}}$$

where $f_i$ and $o_i$ are the forecast and observation at each observation point $i$, $\overline{f}$ and $\overline{o}$ are forecast and observation averages, $ns$ indicates the total number of observations at each time step (~~i.e.~~i.e., number of stations), and overbars indicate the mean. Occasionally $ns$ was reduced slightly whenever a missing value occurred.

The **root mean square error (RMSE)** is a scale-dependent diagnostic defined simply as the square root of the mean square error (MSE) of the forecast:

$$RMSE = \sqrt{MSE} = \sqrt{\frac{1}{ns} \sum_{i=1}^{ns} (f_i - o_i)^2}$$

The **Bias** is a measure of overall error, including sign, defined as:

$$Bias = \frac{1}{ns} \sum_{i=1}^{ns} (f_i - o_i) = (\bar{f} - \bar{o})$$

These diagnostics were generated for 2015 for the region and time period and their temporal distribution expressed in boxplots (Section 3, Figure 7) showing mean, median, 25%-75% percentiles (box range), and 5% and 95% percentiles (whiskers).

Finally, a closer look at the diurnal evolution of the forecast is useful to investigate performance at specific times of day such
as local noon and at PBL transition periods, where models often have biases. Hence, we generated mean hourly cycles of the spatial mean and spatial standard deviations for both forecast and observations. The mean at each hour is calculated as:

$$Mean(h) = \frac{1}{T} \sum_{t=1}^{T} \frac{1}{ns} \sum_{i=1}^{ns} o_i \ or \ f_i$$

The spatial standard deviation ($\sigma$) at each hour is given as:

$$\sigma(h) = \frac{1}{T} \sum_{t=1}^{T} \sqrt{\frac{1}{ns-1} \sum_{i=1}^{ns} (o_i - \bar{o})^2} \qquad or \ f_i - \bar{f}$$

For the diurnal analysis, we selected the two most extreme seasons in terms of temperature - the (coolest) winter period of January-February (Figure 8) and the (warmest) summer period of June-August (Figure 9), 2015. Again, these figures are divided by region.

**3 Results and Discussion**

In this section, we present a discussion of the results. Before examining the model performance however, we first discuss the
study period of 2015 in context of the long term climate and El Niño (3.1) to assess the representativeness of the 2015 study

period. We then discuss differences in regional climate and their significance to our verification (3.2). Finally, we evaluate the regional model output of T-2m, TD-2m and UV-10m fields across the seasons and time of day (3.3).

## 3.1 2015 in context

Our study period is 2015 from 01 January to 30 November (during which time the full verification dataset was available). 2015 was considered one of the strongest El Niño periods since 1950 (L'Heureux et al., 2017) with an Oceanic Niño Index (ONI) index of up to 2.6 towards the end of the year (see Table 4). A high positive ONI indicates a stronger El Niño event (negative indicate La Niña events). El Niño Southern Oscillation (ENSO) is known to impact upon the climate in this region, including temperatures and precipitation in the UAE (AlEbri et al., 2016; Almazroui, 2012; Chandran et al., 2016) so one might expect significant climate anomalies during 2015. Hence, a comparison was made between the long term climatology and the year 2015, based on ECMWF ERA5 reanalysis data. In Figures 2 and 3, from the geopotential height field, we can see that a positive 2015 winter temperature anomaly exists to the north of the UAE, extending from Turkey to the Caspian Sea (Figure 2, top left). However, conditions over the UAE show less deviation in terms of the temperature, pressure and wind fields. As the year progresses, and the ONI increases, the temperature anomaly becomes more pronounced further south, especially in JJA when higher 2015 temperatures extend further south toward Oman and Yemen than apparent in the climatology (Figure 3, top panels). Overall though, synoptic conditions over the Arabian Peninsula don't appear to be markedly different. They are similar enough in fact, to consider the 2015 regional climate as representative of the climate in general.

## 3.2 Regional and seasonal characteristics

An assessment of regional distributions reveals that clear differences in means and variability do exist (Figure 6). As expected, the marine region is dominated by the Arabian Gulf characteristics, with more moderate temperature maxima and minima (Figure 6a), greater humidity (Figure 6b), and higher wind speeds (Figure 6c) than the inland desert for instance (Figure 6). Hence marine temperatures are lower than at the desert stations in the summer months but remain higher in winter and autumn. In fact, the desert stations have the most extreme T-2m range in all seasons, reflecting the lower heat capacity surface, and consequent strong daytimeday-time surface heating. Rapid nocturnal cooling also occurs due to radiative losses in a much drier inland environment. The mountain region is only a little cooler than the desert (~1˚C) in summer and autumn with the difference further reduced during spring and winter. The majority of mountain stations are located at fairly moderate altitudes (mean altitude 430 m, Table 3) with only one station located over 1000 m high (station ID 41229 - 1485 m.a.s.lm ASL, see Table A1 in Appendix). Even so, one might have expected larger differences. However, there could be reasons other than the temperature lapse rate for this, such as differences in mountain and desert cloud cover for instance (Branch et al., 2020; Yousef et al., 2019), or in albedo (e.g Nelli et al., 2020b).

TD-2m, or ~~dewpoint~~dew point temperature, is a standard measure of humidity and is in most cases relatively independent of the ambient temperature. It is also a reliable measure of how humid the air feels in terms of human comfort (Wood, 1970). In a hot (and warming) climate like the UAE, forecasting TD-2m accurately is therefore ~~very~~ important for society. Regionally, we observe considerable differences in TD-2m (Figure 6b), which ~~is~~ are more or less expected due to coastal/land gradients and variation in vertical transport/distribution of vapor in different environments. Table 5 shows the difference in observed T-2m and TD-2m means. The inland atmosphere tends to be humid in summer when temperatures are high, but even closer to saturation in autumn and winter as temperatures fall, but humidity remains high. This seasonal range is particularly pronounced in the mountain regions reflecting the predominance of annual rainfall occurring during winter in the mountains and gravel plains of the north-eastern part of the UAE (Sherif et al., 2014; Wehbe et al., 2019). In winter and spring, the marine region is closer to saturation than in the other regions (T-2m minus TD-2m = -8 to -11˚C); however, a reversal of this relationship occurs in summer and autumn as the mountain and desert regions become more humid.

There are significant regional differences in UV-10m, with marine UV-10m being 0.5-1 m s$^{-1}$ higher than in other regions (Figure 6c) and also more variable. This is not unexpected, due to low surface roughness, strong land-sea temperature gradients, and ~~the~~ associated land-sea breezes. Desert UV-10m is the lowest all year round, and mountain UV-10m falls in between those of the desert and marine regions. In general, UV-10m is highest in spring and autumn.

These regional differences justify the need for regional splitting of the dataset and are further addressed ~~in Section 3~~below, in conjunction with model performance.

### 3.3 Model evaluation

Although the simulation of T-2m, TD-2m and UV-10m and causes for any biases may be physically linked, we nevertheless, first examine each field individually for clarity.

### 3.3.1 T-2m

In the scatter plots (Figure 5a-5h) we observe that in the ~~daytime~~day-time, T-2m appears well estimated for the UAE on the whole (Figure 5a) (+0.44˚C) and errors are well distributed over the T-2m range. However, this agreement obscures some compensating regional biases; namely overestimation in the desert (+0.71˚C) and mountains (+1.06˚C), and underestimation in the marine region (-0.93˚C).

Reasons for the warm bias may be attributable to a combination of reasons. Firstly, a WRF overestimation of downwelling surface shortwave radiation has been observed before (Fonseca et al., 2020; Nelli et al., 2020b). This has been attributed to a

lack of cloud cover, but may also relate to the performance of the radiative transfer scheme and interaction with aerosols.

Secondly, the soil representation, such as soil texture classification – and the-associated parameters like heat capacity, thermal diffusivity, and albedo – may require adjustment. Underestimations of albedo in WRF have recently been observed particularly for bright desert soils where measurements show typical albedo values of 0.3 to 0.34 (Nelli et al., 2020b). The WRF albedo value in this study is around 0.23 for much of the UAE lowlands, which would likely result in a too-high net radiation and sensible heating, especially on dry soils. This is consistent with the reported positive daytimeday-time temperature biases in

the inland desert. A third factor may be the prescribed aerodynamic roughness length parameters used by WRF. Nelli et al., (2020a) found that a new value for the parameter, derived from eddy covariance measurements, reduced the warm daytimeday-time bias in WRF simulations (Nelli et al., 2020b). These causes may account for some or all of the daytimeday-time temperature biases and therefore need to be considered for future simulations in this region.

Nocturnally, we observe a cold bias over the UAE (Figure 5e). This is quantified in Figure 7b as a mean negative bias of just

355 over -2˚C. One can also see that this nocturnal bias tends to worsen with an increase in daily T-2m, which implies that the cold bias gets worse in the hotter months. This is confirmed in the seasonal diurnal cycles (Figure 8a and 9a) where the mean nocturnal bias in winter is ~ -2˚C, but increases to greater than -4 ˚C in summer. This nocturnal cold bias is reflected in all sub-regions, but not to the same degree. The best nocturnal performance is in the marine region (Figure 5g) (bias of -0.75 ˚C), with an even error distribution across the temperature range. The largest nocturnal cold bias is in the desert region (-3.1 ˚C)

(Figure 5h), with a steady increase in bias with temperature. The switch from positive to cold biases usually occurs more or less around the twice-daily transition times of the boundary layer between stable and convective states. Such arid nocturnal biases have been noted before (Branch et al., 2014; Fekih and Mohamed, 2017; Weston et al., 2018). It may be that a too-dry lower atmosphere results in a lower downward flux of longwave, as found by (Fonseca et al., 2020) in a comparison of WRF with radiation measurements. All else being equal this dryness would lead to a reduction of 'buffering' at night timenight-

time. They also found a too-high upward ground heat flux during the night, which could be associated with sub-optimal soil parameters or a too-strong soil-air temperature gradient. Overall, their net radiation losses at night were higher in WRF than from the radiation measurements.

### 3.3.2 TD-2m

TD-2m is relatively well estimated in 2015 for over the UAE as a whole, with correlations around 0.7 and biases of less than 1˚C (Figure 7d and 7e, UAE sections). However, we can look at regional/seasonal differences for more detail. In the desert and marine regions, the biases are ≤1˚C during both day and night. Marine TD-2m is slightly overestimated in general, indicating the model to be more humid over the Gulf and coast than observed. Mountain nocturnal dew points are more of a

problem with a negative bias of ~ -2˚C, and a larger error spread than the other regions (Figure 7e). There is also a corresponding T-2m nocturnal bias of ~ -2˚C which could indicate a deficiency in the longwave surface budget as just mentioned, but also a model deficiency in representing the intermittent shear-driven turbulence that appears in ~~night time~~night-time stable boundary layers. However, such biases in complex terrain have been already well documented (~~e.g.~~e.g., Warrach-Sagi et al., 2013; Zhang et al., 2013). One of the reasons cited is that the CP scale is not fine enough to resolve mountain slopes, and therefore cannot capture certain processes in the same way that large-eddy scale models can, with grid spacings on the order of $\Delta x = 100$m. While such fine resolutions may be appropriate in a research context though, they may remain prohibitively expensive and inappropriate in the context of operational forecasting.

An additional problem in complex terrain is the validity of the traditional Monin-Obukhov similarity theory (MOST) (~~e.g.~~e.g., see Foken, 2006) that is typically used in atmospheric models, including WRF, for calculation of model diagnostics like T-2m or TD-2m. MOST assumes homogeneous underlying land surface and stationary fluxes, and there are multiple evidences that in complex and heterogeneous landscapes MOST needs significant improvements in scaling of turbulent kinetic energy profiles in the lowest part of the boundary layer (~~e.g.~~e.g., Figueroa-Espinoza et al., 2014; Wulfmeyer et al., 2018). The latter may affect representation of the heat, moisture, and momentum transport from the land surface to the atmosphere, and if misrepresented may lead to such high biases in the surface layer model diagnostics.

Seasonally, diurnal TD-2m is quite well reproduced in both winter and summer (Figures 8 and 9). The mountain nocturnal negative bias becomes more significant in summer (Figure 9e). In the desert, a positive bias occurs over midday starting around 10 am LT (Figure 9k) showing an overestimation of water vapor in summer. This is likely to be too early in the day for a sea breeze driven anomaly but may relate to simulated soil moisture being higher than reality. This was observed in a study by Wehbe et al. (2018) that~~who~~ found a wet bias in dry soils and a dry bias in wetter soils in WRF over the UAE when not coupled with a more advanced hydrological model.

### 3.3.3 UV-10m

WRF overestimates UV-10m during the day and night, in all regions and seasons. Positive biases of 1-2 m s$^{-1}$ are typical over the whole year (seen in Figure 7h). Mountain ~~daytime~~day-time biases are strongest at 2 m s$^{-1}$, followed by ~~daytime~~day-time desert biases at 1.5 m s$^{-1}$. Marine biases are lowest with mean biases of <1 m s$^{-1}$. Notably, there is a trend where positive biases increase with wind speed (Figure 5p, 5q, 5s). There is a significant increase in bias during the ~~daytime~~day-time, and also in the summer, particularly in the mountain and desert regions (Figure 9f and 9i). In fact, the strongest wind biases occur in the same situations when ~~daytime~~day-time T-2m is overestimated, particularly in the mountain and desert regions (Figures 7, 8, 9), hinting at a relationship between the two. Indeed, ~~there is a good chance~~it is likely that a too-strong sea breeze may account for this. During summer, the desert-marine T-2m ~~daytime~~day-time gradient is highest (~5 ˚C, see Figure 9g and 9j, red curves)

than in winter (~3 ˚C, see Figure 8g and 8j), although the seasonal warm-biases are similar (~1.5-2 ˚C). The higher gradient coincides with a greater UV-10m bias in summer. Weston et al., (2018) improved the duration and direction of UAE sea breezes by tuning a thermal roughness length parameter in WRF. The PBL and surface layer parameterization schemes could also be a cause of the bias. Schwitalla et al., (2020) found an overestimation of UV-10m in all members of a UAE physics ensemble, with magnitudes of around 1.5 m s$^{-1}$. The bias was worse when using the MYNN 2.5 TKE PBL and MYNN surface layer schemes, when compared with the Yonsei University (YSU) scheme (Hong et al., 2006) paired with the MM5 Jiménez~~Jimenez~~ surface layer scheme (Jiménez et al., 2012).

Using a non-local PBL scheme like YSU tends to produce a deeper and drier PBL with a stronger vertical mixing, in comparison to local schemes like MYNN (see Milovac et al., 2016; Yang et al., 2017). This may lead to a reduction in wind speeds, heat, and moisture close to the surface. However, another study however found that switching between 7 different PBL schemes had little effect on positive UV bias (Shimada et al., 2011). One additional factor is that there are several parameters within the MYNN scheme itself, which may benefit from retuning for arid regions like the UAE (~~e.g.~~e.g., Yang et al., 2017). However, the total impact of the PBL scheme selection on reproduction of the T-2m, TD-2m and UV-10m diagnostics is not completely clear. This is because the method of calculation of transfer coefficients/fluxes are executed in NOAH-MP, the PBL scheme, and the surface layer scheme (SLS) depends on the land surface type. In WRF, PBL schemes are generally coupled to the SLS, and typically all variables between the land surface and lowest model layer are diagnosed (e.g. T-2m, U-10m, V-10m). These calculations in the SLS are based on Monin-Obhukov similarity theory, and are represented in the model as hard-coded parameters and/or formulations of similarity functions. The latter are used to obtain dimensionless bulk transfer coefficients which are used for calculating momentum, heat, and moisture fluxes, and for diagnosing near surface quantities like T-2m. These coefficients re-enter the LSM and are to calculate the surface fluxes which then enter the PBL scheme, as the lower boundary condition. Therefore, bias in near-surface variables is strongly related to the choice of LSM and SLS. In this WRF configuration, the communication link between the SLS and NOAH-MP is broken, as NOAH-MP itself calculates transfer coefficients and diagnostics over land surfaces, effectively bypassing the SLS (Nielson et al, 2013). The SLS only becomes active over water surfaces. This means that when NOAH-MP is used, the LSM probably has a stronger impact on the bias of near surface variables than the PBL and SLS (e.g. Milovac et al. 2016).

Incorrect aerodynamic roughness length parameters, as mentioned previously, may also play a large role in determining UV-10m – this parameter is used within the surface layer scheme. Nelli et al., (2020a) found positive wind speed biases over the same region ~~of~~ when wind speeds were < 4 ~~ms$^{-1}$~~m s$^{-1}$ and negative biases for wind speeds which were > 6 ~~ms$^{-1}$~~m s$^{-1}$ within a WRF V3.8 simulation. We have a similar behaviour at night in the marine and desert regions, as exhibited by the positive-to-

negative distribution of errors increasing with wind speed. Nelli et al., (2020a) reduced these biases by retuning the roughness length parameter based on eddy covariance measurements (Nelli et al., 2020b).

Another possibility is the length of the forecast ~~spinup~~spin-up, the required length of which may still be uncertain. We have already mentioned that Chaouch et al., (2017) cited a 5-h ~~spinup~~spin-up as being sufficient, but Hahmann et al., (2015) posits that the necessary ~~spinup~~spin-up over land could be 12 hours or even more (primarily for effective use of the PBL scheme). However, ~~. A longer daily~~ such long ~~spinup~~spin-ups ~~may be preferable but is~~ are likely to be ~~very~~ (i) prohibitively expensive and ~~perhaps~~ (ii) too time consuming for forecasting purposes.

## 4 Summary and Outlook

The aim of this study was to (i) assess the skill of WRF-~~NOAHMP~~NOAH-MP in reproducing surface quantities over the UAE, (ii) identify regional, seasonal, and diurnal differences in performance and (iii) estimate potential sources of model deficiencies. We have demonstrated the value of splitting the model evaluation temporally and spatially. For while assessment of diagnostics for the whole UAE region remains useful, it can obscure regional, diurnal and seasonal differences and also compensating biases, all of which are scientifically interesting, and importantly may reveal information on model performance ~~in~~ with respect to specific processes and land surface types, and how they are simulated.

An analysis of model predictions has revealed that WRF-~~NOAHMP~~NOAH-MP represents the mean T-2m field reasonably well during the ~~daytime~~day-time ~~—~~. although with a tendency for slight overestimation (≤1˚C). The nocturnal T-2m is underestimated more strongly though (1-4˚C), and with larger biases during the hotter months, particularly in the desert and mountains, likely due to a combination of deficiencies. The marine region has the lowest T-2m biases, which is encouraging, and highlights the value of ingesting quality SST data, especially in coastal regions. WRF shows a good performance regarding TD-2m in general, with mean biases being ≤1˚C. Humidity over the marine region tends to be slightly overestimated though, whilst nocturnal mountain TD-2m is underestimated (bias ~-2˚C). UV-10m performance on land still needs be improved, with biases of 1-2 m s$^{-1}$. Furthermore, performance for UV-10m tends to worsen during the hot months, particularly inland. UV-10m in the marine region is generally much better simulated than in the other regions (bias ≤1 m s$^{-1}$). There is an apparent relationship between T-2m bias and UV-10m bias, and this could be due to deficiencies in sea-land breeze simulation. TD-2m biases appear to be more independent. The only exception to this is during the night, when T-2m and TD-2m biases do appear linked.

Ultimately, no model downscaling forecast (at scales economically viable for forecasting) can be expected to exhibit exceptional skill in all conditions. A caveat generally when evaluating models is that one must factor in a certain level of error in station or gridded observational datasets themselves (e.g., as discussed by Prein and Gobiet, 2017). Nevertheless, assuming

a high level of observational accuracy, we have discussed several avenues for improvement on this application of WRF. For instance, we should continue to devise and ingest new and improved datasets for land usecover, terrain and soil texture, and albedo. In particular, within a vegetation sparse region like the UAE, soil texture, moisture and other parameters are likely to be of prime importance. Certainly, ingesting SST data appears to have been valuable, given the lower coastal biases in all variables.

We have mentioned several very useful experiments carried out on parameters like aerodynamic and thermal roughness lengths (Nelli et al., 2020a; Weston et al., 2018), and also process-based observational studies related to the surface energy balance, and verification studies (Fonseca et al., 2020; Nelli et al., 2020b). Further experiments should now be coordinated in order to improve model predictions further. In terms of parameterization schemes, ensemble experiments (in the manner of Chaouch et al., 2017; Milovac et al., 2016; Schwitalla et al., 2020) are still required to identify optimal land surface/surface layer/PBL/microphysics combinations for arid regions. Such studies can also address the tunabletuneable parameters defined inside parameterization schemes similarly to those conducted by Quan et al. (2016) and Yang et al. (2017). The most relevant ones can then be measured during dedicated field campaigns and subsequently ingested in the model.

Seasonal scale studies such as these are vital for accurate assessment of WRF nowcasting performance and to identify model deficiencies and areas for improvement. By combining seasonal verification with sensitivity tests, and process and observational studies, we will move towards improved forecasting systems for the UAE, and other arid regions.

**Appendix**

**Observation stations**

See Table A1 for details on individual weather stations.

**Code availability**

**WRF** - To download the WRF source code, users need to register on the following website: http://www2.mmm.ucar.edu/wrf/users/download/wrf-regist.php.

The **namelist**.input file which is used for the WRF configuration, and **scripts for running WRF in NWP mode** are uploaded with open access to Zenodo:

DOI: 10.5281/zenodo.3894491

**Model Evaluation Tools V9.0 (MET)** open source - NCAR Research Applications Laboratory – Generation of verification
statistics. Available from: https://ral.ucar.edu/solutions/products/model-evaluation-tools-met

**NCAR Command Language (NCL) V6.2** open source – Graphics, and used for overwriting soil moisture data when running NWP mode.

Available from: https://www.ncl.ucar.edu/

**ArcGIS V10.5** proprietary – Graphics and Mapping

Information: https://www.esri.com/en-us/arcgis/products/arcgis-desktop/overview

**Originlab 2020 V9.7.0.185** (Academic) proprietary – Statistical analysis and Graphics

Available from: https://www.originlab.com/index.aspx?go=Products/Origin

**Data availability**

**WRF output data** - available, on reasonable request as it is extremely large in size (many TB). It is archived on the German Climate Computing Center (Deutsches Klimarechenzentrum, DKRZ) and will be there for a minimum of 10 years.

**Verification data** - uploaded to Zenodo in the form of Excel files – open access. Data is courtesy of NCM, UAE:

Observation data

https://zenodo.org/deposit/3894544

Verification statistics dataset

https://zenodo.org/record/4004195

**Team List**

Oliver Branch[1], Thomas Schwitalla[1], Marouane Temimi[2], Ricardo Fonseca[2], Narendra Nelli[2], Michael Weston[2], Josipa
Milovac[3], Volker Wulfmeyer[1]

[1]Institute of Physics and Meteorology, University of Hohenheim, 70593 Stuttgart, Germany

[2] Khalifa University of Science and Technology, Abu Dhabi, United Arab Emirates

[3] Meteorology Group. Instituto de Física de Cantabria, Santander, Spain

**Author contributions.**

O. Branch is the first author who conceived the experiment, carried out the simulations and analysis, and wrote the publication. T. Schwitalla contributed greatly with scientific support and co-writing of the manuscript, provided much technical assistance, and formatted the observation data for use in the MET software. Marouane Temimi, Ricardo Fonseca, Narendra Nelli, Michael Weston, and Volker Wulfmeyer provided specialist scientific support and assisted with the drafting and improvement of key

aspects of the manuscript.

**Conflicts of interest**

The authors declare that they have no conflict of interest.

**Acknowledgements.**

This material is based on work supported by the UAE Research Program for Rain Enhancement Science, under the National Center of Meteorology, Abu Dhabi, UAE. Furthermore, we are grateful to the High Performance Computing Center Stuttgart (HLRS) for providing support and computing time on the XC40 system. We are also grateful to ECMWF for providing operational analysis data.

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

**Figures**

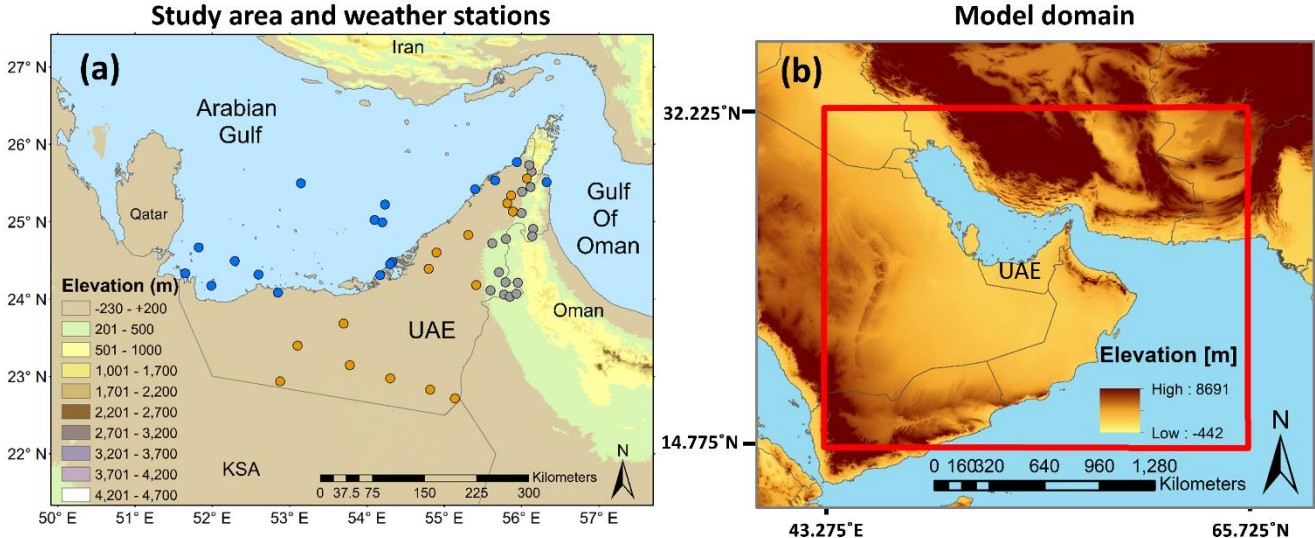

**Figure 1: Panel (a) is a closeup of the study area overlaid with classified topography and 48 UAE surface weather stations used for verification of WRF. Weather data was provided by the National Centre for Meteorology (NCM) in the UAE. The weather stations were grouped into geophysical regions for statistical analysis. The 17 blue dots indicate coastal/marine stations (criteria – on islands or within 5 km from coastline). The 16 grey dots are mountain stations (any station ≥200 m a.s.l. and > 5 km from coast). The 15 orange dots are inland desert stations (criteria –all remaining stations). Panel (b) is the 900(x) × 700(y) grid cell model domain ($\Delta x$ 2.7 km, 2430 × 1890 km). The four corner model grid cells are located at 14.775°-N, 32.225° N, 43.275°-E, and 65.725°-E.**

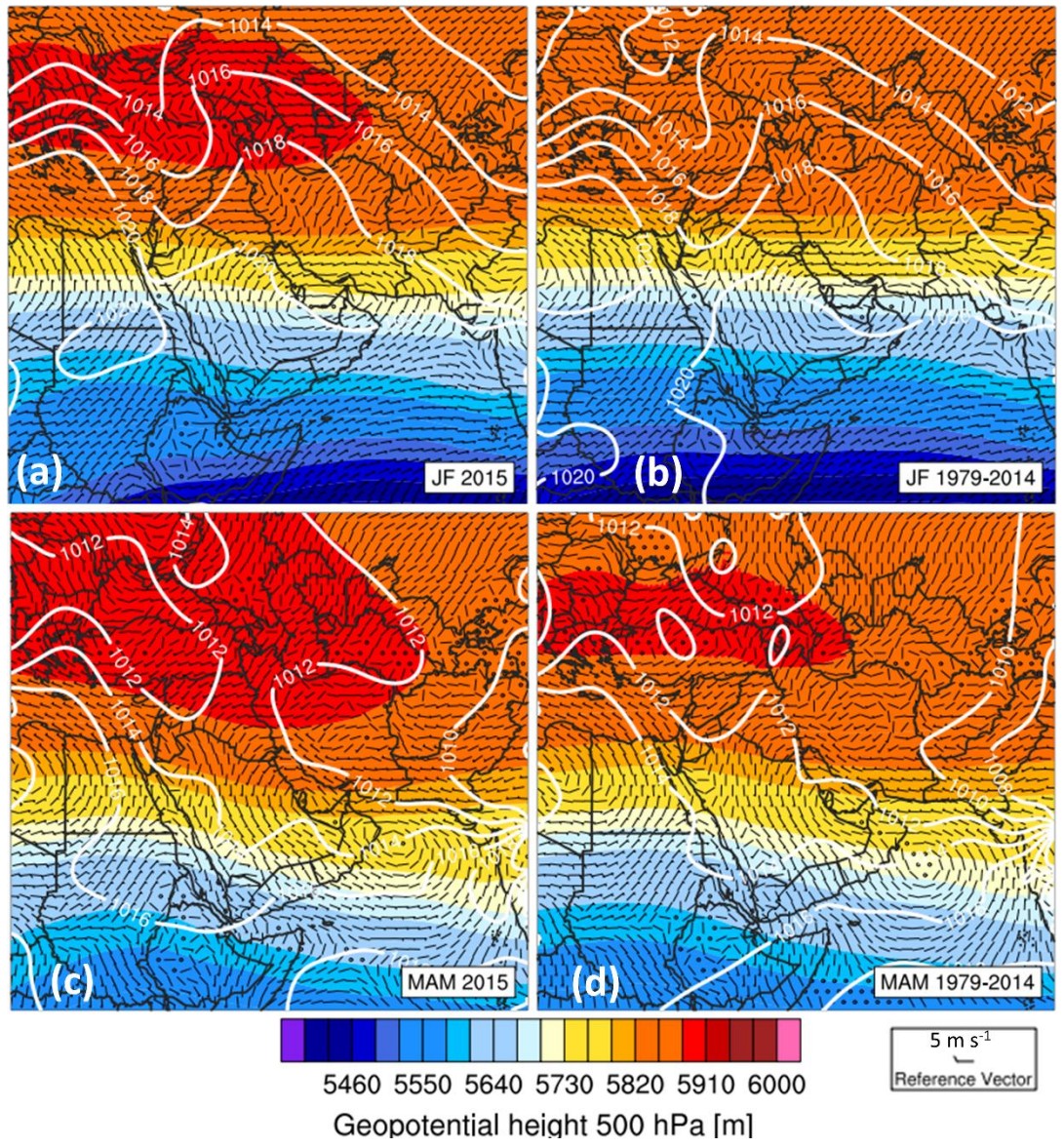

**FIgure 2: Comparison of the 2015 (a) winter (January-February, JF) and (c) spring (March-May, MAM) large-scale fields at 08:00 UTC. (b) and (d) are an equivalent 36 year climatology between 1979 and 2014. Variables shown are geopotential height at 500 hPa [m; shading], wind velocity at 850 hPa [m s-1, see reference vector at bottom right] and mean sea level pressure [hPa; white contours]. Data is taken from the ECMWF ERA5 reanalysis dataset.**

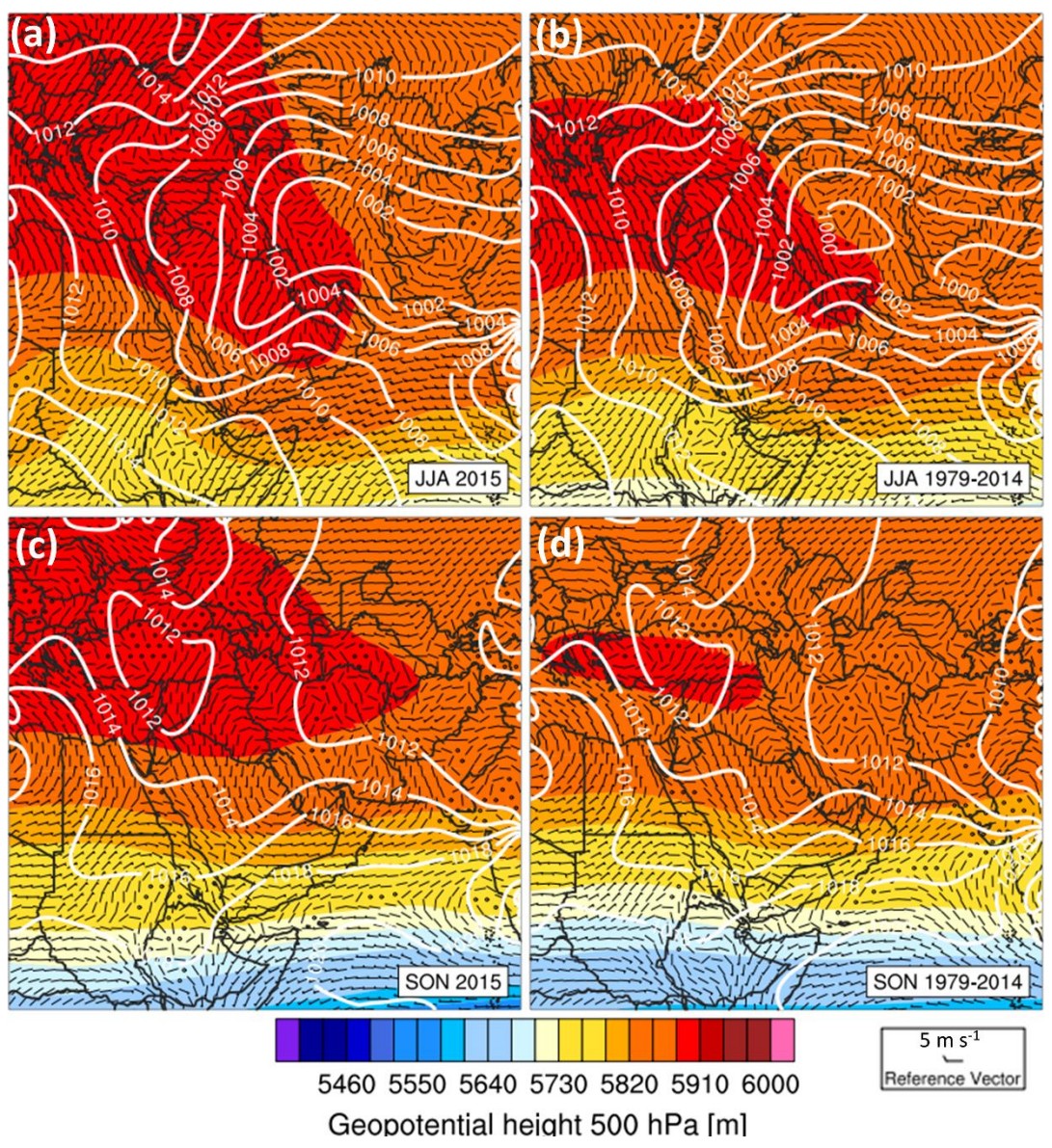

**Figure 3: As for Figure 2 but for summer and autumn (Jun-Aug upper panels and Sep-Nov, lower panels). Data also taken from the ECMWF ERA5 reanalysis dataset.**

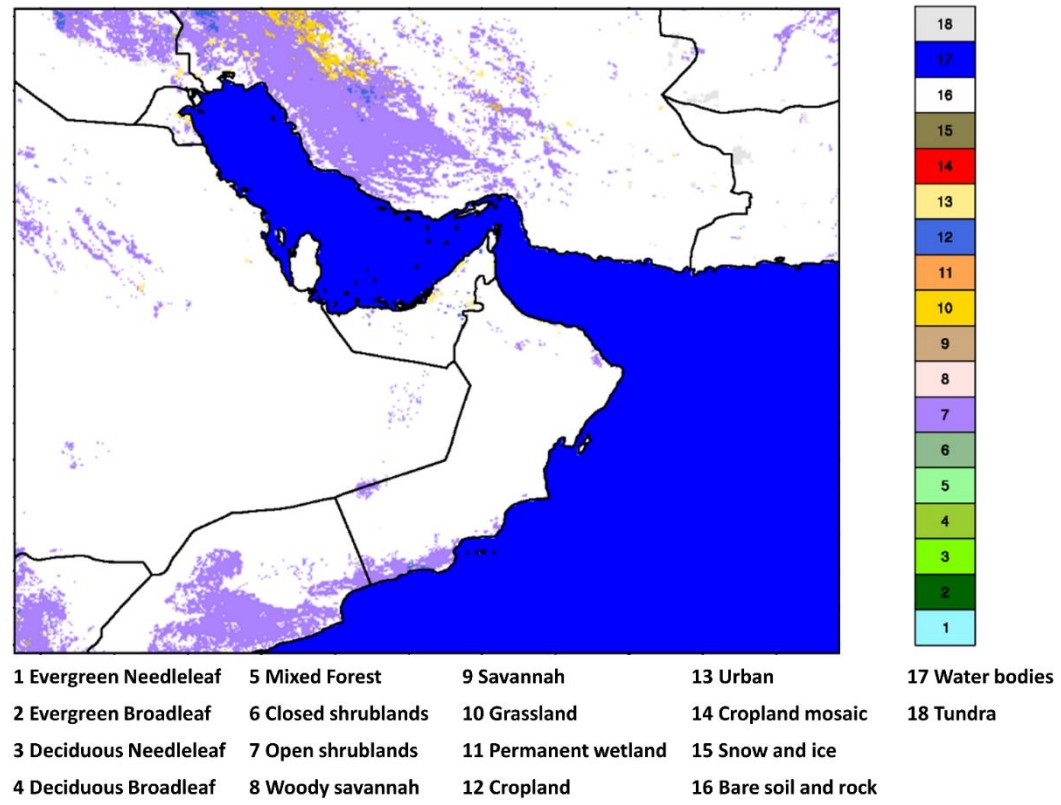

**Figure 4: Map of whole model domain with the land cover data set used in the simulation. It is a composite of the standard 30 arc second (~1 km) IGBP 21 class MODIS dataset included as standard with WRF, with 2 local datasets superimposed: Abu Dhabi and Dubai Emirates, obtained respectively from the Environment Agency of Abu Dhabi (EAD) and the International Center for Biosaline Agriculture (ICBA) in Dubai. The local datasets were first reclassified in a logical manner into MODIS categories. 18**

15 **classes are shown here. There is a reduction in resolution due to the grid increment of 2.7 km.**

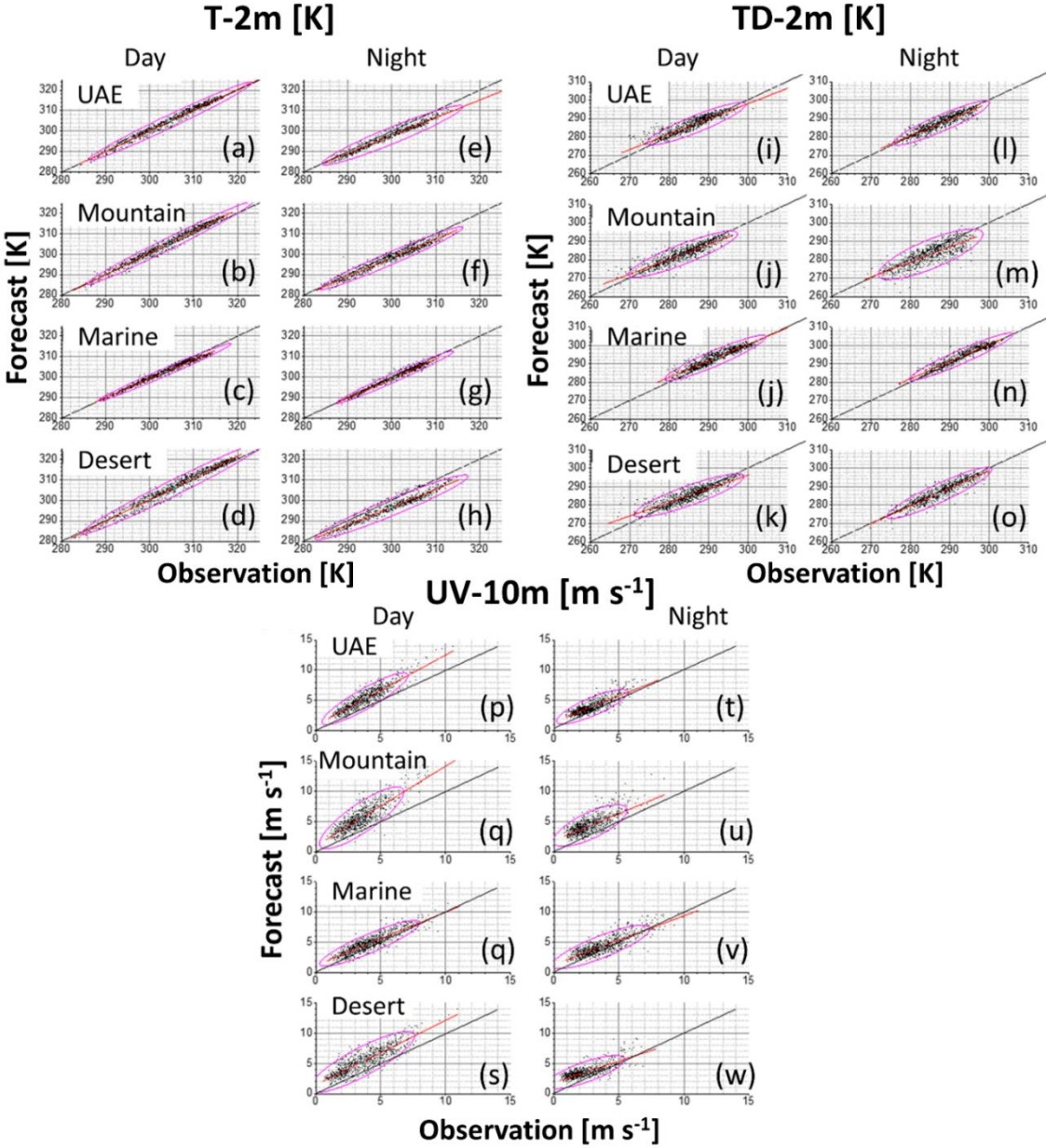

**Figure 5: Scatter plots of observation vs forecast for all time steps over the period of January-November 2015, comparing each weather station at the corresponding WRF grid point. The plots are split by ~~daytime~~day-time (left panels) and ~~nighttime~~night-time periods (right) (respectively, day 06:00-17:00 (left panels) and night 18:00-05:00 (right) in local time), and by region (UAE, Mountain, Marine, Desert). The variables compared are 2--m air temperature (T-2m, K) in panels (a – h) ~~and~~, 2--m dew point (T_D-2m, K) in panels (i – o), and 10--m wind speed (UV-10m, m s[-1]) in panels (p – w). Also shown is a line of best fit (red) and a line of perfect fit (black), and 95% confidence ellipse (magenta).**

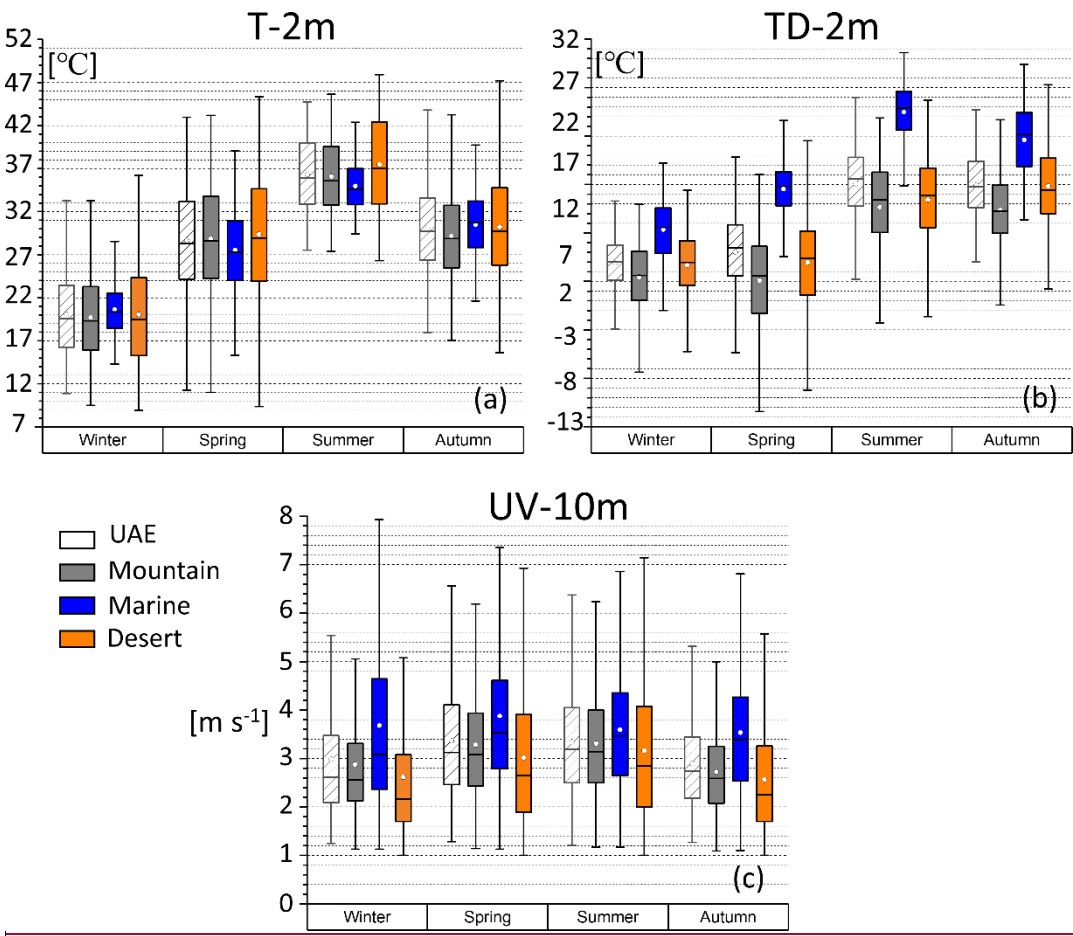

Figure 6: Regional seasonal statistics of mean observations (T-2m (a), TD-2m (b), and UV-10m (c)). Box plots show the mean as a centre line, median as a dot, box ends are 25% and 75% percentiles, and whiskers are 5% and 95% percentiles.

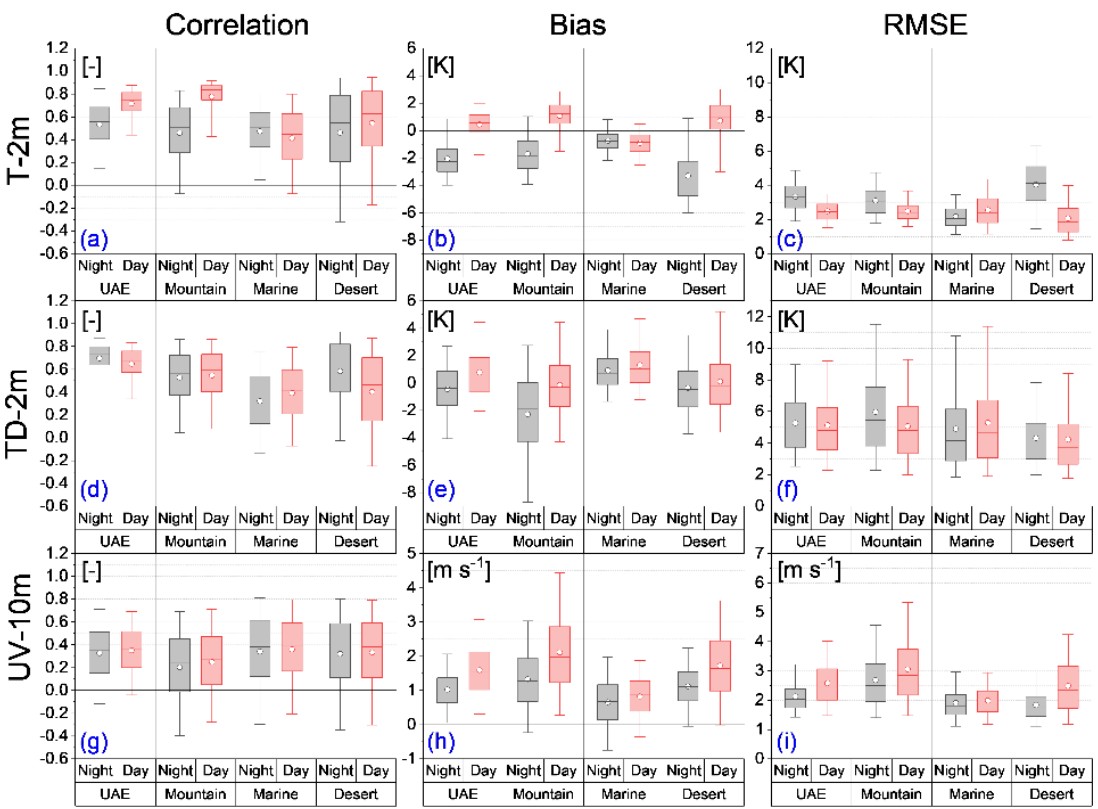

**Figure 7: Box plots of T-2m, TD-2m, and UV-10m (respectively, panels (a-c), (d-f) and (g-i)) for all time steps over the period of January-November 2015. Statistics are divided by region (UAE, Mountain, Marine, Desert) and then by ~~nighttime~~night-time and ~~daytime~~day-time hours (respectively, night 18:00-05:00 (grey boxes) and day 06:00-17:00 (red boxes) in local time). Statistics shown are Pearson correlation (left panels), Bias (centre) and RMSE (right). On the box plots the centre line represents the mean, the white circle is the median, box ends represent 25% and 75% percentiles and the whiskers are 5% and 95 % percentiles. Also marked is a horizontal zero reference line for the Pearson and Bias statistics.**

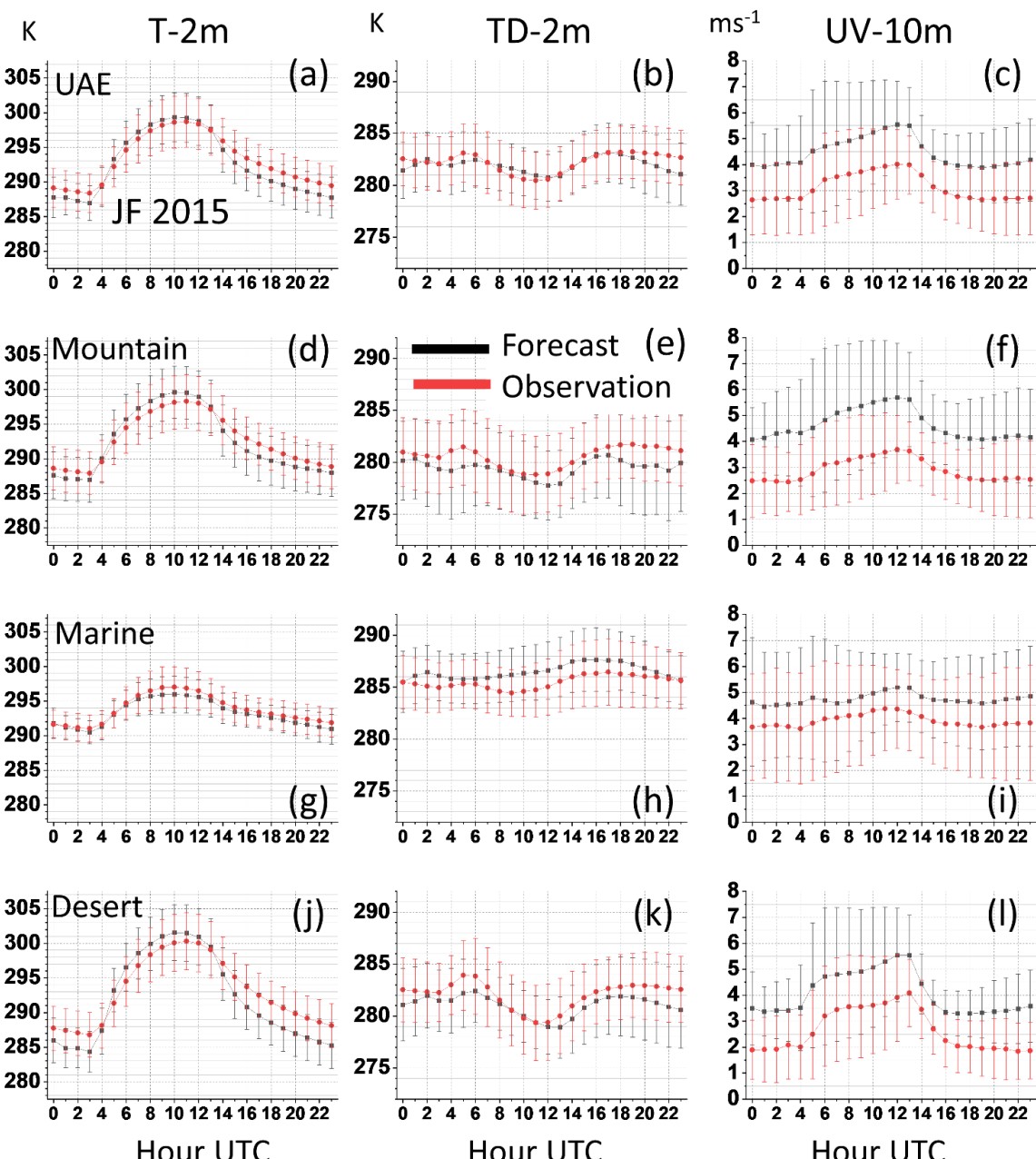

**Figure 8: Winter diurnal cycles of spatial mean values of forecast (black lines ) vs observations (red) - January-February, 2015. The error bars represent the mean spatial standard deviation for each hour. Variables shown are T-2m (K, left panels), TD-2m (K, centre) and UV-10 (m s$^{-1}$, right). Again the statistics are divided by region (UAE (top row), Mountain (2nd), Marine (3rd), Desert (4th)).**

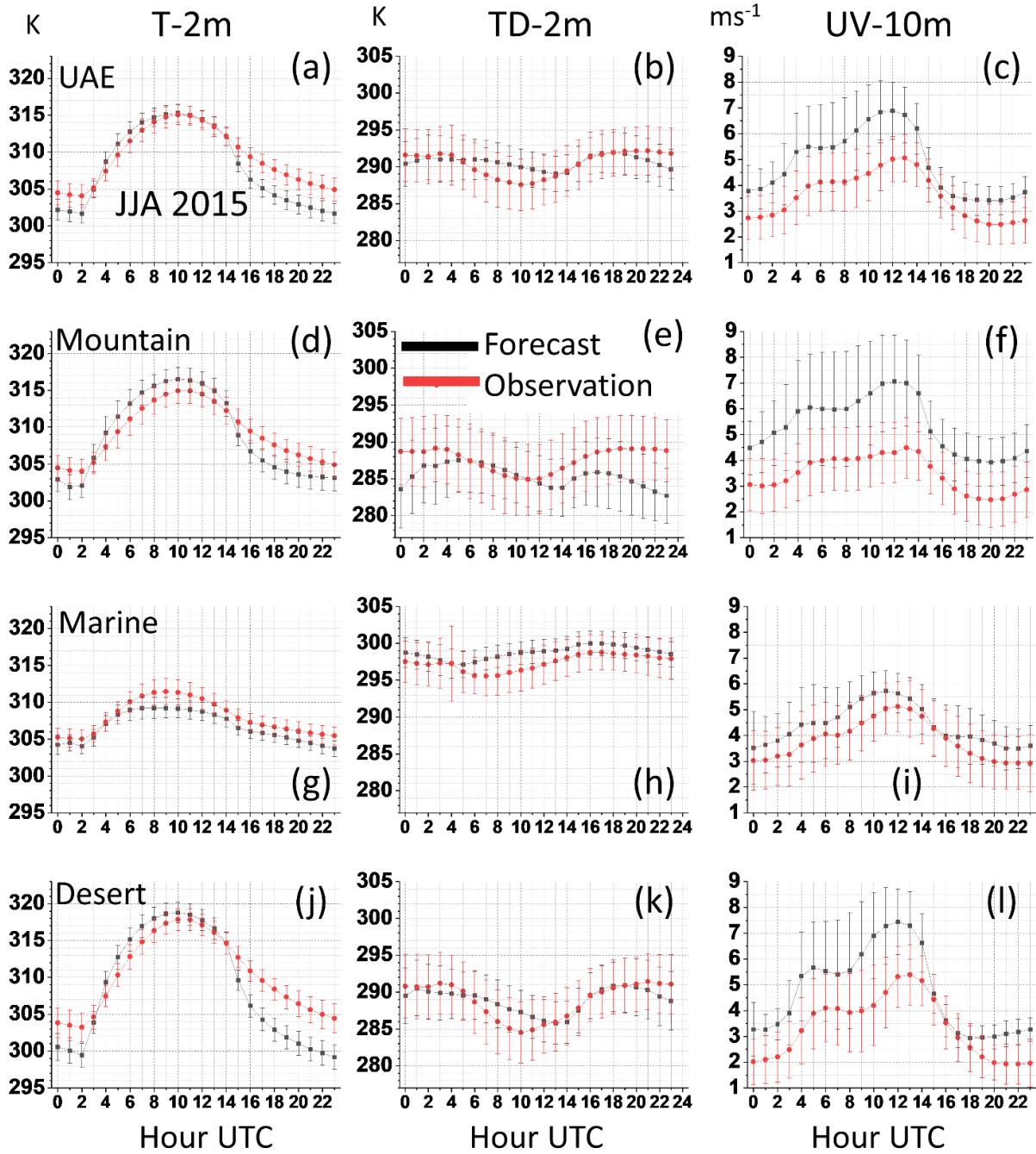

**Figure 9: Summer diurnal cycles. As for Figure 8 except for the period June-August, 2015.**

**Tables**

**Table 1: Selected physics schemes in WRF for sub-grid processes**

| Physics type | Scheme/Option | Reference |
|---|---|---|
| **Land surface scheme** | NOAH-MP | Niu et al., 2011 |
| **Atmospheric surface layer** | MYNN | Nakanishi and Niino, 2006 |
| **Atmospheric boundary layer** | MYNN 2.5 level TKE | Nakanishi and Niino, 2006 |
| **SW radiation** | RRTMG | Mlawer et al., 1997 |
| **LW radiation** | RRTMG | Iacono et al., 2008 |
| **Microphysics** | Thompson-Eidhammer | Thompson and Eidhammer, 2014 |

**Table 2: Summary of main aspects of simulation**

| | | |
|---|---|---|
| **Total duration of daily forecasts** | 01 December 2014 to 30 November 30th 2015 | |
| **Period of analysis** | 01 January 2015 to 30 November 2015 | |
| **WRF output frequency** | 1-hourly | |
| **Verification data frequency** | 1-hourly | 48 surface weather stations |
| **Boundary forcing frequency** | 6-hourly | ECMWF operational analysis (0.12˚) |
| **SST forcing frequency** | 6-hourly | OSTIA data |
| **AOD forcing frequency** | 6-hourly | ECMWF MACC reanalysis |
| **Land use data** | Static | MODIS IGBP - 21 classes |
| **Soil texture** | Static | Modified HWSD (Milovac et al. 2018) |
| **Terrain** | Static | GMTED 2010 |
| **Cold start initialisation** | 18:00 UTC daily | |
| **Fields for reinitialisation** | All except soil moisture – all four soil levels | |
| **Forecast length** | 30 hours (first 6 hours discarded) | |
| **Forecast analysis** | 24 hours - 00:00 to 23:00 UTC | |
| **Model integration timestep** | 15 seconds | |

**Table 3: Number and altitude statistics for the regions – Marine, Desert and Mountain**

| Region | Number of stations | Mean altitude (m) | Minimum (m) | Maximum (m) |
|---|---|---|---|---|
| **Marine** | 17 | 13.8 | 0 | 101 |
| **Mountain** | 16 | 430.2 | 303 | 1485 |
| **Desert** | 15 | 120.0 | 114 | 204 |

**Table 4: 2015 Oceanic Niño Index (ONI) [3 month running mean of ERSST.v5 SST anomalies in the Niño 3.4 region (50˚N-50˚S, 120˚-170˚W)], based on centered 30-year base periods updated every 5 years – NOAA.**

| Jan | Feb | Mar | Apr | May | Jun | Jul | Aug | Sep | Oct | Nov | Dec |
|-----|-----|-----|-----|-----|-----|-----|-----|-----|-----|-----|-----|
| **0.6** | 0.6 | 0.6 | 0.8 | 1 | 1.2 | 1.5 | 1.8 | 2.1 | 2.4 | 2.5 | 2.6 |

**Table 5: Seasonal and regional differences in observed T-2m and TD-2m means to show the closeness to saturation. Included are the number of ~~data points~~time steps for each period ($N_T$). Note that this is not a mean of the T-2m/TD-2m differences calculated at each time step, but an overall difference in means.**

| Season | Region | $N_T$ total | Mean (T-2m - Td-2m) [°C] |
|--------|--------|-------------|--------------------------|
| Winter | UAE | 1416 | 11.2 |
| Winter | Mountain | 1416 | 12.2 |
| Winter | Marine | 1416 | 8.3 |
| Winter | Desert | 1416 | 11.4 |
| Spring | UAE | 2207 | 18.6 |
| Spring | Mountain | 2207 | 21.7 |
| Spring | Marine | 2207 | 11.0 |
| Spring | Desert | 2207 | 20.4 |
| Summer | UAE | 2207 | 21.1 |
| Summer | Mountain | 2208 | 17.3 |
| Summer | Marine | 2208 | 18.2 |
| Summer | Desert | 2207 | 16.6 |
| Autumn | UAE | 2042 | 14.0 |
| Autumn | Mountain | 2182 | 10.2 |
| Autumn | Marine | 2176 | 14.5 |
| Autumn | Desert | 2051 | 11.6 |

**Table A1 (appendix): List of weather stations used for verification of WRF, including ID, coordinates, altitude and assigned region**

| Number | Name | Station ID | Lon | Lat | Altitude (m.a.sl) | Region |
|--------|------|-----------|-----|-----|-------------------|--------|
| 1 | AlAryam | 41202 | 54.1719 | 24.3083 | 11 | Marine |
| 2 | AlDhaid | 41203 | 55.8169 | 25.2369 | 104 | Desert |
| 3 | AlFaqa | 41204 | 55.6214 | 24.7189 | 215 | Mountain |
| 4 | AlMalaiha | 41209 | 55.8881 | 25.1306 | 152 | Desert |
| 5 | AlQor | 41212 | 56.1519 | 24.9064 | 228 | Mountain |
| 6 | AlRuwais | 41214 | 52.8497 | 24.0833 | 13 | Marine |
| 7 | AlShiweb | 41215 | 55.7981 | 24.7761 | 292 | Mountain |
| 8 | AbuDhabi | 41217 | 54.3278 | 24.4772 | 8 | Marine |
| 9 | AlAin | 41218 | 55.7933 | 24.2156 | 302 | Mountain |
| 10 | Dalma | 41220 | 52.2914 | 24.4908 | 10 | Marine |
| 11 | Damsa | 41221 | 55.4133 | 24.18 | 169 | Desert |
| 12 | Dhudna | 41223 | 56.325 | 25.511 | 51 | Marine |
| 13 | FalajAlMoalla | 41224 | 55.8661 | 25.3378 | 96 | Desert |
| 14 | Hamim | 41225 | 54.3028 | 22.9736 | 115 | Desert |
| 15 | Hatta | 41226 | 56.138 | 24.811 | 304 | Mountain |
| 16 | JabalHafeet | 41227 | 55.7753 | 24.0567 | 910 | Mountain |
| 17 | JabalMebreh | 41229 | 56.1294 | 25.6469 | 1485 | Mountain |
| 18 | KhatamAlShaklah | 41230 | 55.9519 | 24.2111 | 406 | Mountain |

| 19 | MadinatZayed | 41231 | 53.6986 | 23.6817 | 113 | Desert |
| 20 | Makassib | 41232 | 51.824 | 24.666 | 0 | Marine |
| 21 | Manama | 41233 | 56.0081 | 25.3853 | 204 | Mountain |
| 22 | Masafi | 41234 | 56.1172 | 25.4475 | 453 | Mountain |
| 23 | Mezaira | 41235 | 53.7786 | 23.145 | 204 | Desert |
| 24 | Mezyed | 41236 | 55.8478 | 24.0286 | 316 | Mountain |
| 25 | Mukhariz | 41237 | 52.8778 | 22.9347 | 142 | Desert |
| 26 | Owtaid | 41238 | 53.1028 | 23.3956 | 145 | Desert |
| 27 | Qasyoura | 41240 | 54.8194 | 22.8286 | 95 | Desert |
| 28 | Raknah | 41242 | 55.7081 | 24.3456 | 282 | Mountain |
| 29 | RasMusherib | 41243 | 51.65 | 24.33 | 0 | Marine |
| 30 | SaihAlSalem | 41246 | 55.3119 | 24.8275 | 78 | Desert |
| 31 | SirBaniYas | 41248 | 52.5978 | 24.3169 | 101 | Marine |
| 32 | SirBuNair | 41249 | 54.2339 | 25.22 | 4 | Marine |
| 33 | Tawiyen | 41251 | 56.0703 | 25.56 | 164 | Desert |
| 34 | UmAzimul | 41252 | 55.1386 | 22.7142 | 114 | Desert |
| 35 | UmGhafa | 41253 | 55.9333 | 24.0667 | 361 | Mountain |
| 36 | UmmAlQuwain | 41254 | 55.6583 | 25.5333 | 12 | Marine |
| 37 | Yasat | 41255 | 51.9883 | 24.1722 | 15 | Marine |
| 38 | ALEjeili | 41256 | 54.1 | 25.02 | 0 | Marine |
| 39 | Ajman | 41258 | 55.4 | 25.42 | 0 | Marine |

| 40 | AlRass | 41259 | 54.3 | 24.45 | 3 | Marine |
| 41 | AlAjban | 41260 | 54.9 | 24.6 | 51 | Desert |
| 42 | AlShuaibah | 41261 | 55.6 | 24.11 | 209 | Mountain |
| 43 | Arylah | 41262 | 54.2 | 24.99 | 0 | Marine |
| 44 | Ashaab | 41264 | 54.8 | 24.39 | 58 | Desert |
| 45 | JabalYanas | 41266 | 56.1 | 25.73 | 684 | Mountain |
| 46 | RasAlkhaimah | 41267 | 55.94 | 25.77 | 7 | Marine |
| 47 | Shoukah | 41269 | 56 | 25.11 | 232 | Mountain |
| 48 | AbuAlBukhoosh | 41274 | 53.146 | 25.495 | 0 | Marine |

100