# Peer review of "Seasonal and diurnal performance of daily forecasts with WRF V3.8.1 over the United Arab Emirates"

_Geoscientific Model Development, 2020_

## Referee Comment (RC1) · Anonymous Referee #1 · 10 Nov 2020

Review of GMD-2020-201 "Seasonal and diurnal performance of daily forecasts with WRF-NOAHMP V3.8.1 over the United Arab Emirates" By Oliver Branch, Thomas Schwitalla, Marouane Temimi, Ricardo Fonseca, Narendra Nelli, Michael Weston, Josipa Milovac, and Volker Wulfmeyer

Summary: This study assessed the performance of a high-resolution (dx=2.7 km) configuration of WRF with the NoahMP land surface model (LSM), over 11 months of daily simulations over the United Arab Emirates (UAE). The variables assessed were 2-m temperature (T2), 2-m dewpoint (TD2), and 10-m wind speed (WS10), with observations from 48 surface weather stations across the UAE, divided into three distinct

climatic zones (mountain, marine, and desert). Analysis of RMSE, bias, and correlation was performed diurnally, seasonally, and annually. Overall, temperatures had a small positive bias in daytime and a stronger negative bias at night and away from the coast. Dewpoint generally had a small bias (especially near the coast) that changed little from day to night. 10-m wind speed had a consistently positive bias, and was largest both inland and during daytime hours. Potential causes for these model biases were identified.

Recommendation: Major Revision.

Major Comments:

1. I think calling this model "WRF-NOAHMP V3.8.1" is misleading. Hyphenating WRF with something else (e.g., WRF-Hydro, WRF-Solar, WRF-Crop, WRF-Urban, WRF-Fire) implies a new set of capabilities based on the WRF modeling system, whereas Noah-MP (and it is Noah-MP, not NOAHMP) is simply one of the physics parameterizations that can be used with WRF. If we use WRF with Thompson microphysics, for instance, we do not refer to it as WRF-Thompson, but simply as WRF. I suggest calling this WRF v3.8.1 with the Noah-MP LSM. I am not sure exactly how best to phrase that in the title, abstract, and elsewhere, but that is something that should be changed throughout.

2. Ensure all tables and figures are numbered according to their order of first mention in the text. What is currently Table 1 clearly should be placed as Table 4, judging from the references in the text that refer to the wrong tables. Several figures are also referenced out of order, and references to incorrect figures are made several times (e.g., lines 311–314). I suggest using the caption/cross-reference macros in Microsoft Word to handle automatic table/figure numbering.

3. Section 3.1/Figs. 2 & 3: I am interested in seeing ERA5 climatologies from El Niño years as a third column in those two figures. How consistent is 2015's geopotential height pattern with the average El Niño pattern in that part of the world?

4. Section 3.3, where you surmise many of the causes that may explain the model biases, are all plausible, but this section still feels a little thin. For instance, when discussing the unclear impact of the PBL scheme on T2, TD2, and WS10 diagnostics, you mention that they are computed either by the PBL scheme using MOST or by Noah-MP. Are those diagnostics always computed by Noah-MP? Or only some of the time? And if only some of the time, then when? There needs to be more clarity and understanding here about what physics scheme is actually controlling the diagnosis of those variables. Also, you mention that Nelli et al. (2020a) found similar wind speed biases within a WRF v3.8 simulation. Was that also in the UAE or a similar environment? That should be pointed out explicitly in the text to further bolster your point that these biases are recurring issues and not unique to your study. Furthermore, when it comes to model spin-up time over UAE for these surface variables, could you test that hypothesis of the benefit of a longer spin-up time over, say, 1 month (or even 2 weeks) of simulations? Or would Nelli's retuned roughness length parameter be able to be applied to these simulations for a similar period of time as a demonstration of benefit here?

Minor Comments and Typos:

1. Indent paragraphs throughout. Without either paragraph indentation or blank lines between paragraphs, the paper is harder to read.

2. In most places where a bold phrase followed by an en dash starts a paragraph, either reword the sentence to make the bold phrase part of the sentence, or make the bold heading a new numbered subsection.

3. Date formats throughout the manuscript should be standardized. GMD may have a required format, but most journals typically use dates formatted as, e.g., 12 January 2015 for the main text, and 12 Jan 2015 in table/figure captions.

4. Ensure all abbreviations are defined (several were not defined), and there is no need to redefine abbreviations multiple times.

5. Please use the Oxford comma in lists (i.e., add a comma before the "and").

6. Most of the dashes (also called hyphens) in the middle of sentences throughout the manuscript are incorrectly used. Some should be changed to commas, some should be changed to semicolons or em dashes (that is often a style preference), and some of the sentences should be reworded to either split them into two sentences or to a single sentence that does not require a clause that is set off by an em dash or semicolon. Please consult an English style guide for the difference between dashes, en dashes, and em dashes, and when it is appropriate to use each one, and when a comma or other punctuation might be more appropriate.

7. There needs to be a comma after every instance of "i.e." and "e.g." throughout the manuscript.

8. Hyphenate compound adjectives, such as "2-m air temperature," "2-m dew point," "10-m wind speed," and "2.7-km grid spacing." By contrast, "grid spacing of 2.7 km" does not require a hyphen.

9. Line 40: Change "general circulation model dataset (GCM)" to "general circulation model (GCM) dataset".

10. Line 44: Add Powers et al. (2017, https://doi.org/10.1175/BAMS-D-15-00308.1) as a reference for the WRF model (in addition to the Skamarock et al. (2008) reference for the WRF v3 technical note).

11. Line 48: Add "in" after "WRF".

12. Line 58: "verified configuration of WRF" — What does this mean?

13. Line 85: Change "north east" to "northeast".

14. Line 88 and Fig. 1 caption: When providing latitudes or longitudes, there should be no spaces around the degree symbols (e.g., 14.775°N).

15. Line 90: I suggest changing "weather systems" to "predominant patterns." A

weather system evokes something more transient and singular, whereas a pattern is something larger-scale and longer-lasting.

16. Line 110: Change "north-westerlies" to "northwesterly".

17. Line 111: Change "direction" to "directions".

18. Lines 111–113: "The sea surface of the Arabian Gulf..." — Please rephrase this sentence so as to not imply that the sea surface itself is shallow.

19. Lines 120–121: "This domain was selected..." — This sentence is awkward. Please revise. Also, what do you mean by "twin experiment"? Additionally, it is customary to describe domain dimensions as "900 x 700" rather than "900(x) by 700(y)". It is understood by convention that the x-dimension is given first.

20. Section 2.3.1 and Table 2: The microphysics scheme should be referred to as the "Thompson-Eidhammer aerosol-aware microphysics scheme" or, for short, "Thompson-Eidhammer". And on this note, did you verify that all the options in WPS & WRF were set properly to use the QNWFA/QNIFA monthly aerosol climatology with the Thompson-Eidhammer scheme? If not, then what did you use for aerosols? Did it default to settings in the source code? In my experience, it is really easy for people to think they are using Thompson-Eidhammer, when actually they are essentially using the basic Thompson scheme because of forgetting some namelist settings in WPS and/or WRF.

21. Line 147: Change "Surface Exchange over Land model - HTESSEL" to "Surface Exchange over Land (HTESSEL) model".

22. Line 198: Change "year" to either "yearly" or "annual", and change "season" to "seasonal".

23. Line 165: Change "((Danielson, J.J., Gesch, 2011))" to "(Danielson and Gesch, 2011)".

24. Line 181: Insert a comma after "strongly".

25. Line 199: Add "aim" after "Another".

26. Line 231: Change "In order to get" to "To obtain".

27. Line 249: Add "the" before "diurnal".

28. Line 286 and elsewhere: Change "m.a.s.l" to "m ASL" or "m AMSL". You could also say "station 41229 at elevation 1485 m" and not even mention ASL/AMSL explicitly, because it is implied when giving a station elevation.

29. Line 288: Insert commas around the phrase, "such as differences in mountain and desert cloud cover", and after the references that follow "for instance".

30. Line 292: Delete "very".

31. Line 293: Add a comma after "(Figure 6b)" and change "is" to "are".

32. Line 299: Add a semicolon before "however" and a comma after it.

33. Line 305: Change "in Section 3" to "below".

34. Line 309: Delete the comma after "nevertheless".

35. Line 316: Add "radiation" after "shortwave".

36. Line 318: Delete "the" before "associated".

37. Line 326: Change "(5e)" to "(Fig. 5e)".

38. Line 351: Change "(delta)x 100m" to "(delta)x = 100 m".

39. Line 352: Change "operation" to "operational".

40. Line 364: Change "who" to "that".

41. Line 372: Change "there is a good chance" to "it is likely".

42. Line 378: Change "Yonsei YSU scheme" to "Yonsei University (YSU) PBL scheme".

43. Line 379: Change "Jimenez" to "Jiménez" (add the accent mark, as in the reference).

44. Lines 390–391: The sentence that starts, "Nelli et al., (2020a) found..." is confusing and must be reworded. Did you mean that there are positive biases when the wind speed is < 4 m/s? Also, "windspeeds" is not a word. It should be "wind speed".

45. Line 398: I sincerely doubt that if a center is running a daily WRF run for 30 h operationally (6 h spin-up + 24 h data retained) that it would be unfeasible to run it for 36 h in order to gain an extra 6 h of spin-up. The extra computational cost is only marginal for a once-daily run. Please revise the comment about this issue (there are also some grammatical issues with it as-is).

46. Line 404: Change "information on model in respect to" to "information about the model with respect to".

47. Line 408: Add a comma before "particularly".

48. Line 409: Add a comma after "biases".

49. Line 415: Add a comma after "UV-10m bias".

50. Fig. 2 caption: Change "ms-1" to "m s-1". Without the space, the unit becomes inverse milliseconds.

51. Fig 5 caption: Only use "and" before the last item in a series.

52. Fig. 6: I suggest changing the units from K to °C.

53. Table 1: First, this should be Table 4. Second, consider making this a simple figure?

---

## Author Comment (AC1) · 16 Nov 2020

Dear Reviewer 1,

Many thanks for your detailed and very useful review. We will wait for all reviews to be received before responding in detail.

Best Regards Oliver Branch
* * *

---

## Referee Comment (RC2) · Anonymous Referee #2 · 30 Nov 2020

The manuscript "Seasonal and diurnal performance of daily forecasts with WRF-NOAHMP V3.8.1 over the United Arab Emirates" by Branch et al. presents the verification of high resolution ($\Delta x = \Delta y \sim 3$km) daily forecasts (T+6h – T+30h), produced by a regional numerical weather prediction model (WRFv3.8.1) over the United Arab Emirates, at seasonal and diurnal time scales. The simulations performed according to well-established methods (daily re-initialization with operational analysis data, relative sufficient spin-up time, soil moisture treatment, high quality soil, land and topography data, inclusion of re-analysis AOD data) and have been evaluated utilizing robust statistical metrics. The manuscript is well structured, the materials and methods are referenced accordingly, the results are presented in a comprehensive way and the

summary outlines the findings of the study and elaborates possible limitations. However, some adjustments and minor revision are required prior to publication. Overall, I would recommend publication subject to the general and specific comments below.

General comments:

1. In Figure 1b, the authors present the latitude-longitude values of the four model corner grid cells. The authors should clarify whether the model domain is on regular lat-lon projection or not.

2. Although it is mentioned later in the manuscript, the authors are encouraged to add a short explanation in sub-section 2.2, why the December of 2014 was excluded from their analysis.

3. In Table 1, where the physical suite is presented, the authors should add in parentheses the corresponding namelist options for clarity.

4. Regarding the treatment of soil moisture from one initialization to the next (L183-189), the authors should elaborate more on the employed method (e.g. which time stamp was considered as valid between two consecutive initializations). From the way it is written, the simulated soil moisture from the WRF-NOAMP replaces the corresponding field in the initial condition file at initialization time (18UTC), rather than the soil moisture from the operation analyses. If the latter is true, what about the lateral conditions?

5. In subsection 2.5.2, the authors should clarify if the closest to observation model grid point was considered for the verification, or other approach (e.g. the 4-point weighted mean) was used.

6. Figure 7 is cited before Figure 6

7. In Figures 2 and 3, the shaded contours corresponding to geopotential heights at 500hPa isobaric level, according to the legend and Figures' captions. However, their values are relative low and their range corresponds to a lower isobaric level (perhaps at

700-750hPa). The authors should check for any inconsistencies between each figure legend/caption and the corresponding contours.

8. In Figure 6, the y-axes in T-2m and TD-2m panels should share a common range in their values, in order the saturation conditions to be more comparable across different seasons.

9. In Table 5, the number of UAE data points (from all 48 stations; 3rd column) are equal to the data points on each sub-region. How is this possible, since a subset of the available stations are utilized in each region? The authors should clarify the latter.

10. The authors should elaborate if the biases in UV-10m speed are introduced also due to differences in height between observed and simulated values. In other words, how certain are the authors that the observed wind speed values are at 10 m above ground level?

11. In order to point further the tremendous effort on performing the simulations, the authors should consider to add some information about the wall-clock time of each re-initialization, the number of cores and some hardware specifications.

12. As a general comment, the authors should also considered the uncertainty of the observations themselves (Prein and Gobiet. 2017).

Specific comments:

L40: Citation Coppola et al., 2018 should now Coppola et al., 2020. Please revise.

L47-48: Please rephrase.

L80: Add "and" after "(2.4).

L179-180: Please rephrase.

L311: Please change "Figure 6a-6h" to Figure "5a-5h".

L341: Please in which Figure this sentence refers.

[Figure]

**References**

Prein, Andreas F., and Andreas Gobiet. "Impacts of Uncertainties in European Gridded Precipitation Observations on Regional Climate Analysis." International Journal of Climatology 37, no. 1 (January 2017): 305–27. https://doi.org/10.1002/joc.4706.
* * *

---

## Author Response (AR1)

**Point-by-point letter to reviewers**

Title: "Seasonal and diurnal performance of daily forecasts with WRF V3.8.1 over the United Arab Emirates"

Authors: Oliver Branch, Thomas Schwitalla, Marouane Temimi, Ricardo Fonseca, Narendra Nelli, Michael Weston, Josipa Milovac, Volker Wulfmeyer

Dear Editor,

We would like to thank both reviewers for their insightful and constructive comments, which we would like to address point by point:

Note: Line number references relate to the revised un-marked up manuscript, not the original manuscript.

Note: Authors replies are in bold. New lines added are in quotes in bold italic.

Note: We have uploaded all final revised figures to the Supplement section as a zip.

**Referee 1:**

Summary: This study assessed the performance of a high-resolution (dx=2.7 km) configuration of WRF with the NoahMP land surface model (LSM), over 11 months of daily simulations over the United Arab Emirates (UAE). The variables assessed were 2-m temperature (T2), 2-m dewpoint (TD2), and 10-m wind speed (WS10), with observations from 48 surface weather stations across the UAE, divided into three distinct climatic zones (mountain, marine, and desert). Analysis of RMSE, bias, and correlation was performed diurnally, seasonally, and annually. Overall, temperatures had a small positive bias in daytime and a stronger negative bias at night and away from the coast. Dewpoint generally had a small bias (especially near the coast) that changed little from day to night. 10-m wind speed had a consistently positive bias, and was largest both inland and during daytime hours. Potential causes for these model biases were identified.

Recommendation: Major Revision.

**Major Comments:**

*1.		I think calling this model "WRF-NOAHMP V3.8.1" is misleading. Hyphenating WRF with something else (e.g., WRF-Hydro, WRF-Solar, WRF-Crop, WRF-Urban, WRF-Fire) implies a new set of capabilities based on the WRF modeling system, whereas Noah-MP (and it is Noah-MP, not NOAHMP) is simply one of the physics parameteri-zations that can be used with WRF. If we use WRF with Thompson microphysics, for instance, we do not refer to it as WRF-Thompson, but simply as WRF. I suggest calling this WRF v3.8.1 with the Noah-MP LSM. I am not sure exactly how best to phrase that in the title, abstract, and elsewhere, but that is something that should be changed throughout.*

**The authors agree. We have now simply dropped the reference to NOAHMP within the title so it now reads (Lines 1-2):**

'***Seasonal and diurnal performance of daily forecasts with WRF V3.8.1 over the United Arab Emirates***'.

**NOAH-MP is in any case introduced and described with the other physics schemes within the model setup section.**

*2.		Ensure all tables and figures are numbered according to their order of first mention in the text. What is currently Table 1 clearly should be placed as Table 4, judging from the references in the text that refer to the wrong tables. Several figures are also referenced out of order, and references to incorrect figures are made several times (e.g., lines 311–314). I suggest using the caption/cross-reference macros in Microsoft Word to handle automatic table/figure numbering.*

**Many thanks for spotting this. All 'first' mentions of figures and tables are now in order, and all references to figures and tables are now checked and relate to the correct figures/tables.**

*3.		Section 3.1/Figs. 2 & 3: I am interested in seeing ERA5 climatologies from El*

*Niño years as a third column in those two figures. How consistent is 2015's geopotential height pattern with the average El Niño pattern in that part of the world?*

**The authors concur that this would be interesting. However, adding a third column would make the other panels much too small to read, and due to space constraints it is not feasible to add extra figures. In any case, there were only 5 'strong' El Niño years since 1979 (the span of our ERA5 dataset), which represents a very small sample size for judging mean El Niño conditions. One would then also require further analyses of variability.**

**Further, we are convinced that, although El Niño conditions are interesting, a deeper analysis of El Niño is moving away from the main purpose of this study. We are satisfied that we have sufficiently demonstrated the representativeness of this year 2015 in Figures 2 and 3.**

*4.      Section 3.3, where you surmise many of the causes that may explain the model biases, are all plausible, but this section still feels a little thin. For instance, when dis-cussing the unclear impact of the PBL scheme on T2, TD2, and WS10 diagnostics, you mention that they are computed either by the PBL scheme using MOST or by Noah-MP. Are those diagnostics always computed by Noah-MP? Or only some of the time? And if only some of the time, then when? There needs to be more clarity and understanding here about what physics scheme is actually controlling the diagnosis of those variables.*

**The authors agree that this section could benefit from additional explanation. Hence, we have now added the following text and 2 citations (L406-418):**

**"However, the total impact of the PBL scheme selection on reproduction of the T-2m, TD-2m and UV-10m diagnostics is not completely clear. This is because the method of calculation of transfer coefficients/fluxes are executed in NOAH-MP, the PBL scheme, and the surface layer scheme (SLS) depends on the land surface type. In WRF, PBL schemes are generally coupled to the SLS, and typically all variables between the land surface and lowest model layer are diagnosed (e.g. T-2m, U-10m, V-10m). These calculations in the SLS are based on Monin-Obhukov similarity theory, and are represented in the model as hard-coded parameters and/or formulations of similarity functions. The latter are used to obtain dimensionless bulk transfer coefficients which are used for calculating momentum, heat, and moisture fluxes, and for diagnosing near surface quantities like T-2m. These coefficients re-enter the LSM and are to calculate the surface fluxes which then enter the PBL scheme, as the lower boundary condition. Therefore, bias in near-surface variables is strongly related to the choice of LSM and SLS. In this WRF configuration, the communication link between the SLS and NOAH-MP is broken, as NOAH-MP itself calculates transfer coefficients and diagnostics over land surfaces, effectively bypassing the SLS (Nielson et al, 2013). The SLS only becomes active over water surfaces. This means that when NOAH-MP is used, the LSM probably has a stronger impact on the bias of near surface variables than PBL and SLS (e.g. Milovac et al. 2016)."**

*Also, you mention that Nelli et al. (2020a) found similar wind speed biases within a WRF v3.8 simulation. Was that also in the UAE or a similar environ-ment? That*

*should be pointed out explicitly in the text to further bolster your point that these biases are recurring issues and not unique to your study.*

**This is a good suggestion, and indeed these findings were from our colleagues and relate to the same region. We have now added the words "over the UAE region" to the following sentence (L420-423):**

***"Nelli et al., (2020a) found positive wind speed biases over the UAE region of < 4 ms⁻¹ and negative biases for wind speeds > 6 ms⁻¹ within a WRF V3.8 simulation. We have a similar behavior at night in the marine and desert regions, as exhibited by the positive-to-negative distribution of errors increasing with wind speed."***

*Furthermore, when it comes to model spin-up time over UAE for these surface variables, could you test that hypothesis of the benefit of a longer spin-up time over, say, 1 month (or even 2 weeks) of simulations?*

**This is an interesting idea. However, we argue that spin-up (scale of days/weeks) is of minor significance when using a NWP configuration, because the inside-domain atmospheric conditions are re-initialized daily (unlike a climate mode run where only the boundary conditions are updated). Thus, aside from the soil moisture state, these are essentially one-day forecasts with a daily spin-up. This renders long spin-up tests as rather redundant. Given that the regional soils are for the most part consistently arid over the UAE, a one-month spin-up is more than sufficient, and in fact a considerable 'overkill'.**

**However, in accordance with your comment, we have added further information on the one-month spin-up within the following modified sentence in section 2.3.3 (L173-174):**

***"The objective of this study was to run a series of daily forecasts with WRF for the period 01 January to 30 November 2015, with a discarded one-month spin up run from 01 December 2014"***

*Or would Nelli's retuned roughness length parameter be able to be applied to these simulations for a similar period of time as a demonstration of benefit here?*

**The authors agree that this is an interesting prospect for future simulations. Sadly, one cannot test the sensitivity of every model parameter/component within a single study. However, we trust that it is sufficient that the potential relevance of the published roughness parameter findings is recognized and discussed several times in the study, including in the outlook.**

**Minor Comments and Typos:**

*5.    Indent paragraphs throughout. Without either paragraph indentation or blank lines between paragraphs, the paper is harder to read.*

**We agree. We have now added a space between paragraphs to aid clarity.**

*6.    In most places where a bold phrase followed by an en dash starts a paragraph,*

*either reword the sentence to make the bold phrase part of the sentence, or make the bold heading a new numbered subsection.*

**These bold sections have now all been amended to subsections.**

*7.	Date formats throughout the manuscript should be standardized. GMD may have a required format, but most journals typically use dates formatted as, e.g., 12 January 2015 for the main text, and 12 Jan 2015 in table/figure captions.*

**Agreed. These have now been standardized in all text/captions/tables to conform to the preferred GMD format, e.g. 20 January 2015.**

*8.	Ensure all abbreviations are defined (several were not defined), and there is no need to redefine abbreviations multiple times.*

**This has been checked, but no omissions or repetitions were found.**

*9.	Please use the Oxford comma in lists (i.e., add a comma before the "and").*

**These have now been added throughout the text.**

*10.	Most of the dashes (also called hyphens) in the middle of sentences throughout the manuscript are incorrectly used. Some should be changed to commas, some should be changed to semicolons or em dashes (that is often a style preference), and some of the sentences should be reworded to either split them into two sentences or to a single sentence that does not require a clause that is set off by an em dash or semicolon. Please consult an English style guide for the difference between dashes, en dashes, and em dashes, and when it is appropriate to use each one, and when a comma or other punctuation might be more appropriate.*

**Agreed. These have been checked and where necessary modified throughout.**

*11.	There needs to be a comma after every instance of "i.e." and "e.g." throughout the manuscript.*

**Agreed. This has been done.**

*12.	Hyphenate c to add more figures. The authors are convinced thatompound adjectives, such as "2-m air temperature," "2-m dew point," "10-m wind speed," and "2.7-km grid spacing." By contrast, "grid spacing of 2.7 km" does not require a hyphen.*

**This has now been modified throughout.**

*13.	Line 40: Change "general circulation model dataset (GCM)" to "general circulation model (GCM) dataset".*

**Done.**

*14.	Line 44: Add Powers et al. (2017, https://doi.org/10.1175/BAMS-D-15-00308.1) as a reference for the WRF model (in addition to the Skamarock et al. (2008) reference for the WRF v3 technical note).*

**Done (L44 and L120)**.

*15.    Line 48: Add "in" after "WRF".*

**Done**.

*16.    Line 58: "verified configuration of WRF" âAT What does this mean?*

**The word verified has been removed here, as it is anyway mentioned a few lines later in conjunction with our sensitivity study (Schwitalla et al., (2020))**.

*Line 85: Change "north east" to "northeast".*

**Done**.

*17.    Line 88 and Fig. 1 caption: When providing latitudes or longitudes, there should be no spaces around the degree symbols (e.g., 14.775 N).*

**Done**.

*18.    Line 90: I suggest changing "weather systems" to "predominant patterns." A C4*

**Done**.

*19.    weather system evokes something more transient and singular, whereas a pattern is something larger-scale and longer-lasting.*

***Agreed.***

*20.    Line 110: Change "north-westerlies" to "northwesterly".*

**Done**.

*21.    Line 111: Change "direction" to "directions".*

**Done**.

*22.    Lines 111–113: "The sea surface of the Arabian Gulf: : :" Please rephrase this sentence so as to not imply that the sea surface itself is shallow.*

**This sentence has been changed to (L114-116):**

**"In the Arabian Gulf, which is relatively shallow (maximum depth ~90m), particularly close to the UAE coast, the sea surface can heat rapidly, with temperatures often exceeding 30˚C (Al Azhar et al., 2016)."**

*23.    Lines 120–121: "This domain was selected: : :" This sentence is awkward. Please revise.*

*Also, what do you mean by "twin experiment"?*

**The word twin is removed and the sentence rephrased to (L122-124):**

*"The domain size and grid spacing matches that of a previous simulation (see Schwitalla et al., (2020)), and is comprised of 900 by 700 grid cells horizontally (see Figure 1b)."*

*Additionally, it is custom-ary to describe domain dimensions as "900 x 700" rather than "900(x) by 700(y)". It is understood by convention that the x-dimension is given first.*

**Done.**

*24.    Section 2.3.1 and Table 2: The microphysics scheme should be referred to as the "Thompson-Eidhammer aerosol-aware microphysics scheme" or, for short, "Thompson-Eidhammer". And on this note, did you verify that all the options in WPS & WRF were set properly to use the QNWFA/QNIFA monthly aerosol climatology with the Thompson-Eidhammer scheme? If not, then what did you use for aerosols? Did it default to settings in the source code? In my experience, it is really easy for people to think they are using Thompson-Eidhammer, when actually they are essentially us-ing the basic Thompson scheme because of forgetting some namelist settings in WPS and/or WRF.*

**Yes, this requires clarification in the text. The aerosol-aware component was not used for this configuration. We incorporated the 12-hourly aerosol optical depth data instead in this case. We have now clarified this in the text as follows (L138-141):**

*"Other physics schemes included were RRTMG for long and shortwave radiation transfer (Iacono et al., 2008; Mlawer et al., 1997), the Thompson-Eidhammer microphysics scheme (Thompson and Eidhammer, 2014), (although without the aerosol-aware component activated), the MYNN scheme for the atmospheric surface layer, and the MYNN 2.5 level TKE scheme for the boundary layer (Nakanishi and Niino, 2006) (See Table 1 for a synopsis of physics schemes and their associated references)."*

*25.    Line 147: Change "Surface Exchange over Land model - HTESSEL" to "Surface Exchange over Land (HTESSEL) model".*

**Done.**

*26.    Line 198: Change "year" to either "yearly" or "annual", and change "season" to "seasonal".*

**Done.**

*27.    Line 165: Change "((Danielson, J.J., Gesch, 2011))" to "(Danielson and Gesch, 2011)".*

**Done.**

*28.    Line 181: Insert a comma after "strongly".*

**Done.**

29.   Line 199: Add "aim" after "Another".

**Done.**

30.   Line 231: Change "In order to get" to "To obtain".

**Done.**

31.   Line 249: Add "the" before "diurnal".

**Done.**

32.   Line 286 and elsewhere: Change "m.a.s.l" to "m ASL" or "m AMSL". You could also say "station 41229 at elevation 1485 m" and not even mention ASL/AMSL explicitly, because it is implied when giving a station elevation

**Done.**

33.   Line 288: Insert commas around the phrase, "such as differences in mountain and desert cloud cover", and after the references that follow "for instance".

**Done.**

34.   Line 292: Delete "very".

**Done.**

35.   Line 293: Add a comma after "(Figure 6b)" and change "is" to "are".

**Done.**

36.   Line 299: Add a semicolon before "however" and a comma after it.

**Done.**

37.   Line 305: Change "in Section 3" to "below".

**Done.**

38.   Line 309: Delete the comma after "nevertheless".

**Done.**

39.   Line 316: Add "radiation" after "shortwave".

**Done.**

40.   Line 318: Delete "the" before "associated".

**Done.**

*41.    Line 326: Change "(5e)" to "(Fig. 5e)".*

**Done.**

*42.    Line 351: Change "(delta)x 100m" to "(delta)x = 100 m".*

**Done.**

*43.    Line 352: Change "operation" to "operational".*

**Done.**

*44.    Line 364: Change "who" to "that".*

**Done.**

*45.    Line 372: Change "there is a good chance" to "it is likely".*

**Done.**

*46.    Line 378: Change "Yonsei YSU scheme" to "Yonsei University (YSU) PBL scheme".*

**Done.**

*47.    Line 379: Change "Jimenez" to "Jiménez" (add the accent mark, as in the reference).*

**Done.**

*48.    Lines 390–391: The sentence that starts, "Nelli et al., (2020a) found: : :" is confus-ing and must be reworded. Did you mean that there are positive biases when the wind speed is < 4 m/s? Also, "windspeeds" is not a word. It should be "wind speed".*

**The authors agree it needs re-phrasing. To clarify this, we have modified the sentence to make it clearer when +/- ve biases occur (L420-422):**

**"Nelli et al., (2020a) found positive wind speed biases over the same region when wind speeds were < 4 m s-1 and negative biases for wind speeds which were > 6 m s-1 within a WRF V3.8 simulation."**

*49.    Line 398: I sincerely doubt that if a center is running a daily WRF run for 30 h operationally (6 h spin-up + 24 h data retained) that it would be unfeasible to run it for 36 h in order to gain an extra 6 h of spin-up. The extra computational cost is only marginal for a once-daily run. Please revise the comment about this issue (there are also some grammatical issues with it as-is).*

**The authors agree regarding the feasibility of extended spin-ups for operational forecasting in NWP mode. Therefore, we have clarified these points**

**by modifying the end of the following paragraph to reflect this (L425-428):**

*"Another possibility is the length of the forecast spin-up, the required length of which may still be uncertain. We have already mentioned that Chaouch et al., (2017) cited a 5-h spin-up as being sufficient, but Hahmann et al., (2015) posits that the necessary spin-up over land could be 12 hours or even more (primarily for effective use of the PBL scheme). However, such long spin-ups are likely to be (i) prohibitively expensive and (ii) too time consuming for forecasting purposes."*

*50. Line 404: Change "information on model in respect to" to "information about the model with respect to".*

**Done.**

*51. Line 408: Add a comma before "particularly".*

**Done.**

*52. Line 409: Add a comma after "biases".*

**Done.**

*53. Line 415: Add a comma after "UV-10m bias".*

**Done.**

*54. Fig. 2 caption: Change "ms-1" to "m s-1". Without the space, the unit becomes inverse milliseconds.*

**Corrected.**

*55. Fig 5 caption: Only use "and" before the last item in a series.*

**Corrected.**

*56. Fig. 6: I suggest changing the units from K to C.*

**Done.**

*57. Table 1: First, this should be Table 4. Second, consider making this a simple figure?*

**The table numbers and order have been checked and corrected. We are satisfied with the table format.**

**Referee 2:**

*The manuscript "Seasonal and diurnal performance of daily forecasts with WRF-NOAHMP V3.8.1 over the United Arab Emirates" by Branch et al. presents the ver-ification of high resolution ( x= y 3km) daily forecasts (T+6h – T+30h), produced by a regional numerical weather prediction model (WRFv3.8.1) over the United Arab Emirates, at seasonal and diurnal time scales. The simulations performed accord-ing to well-established methods (daily re-initialization with operational analysis data, relative sufficient spin-up time, soil moisture treatment, high quality soil, land and to-pography data, inclusion of re-analysis AOD data) and have been evaluated utilizing robust statistical metrics. The manuscript is well structured, the materials and methods are referenced accordingly, the results are presented in a comprehensive way and the summary outlines the findings of the study and elaborates possible limitations. How-ever, some adjustments and minor revision are required prior to publication. Overall, I would recommend publication subject to the general and specific comments below.*

**The authors thank the reviewer for this positive assessment.**

*General comments:*

*1.      In Figure 1b, the authors present the latitude-longitude values of the four model corner grid cells. The authors should clarify whether the model domain is on regular lat-lon projection or not.*

**The authors agree on clarifying this point, and have changed the text accordingly (L89):**

**"The model uses a regular latitude-longitude grid and has corner grid cells located at 14.775˚N, 32.225˚N, 43.275˚E, and 65.725˚E."**

**We have also added it here (L123):**

**"The domain size and grid spacing matches that of a previous simulation by Schwitalla et al., (2020)), and is comprised of a regular latitude-longitude grid with 900 by 700 cells horizontally (see Figure 1b)."**

*2.      Although it is mentioned later in the manuscript, the authors are encouraged to add a short explanation in sub-section 2.2, why the December of 2014 was excluded from their analysis.*

**The authors agree. We have now added a sentence in 2.3.3 (L174-176).**

**"Note that December 2014 was not used for verification (observation data was in any case not available at that time. See Section 2.4). It also makes sense not to analyze a winter season split over two years."**

*In Table 1, where the physical suite is presented, the authors should add in paren-theses the corresponding namelist options for clarity.*

We have fully discussed the selection of schemes used within the study, and we judge that the actual namelist settings are technical rather than scientific information. However, this information is fully accessible to modellers who want to see the formatted namelist. It, and all other relevant files, is linked to and freely available on the Zenodo servers at DOI: 10.5281/zenodo.3894491

*3.       Regarding the treatment of soil moisture from one initialization to the next (L183-189), the authors should elaborate more on the employed method (e.g. which time stamp was considered as valid between two consecutive initializations). From the way it is written, the simulated soil moisture from the WRF-NOAMP replaces the corre-sponding field in the initial condition file at initialization time (18UTC), rather than the soil moisture from the operation analyses. If the latter is true, what about the lateral conditions?*

**The authors agree this could be clearer. The soil moisture is overwritten at 18:00 from each consecutive day to the next, for the start of each new forecast. The lateral boundary conditions are as for a climate mode run, i.e., every 6 hours. The atmospheric state within the domain boundaries is reinitialized each day at 18:00.**

**We have clarified this by adding a sentence at the end of 2.3.3 (L198-201):**

**"To summarize the NWP configuration: the soil moisture is overwritten at 18:00 from each consecutive day to the next, for the start of each new forecast. The lateral boundary conditions are as for a climate mode run, i.e., input every 6 hours from the forcing data. The atmospheric state within the domain boundaries is reinitialized each day at 18:00."**

*4.       In subsection 2.5.2, the authors should clarify if the closest to observation model grid point was considered for the verification, or other approach (e.g. the 4-point weighted mean) was used.*

**We agree, and have now added the following sentence in 2.5.2 (L247-248):**

**"All comparisons were made using NCAR's Model Evaluation Tools V9.0 (MET) package, utilizing a nearest-grid cell approach on an hourly temporal resolution."**

*5.       Figure 7 is cited before Figure 6*

**All figure and table citations and order of first-mention have now been checked and are correct.**

*6.       In Figures 2 and 3, the shaded contours corresponding to geopotential heights at 500hPa isobaric level, according to the legend and Figures' captions. However, their values are relative low and their range corresponds to a lower isobaric level (perhaps at 700-750hPa). The authors should check for any inconsistencies between each figure legend/caption and the corresponding contours.*

**Thank you for noticing this error. The problem was not an incorrect pressure**

height, but a bug in the NCL code where the geopotential field is divided by gravitational acceleration. We have corrected the plots now. The resulting patterns remain spatially consistent so they appear essentially identical apart from changes to the color bar values. Accordingly, the analysis in the text does not require alteration.

*7.     In Figure 6, the y-axes in T-2m and TD-2m panels should share a common range in their values, in order the saturation conditions to be more comparable across different seasons.*

**Our reasoning for scaling the figures this way is that using a common scale would make the TD-2m figures too squashed and difficult to read, especially in this multi-panel plot. We argue that our scaling is the most optimal compromise and means both variables are highly readable.**

*8.     In Table 5, the number of UAE data points (from all 48 stations; 3rd column) are equal to the data points on each sub-region. How is this possible, since a subset of the available stations are utilized in each region? The authors should clarify the latter.*

**This number of data points represents only the number of timesteps within each period. We have clarified this by modifying the table and caption accordingly:**

**"Table 5: Seasonal and regional differences in observed T-2m and TD-2m means to show the closeness to saturation. Included are the number of time steps for each period ($N_T$)."**

*9.     The authors should elaborate if the biases in UV-10m speed are introduced also due to differences in height between observed and simulated values. In other words, how certain are the authors that the observed wind speed values are at 10 m above ground level?*

**This is a very good point. We can confirm that the wind measurement height is WMO standard at 10 m at all stations in the UAE. They are all operated by the UAE's National Centre of Meteorology NCM, who utilize a consistent tower and instrumentation configuration. The authors have visited several of these stations. This height corresponds with the diagnostic 10 m height in the WRF output.**

**To clarify this compliance with WMO standards, which requires this height, we have modified a sentence in the abstract (L17):**

**"WRF was verified using measurements of 2-m air temperature (T-2m), 2-m dew point (TD-2m), and 10m wind speed (UV-10m) from 48 UAE WMO-compliant surface weather stations. Analysis was made of seasonal and diurnal performance within the desert, marine, and mountain regions of the UAE."**

**And in the introduction (L65):**

**"Our main objective is to assess the seasonal and diurnal performance of WRF – both qualitatively and quantitatively – in reproducing surface air temperature, dew point and wind data from 48 WMO-compliant surface weather stations distributed over the UAE."**

*10.  As a general comment, the authors should also considered the uncertainty of the observations themselves (Prein and Gobiet. 2017).*

**This is a very good idea and we have modified the following text in the summary accordingly (L447-450):**

**"Ultimately, no model downscaling forecast (at scales economically viable for forecasting) can be expected to exhibit exceptional skill in all conditions. A caveat generally when evaluating models is that one must factor in a certain level of error in station or gridded observational datasets themselves (e.g., as discussed by Prein and Gobiet, 2017). Nevertheless, we have discussed several avenues for improvement on this application of WRF."**

**Specific comments:**

*11.  L40: Citation Coppola et al., 2018 should now Coppola et al., 2020. Please revise.*

**Thank you. This has been updated.**

*12.  L47-48: Please rephrase.*

**Agreed. We have re-phrased this sentence for clarity as follows (L47-48):**

**"Until now, there have been few annual-scale verification studies employing the WRF model on a NWP daily forecasting mode at such high spatiotemporal resolution (e.g. $dx < 2 - 3$ km)."**

*13.  L80: Add "and" after "(2.4).*

**Done**

*14.  L179-180: Please rephrase.*

**Agreed. We have re-phrased this sentence for clarity as follows (L187-189):**

**"By reinitialising the 3D state within the domain itself (as opposed to simply inputting lateral boundary conditions), we ensure the atmospheric state is closer to the forecast provided by ECMWF, than would be the case in e.g., typical climate mode simulations."**

*15.  L311: Please change "Figure 6a-6h" to Figure "5a-5h".*

**Corrected**

*16.  L341: Please in which Figure this sentence refers.*

**This sentence refers to Figure 7. We have now updated this sentence to clarify this (L362-363):**

*"TD-2m is relatively well estimated in 2015 over the UAE as a whole, with correlations around 0.7 and biases of less than 1˚C (Figure 7d and 7e, UAE sections)."*

**Concluding remarks from the authors**

Finally, we thank the reviewers once again for their expertise, very useful remarks and correction of errors. We think that the adjustments made and additions suggested have greatly improved the manuscript and we trust that the article is ready for publication in GMD.

**New references**

Coppola, E., Sobolowski, S., Pichelli, E., Raffaele, F., Ahrens, B., Anders, I., Ban, N., Bastin, S., Belda, M., Belusic, D., Caldas-Alvarez, A., Cardoso, R. M., Davolio, S., Dobler, A., Fernandez, J., Fita, L., Fumiere, Q., Giorgi, F., Goergen, K., Güttler, I., Halenka, T., Heinzeller, D., Hodnebrog, Jacob, D., Kartsios, S., Katragkou, E., Kendon, E., Khodayar, S., Kunstmann, H., Knist, S., Lavín-Gullón, A., Lind, P., Lorenz, T., Maraun, D., Marelle, L., van Meijgaard, E., Milovac, J., Myhre, G., Panitz, H. J., Piazza, M., Raffa, M., Raub, T., Rockel, B., Schär, C., Sieck, K., Soares, P. M. M., Somot, S., Srnec, L., Stocchi, P., Tölle, M. H., Truhetz, H., Vautard, R., de Vries, H. and Warrach-Sagi, K.: A first-of-its-kind multi-model convection permitting ensemble for investigating convective phenomena over Europe and the Mediterranean, Clim. Dyn., 55(1–2), 3–34, doi:10.1007/s00382-018-4521-8, 2020.

Nielsen, J., Ebba, D., Hahmann, A. N. and Boegh, E.: Representing vegetation processes in hydrometeorological simulations using the WRF mode. [online] Available from: https://orbit.dtu.dk/files/69208136/JoakimRefslundThesis.pdf, 2013.

Powers, J. G., Klemp, J. B., Skamarock, W. C., Davis, C. A., Dudhia, J., Gill, D. O., Coen, J. L., Gochis, D. J., Ahmadov, R., Peckham, S. E., Grell, G. A., Michalakes, J., Trahan, S., Benjamin, S. G., Alexander, C. R., Dimego, G. J., Wang, W., Schwartz, C. S., Romine, G. S., Liu, Z., Snyder, C., Chen, F., Barlage, M. J., Yu, W. and Duda, M. G.: The weather research and forecasting model: Overview, system efforts, and future directions, Bull. Am. Meteorol. Soc., 98(8), 1717–1737, doi:10.1175/BAMS-D-15-00308.1, 2017.

Prein, A. F. and Gobiet, A.: Impacts of uncertainties in European gridded precipitation observations on regional climate analysis, Int. J. Climatol., 37(1), 305–327, doi:10.1002/joc.4706, 2017.

---

## Author Response (AR2)

Point-by-point letter – 2nd revision

Title: "Seasonal and diurnal performance of daily forecasts with WRF V3.8.1 over the United Arab Emirates"

Authors: Oliver Branch, Thomas Schwitalla, Marouane Temimi, Ricardo Fonseca, Narendra Nelli, Michael Weston, Josipa Milovac, Volker Wulfmeyer

**Suggestions for revision or reasons for rejection (will be published if the paper is accepted for final publication)**

**Summary:** I am satisfied with the authors' responses to my comments from the initial review. Overall, the revised manuscript is greatly improved, though there remains a smattering of mostly typo-level issues to fix, and one bigger discrepancy between Figure 6 and Table 5 that must be resolved. Upon correction of those issues, the manuscript should be ready for publication. I look forward to seeing it in print in final form, and I congratulate the authors on assembling this interesting paper.

**Recommendation: Minor Revision.**

**Major Comments:**

*1. Lines 318–320 (the final sentence of the paragraph) and Table 5 agree with each other, but these do not agree with Figure 6. In Figure 6, the highest dewpoints in each season are*

*observed at the marine stations, which makes physical sense. Correspondingly, in each*

*season, the dewpoint depression (air temperature minus dewpoint temperature) is smallest*

*(i.e., the relative humidity is highest) for the marine stations. This also makes physical sense.*

*The dewpoint depressions that I can roughly calculate from Fig. 6 are substantially different*

*from what is listed in Table 5. For example, for summer in Table 5, the mean dewpoint*

*depression for UAE, Mountain, Marine, and Desert are 21.1°C, 17.3°C, 18.2°C, and 16.6°C,*

*respectively. From eyeballing the mean values in Fig. 6, I calculate approximate values of*

*18.5°C, 20.0°C, 11.5°C, and 23.0°C, respectively. Those are pretty different from each other,*

*and tell a different story! It appears that either Fig. 6 or Table 5 is in error as they report*

*contradictory values, but Fig. 6 makes more physical sense to me than Table 5 or lines 318–*

*320. Upon resolving this discrepancy, please revise the text in lines 318–320 accordingly.*

**You are indeed correct. There is an error in the table values from data copying. This has**

**been amended now and corresponds with Figure 6. Thank you for catching this error.**

**The text from L318-320 has now been adjusted slightly to:**

**"In all seasons, the marine region is closer to saturation than in the other regions (T-2m**

**minus TD-2m range is 8.3 to 11°C); however, this contrast is reduced in the cooler seasons**

**as the mountain and desert regions become more humid."**

**Minor Comments and Typos:**

*1. There are still several instances of "WRF-NOAH-MP" leftover throughout the revised*

*manuscript that were not cleaned up from the original manuscript (lines 15, 70, 78). Please*

*change these to "WRF with Noah-MP" or something similar.*

**These have been replaced exactly as you suggest throughout the manuscript.**

*2. Change "NOAH-MP" to "Noah-MP" throughout the manuscript. This is how it is spelled and capitalized by the developers of the scheme, including in the title of Niu et al. (2011), the primary reference for Noah-MP LSM.*

**Done.**

*3. Line 17: Change "10m" to "10-m".*

**Done.**

*4. Line 20: Add commas before and after "however".*

**Done.**

*5. Line 21: I suggest (but do not insist on) changing "lowest" to "smallest." Smallest bias describes magnitude, while lowest bias can imply position on the number line (either less positive or more negative), which is the opposite of what you intend.*

**Done.**

*6. Line 22: Add a comma after "overestimated".*

**Done.**

*7. Lines 45, 48, 309, 410, 419: Add a comma after "e.g.".*

**Done.**

*8. Line 49: Delete "in particular".*

**Done.**

*9. Line 50: Change the dash (-) to an em dash (—).*

**Done.**

10. Line 75: Change "16, 14, and 18 stations" to "17, 15, and 16 stations" to align with Sec. 2.5.1.4, Fig. 1a, and Table 3.

**Done.**

11. Line 79: Uncapitalize both "Materials" and "Methods".

**Done.**

12. Line 80: Add a comma after "configuration".

**Done.**

13. Line 100: Change "heat-low" to "low". Heat lows (thermal lows) are a summertime phenomenon, and do not occur in January in the Northern Hemisphere mid-latitudes.

**Done.**

14. Line 102: Change "heat-low" to "heat low", and delete "appear".

**Done.**

15. Line 120: WRF was already defined and references provided for it back in Sec. 1, so doing so in both places is unnecessary.

***Citation here removed.***

16. Line 138: For completeness, define RRTMG.

**'RRTMG' changed here to '**Rapid Radiative Transfer Model (RRTMG)**'**

17. Line 140: For completeness, define MYNN.

**'the MYNN' changed to '**the Mellor Yamada 2.5 Level  scheme (MYNN)**'**

18. Lines 150 & 158: Change "resolution" to "grid spacing".

**Done.**

*19. Line 158 & Fig. 4 caption: Change "30 arc second" to "30-arcsecond".*

**Done.**

*20. Line 169: Un-bold "Here, we used". Also, there should only be one set of parentheses around the (Danielson and Gesch, 2011) reference.*

**Done.**

*21. Line 175: Change the period after "that time" to a semicolon.*

**Done.**

*22. Line 178: Change "in a NWP mode" to "in NWP mode".*

**Done.**

*23. Line 188: Delete "e.g.,".*

**Done.**

*24. Line 206: Change "2m" and "10m" to "2 m" and "10 m" (add spaces).*

**Done.**

*25. Lines 207 & 221: Correct the date formats to, for example, 30 November 2015. Most of the rest of the dates were changed to this format, but there were a couple stragglers that got missed.*

**Done.**

*26. Line 231: Change "3" to "three".*

**Done.**

*27. Line 248: For MET, add a reference to Brown et al. (2021, https://doi.org/10.1175/BAMS-D-19-0093.1).*

***Done.***

*28. Lines 255–268: I suggest deleting the parenthetical statement on line 255 (and replacing it with a colon), and making the introduction of these three metrics a bulleted list.*

***Done.***

*29. Line 269: Add a comma after "day".*

***Done.***

*30. Line 286: Change "(negative indicate La Niña events)" to "(negative ONI indicates La Niña events)".*

***Done.***

*31. Line 292: Add a comma after "pressure".*

***Done.***

*32. Line 294: Change "than apparent" to "than is apparent".*

***Done.***

*33. Line 295: Change "don't" to "do not".*

***Done.***

*34. Lines 300 & 308: Delete "for instance".*

***Done.***

*35. Line 330: The section heading should be on a separate line.*

***Done.***

*36. Lines 408–409: Change "This is because the method of calculation of transfer coefficients/fluxes are executed in NOAH-MP, the PBL scheme, and the surface layer scheme (SLS) depends on the land surface type." to "This is because, depending on the land surface*

*type, the calculations of transfer coefficients/fluxes are made in Noah-MP, the PBL scheme,*

*or the surface layer scheme (SLS)."*

**Done.**

*37. Line 433: Change "For while" to "While".*

**Done.**

*38. Line 434: Add a comma after "diurnal", and after "differences".*

*39. Line 435: Change "scientifically interesting, and importantly may reveal" to "scientifically*

*interesting. Importantly, this assessment may reveal". (The original sentence is a run-on*

*sentence with far too many clauses connected by "and".)*

**Agreed. This has now been rephrased to:**

**'While assessment of diagnostics for the whole UAE region remains useful, it can obscure**

**regional, diurnal, and seasonal differences, and also compensating biases. These are all**

**scientifically interesting factors. Importantly, they might reveal information on model**

**performance with respect to specific processes and land surface types, and how they are**

**simulated.'**

*40. Line 449: Change "A caveat generally" to "A general caveat".*

**Done.**

*41. Line 451: Change "on" to "in".*

**Done.**

*42. Line 453: Add a comma after "moisture".*

**Done.**

*43. Line 466: Delete the comma after "UAE".*

***Done.***

*44. Fig. 1a: KSA is likely not a commonly known acronym. I suggest replacing it with Saudi Arabia.*

***Done.***

*45. Fig. 1 caption: Change "200 m a.s.l." to "200 m ASL".*

***Done.***

*46. Fig. 2 caption: Change "FIgure" to "Figure", and superscript the -1 in "m s-1".*

***Done.***

*47. Fig. 4 caption: Change "~1 km" to "~1-km", "21 class" to "21-class", and "2 local datasets" to "two local datasets".*

***Done.***

*48. Fig. 5 caption: Change "observation vs forecast" to "forecast vs. observation".*

***Done.***

*49. Fig. 6: In addition to resolving the discrepancy in Major Point 1, I have the following suggestions for improving the readability of Figure 6:*

*a. Have grid lines at consistent intervals. The intermittent and inconsistent grid lines on all three panels currently make it hard to read. I suggest major grid lines (solid light gray, perhaps) every 5°C and every 1 m/s, and minor grid lines (dashed light gray, perhaps) every 1°C and every 0.2 m/s.*

***This figure has been replaced and already has different lines for every 5°C as for 1°C, and different lines for 1 m/s and 0.2 m/s. The high resolution tiff figures are clearer than those pasted into the manuscript.***

*b. The major tick marks need to be clearly labeled, and these tick marks for panels a and b should be multiples of 5 (e.g., 0, 5, 10, 15, etc.), rather than 2, 7, 12, 17, etc. In Fig. 6b right now, it does not appear the tick mark labels are aligned with the major tick marks, either, which also makes it quite difficult to read the figure accurately or confidently.*

**Thank you for this suggestion. However, the authors are satisfied with the scales as they are. Furthermore, this change was not requested in the previous review.**

*50. Table 5: It is not clear to me what "time steps for each period" is. Also, why does it vary so much in autumn between the different regions?*

**To clarify this, in the caption for Table 5, 'time steps for each period' has now been replaced by 'time steps for each season'. The table values were modified already (related to a previous comment) and the values look quite reasonable.**

**Notes to editor:**

**Figures 1 has been replaced with 'KSA' replaced with 'Saudi Arabia'.**

**Figure 6 has been replaced due to an axis being slightly misaligned.**

**Table 5 has been replaced due to the aforementioned errors.**

**All text is consistent with these changes.**

**Finally, the authors thank the reviewers and editor. We trust that these suggestions have greatly improved the manuscript and that it is now ready for publication in GMD.**